# Mapping urban gullies in the Democratic Republic of the Congo

Guy Ilombe Mawe[1,2,3], Eric Lutete Landu[1,3,4], Elise Dujardin[5,6], Fils Makanzu Imwangana[3,7], Charles Bielders[8], Aurélia Hubert[1], Caroline Michellier[8,9], Charles Nzolang[2], Jean Poesen[5,10], Olivier Dewitte[9] & Matthias Vanmaercke[5✉]

Large urban gullies cause damage in many tropical cities across the Global South[1,2]. They can result from inappropriate urban planning and insufficient infrastructure to safely store and evacuate rainfall in environments that are already highly sensitive to soil erosion[1,3,4]. Although they can cause large destruction and societal impacts such as population displacement[1,2,5], the magnitude of this geo-hydrological hazard remains poorly documented and understood[6,7]. Here we provide an assessment of the extent and impact of urban gullies at the scale of the Democratic Republic of the Congo (DRC). Through mapping, we identify 2,922 urban gullies across 26 cities. By combining their formation and growth rates with population density data[8], we estimate that around 118,600 people (uncertainty range: ± 44,400 people) have been displaced by urban gullies over the period 2004–2023. We find that average displacement rates increased from about 4,650 persons yr$^{-1}$ (pre-2020) to about 12,200 persons yr$^{-1}$ (post-2020). Between 2010 and 2023, the number of people living in the potential expansion zone of urban gullies doubled from 1.6 (±0.6) to 3.2 (±1.3) million, with more likely to be exposed due to urban sprawl[9,10] and climate change[11]. We suggest that there is a need for tools and strategies to prevent and mitigate this hazard.

Driven by natural population growth and rural–urban migrations, the Global South is urbanizing at a rapid pace[9]. The resulting expansion of built-up landscapes often happens in informal and uncontrolled ways[10,12,13] without taking into account environmental constraints[11,14]. Among other problems and risks[13,15,16], this can lead to the formation of large urban gullies (UGs) (Fig. 1). Although they have been reported in various Global South countries[2,4,17–19], the Democratic Republic of the Congo (DRC) seems to be particularly affected by them[1,20–23].

UGs are erosional channels that form when the shear stress of concentrated water exceeds the local resistance of the topsoil against incision[6,24,25]. They usually form in environments that may be prone to water erosion through their intense rainfall, steep hillslopes and erodible soils[1,3]. The urbanization of these landscapes leads to the removal of vegetation and increases in rooftop areas and other impermeable surfaces, which can greatly increase runoff production[3,26]. Roads further play a critical part in the formation of UGs, as they can form direct pathways along which runoff can accumulate[21,27]. They can also modify the topography, leading to local increases in contributing area and hence potential runoff volumes[17]. UGs may often be avoided by adequate urban planning and infrastructure[1,4,5,22,25]. Case studies show, for example, that their formation is often linked to insufficient cisterns and infrastructure for capturing rainfall at the parcel level, or to roads built without adequate drainage systems to handle large volumes of concentrated runoff[2,3,21,28]. Yet, the heterogeneous and complex conditions under which UGs form also make them extremely difficult to characterize, predict and anticipate[21,25,26]. This is especially so in the Global South, where lack of data is the main constraint[29,30].

Once formed, many UGs continue to greatly expand over subsequent large rainfall events through gully head retreat, gully widening and/or the formation of additional bank gully heads[1,25,26,31] (Fig. 1). Given the often enormous size of UGs, their location in densely populated environments and the erratic nature of expansion events, UGs frequently lead to severe impacts. These include damage to property and infrastructure, the displacement of population and casualties[1,2,4,5]. As UGs often form in poor, informal (peri-)urban areas, they frequently affect highly vulnerable populations[4,28,32,33]. Yet, our insight into these impacts remains limited to local case studies and reports in grey literature and (social) media[1,5,22]. UGs and their effects have not yet been thoroughly examined, either for the DRC or for other nations. The overall scale of this problem, therefore, remains poorly quantified and understood.

As with other natural hazards, a better understanding of the magnitude and drivers of the risks associated with UGs is an important step towards disaster risk reduction[34]. Here we address this need by providing one of the first comprehensive assessments on the occurrence of UGs, their expansion rates, their impacts on population displacement and the number of people exposed to this hazard. We conduct

[1]Department of Geography, University of Liège, Liège, Belgium. [2]Department of Geology, Université Officielle de Bukavu, Bukavu, Democratic Republic of the Congo. [3]Department of Geosciences, Université de Kinshasa, Kinshasa, Democratic Republic of the Congo. [4]Department of Natural Resources Management, Université de Kinshasa, Kinshasa, Democratic Republic of the Congo. [5]Department of Earth and Environmental Sciences, KU Leuven, Leuven, Belgium. [6]Research Foundation Flanders – FWO, Brussels, Belgium. [7]Geomorphology and Remote Sensing Laboratory, Geological and Mining Research Center, Kinshasa, Democratic Republic of the Congo. [8]Earth and Life Institute - Environmental Sciences, Université catholique de Louvain, Louvain-la-Neuve, Belgium. [9]Department of Earth Sciences, Royal Museum for Central Africa, Tervuren, Belgium. [10]Institute of Earth and Environmental Sciences, Maria-Curie Sklodowska University, Lublin, Poland. ✉e-mail: matthias.vanmaercke@kuleuven.be

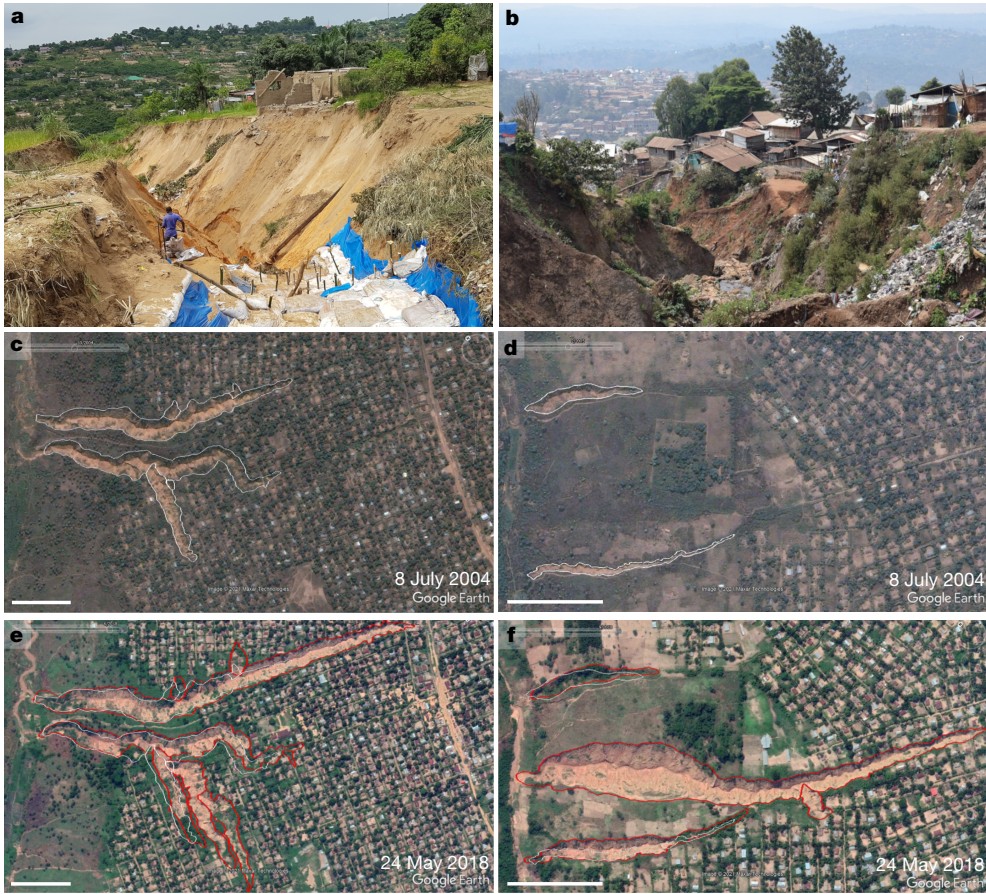

**Fig. 1 | Examples of large UGs and the destruction they cause in the DRC.** **a**, Tchad gully (Kinshasa, 26 November 2019). **b**, Funu gully (Bukavu, 8 July 2021). **c**,**d**, Google Earth imagery of UGs in Kikwit (8 July 2004). Gully extents are mapped with a white polygon. **e**,**f**, The same gullies on 24 May 2018 with their new extent mapped in red. Scale bar, 150 m (**c**–**f**).

our analyses at the scale of DRC, probably one of the countries most affected by this problem[1,21,23,35].

## Extent and controlling factors

Through systematic analyses of recent very high spatial resolution satellite imagery (Methods, 'Identifying cities significantly affected by UGs' and 'Mapping the extent and expansion rates of UGs'), we identified and mapped 2,922 UGs across 26 out of 47 cities (Fig. 2 and Methods, 'Identifying cities significantly affected by UGs'). Cross-checking with historical, panchromatic aerial photographs from the 1950s (that is, predating the development of most built-up areas) confirmed that most of these gullies are linked to urban sprawl and road construction. Despite the observation that deforestation was already well advanced in most areas, only 46 of the mapped gullies were already present in the 1950s (Supplementary Fig. 1). In all cases, these existing gullies were connected to roads and/or had a substantial number of buildings in their direct vicinity.

Although the dimensions of these UGs vary largely within and between cities (Supplementary Fig. 2), they are predominantly large, with an average length of 253 m and an average width at the widest point of 31 m. Combined, these UGs have a total length of 739 km. Especially Kinshasa is severely affected, with 868 UGs and a combined gully length of 221 km (Fig. 2 and Supplementary Table 1).

Analyses (Methods, 'Assessing the factors controlling UG occurrence') show that UGs typically occur in areas that are somewhat steeper and sandier than areas without UGs (Fig. 3a). Overall, 17 out of the 26 affected cities are located on the Kwango-Kwilu and Kasai Plateaus (Fig. 2), which are mainly covered by deep arenosols and other sandy soils that are susceptible to gully formation[24]. Nonetheless, steep and sandy soils are not a strict prerequisite. Other key factors explaining the contrast between areas with and without UGs are vegetation cover, the fraction of built-up area and road density. The important role of roads in explaining the occurrence of UGs was also confirmed by our mapping efforts: 98% of all mapped UGs were connected to the local road network, either by being formed along a road (48%) or by directly receiving runoff from a road (50%). Combined, slope steepness, soil type, the fraction of tree cover, the fraction of built-up area and road density well explain the presence or absence of UGs (Fig. 3b and Supplementary Figs. 3 and 4). Although previous research demonstrated the strong impact of rainfall on gully formation and expansion[25,31,36], rainfall characteristics did not further improve our model. This is probably due to the overall rather limited range of the investigated rainfall variables across DRC as well as the limited spatial resolution and accuracy of the rainfall products available (Supplementary Table 2).

## Expansion, displacement and exposure

By digitizing the spatial extent of UGs on different dates, we reconstructed the spatial expansion history of each urban gully (Methods, 'Mapping the extent and expansion rates of UGs'). We observed that 2,897 (that is, 99%) of the UGs experienced detectable (>10 m²) spatial expansion between 2004 and 2023 (Supplementary Table 1). Especially in Kinshasa, Kananga, Tshikapa, Mbuji-Mayi and Kikwit, UGs showed important expansion over this period, accounting for 70% of the total observed expansion (Supplementary Table 4). This UG

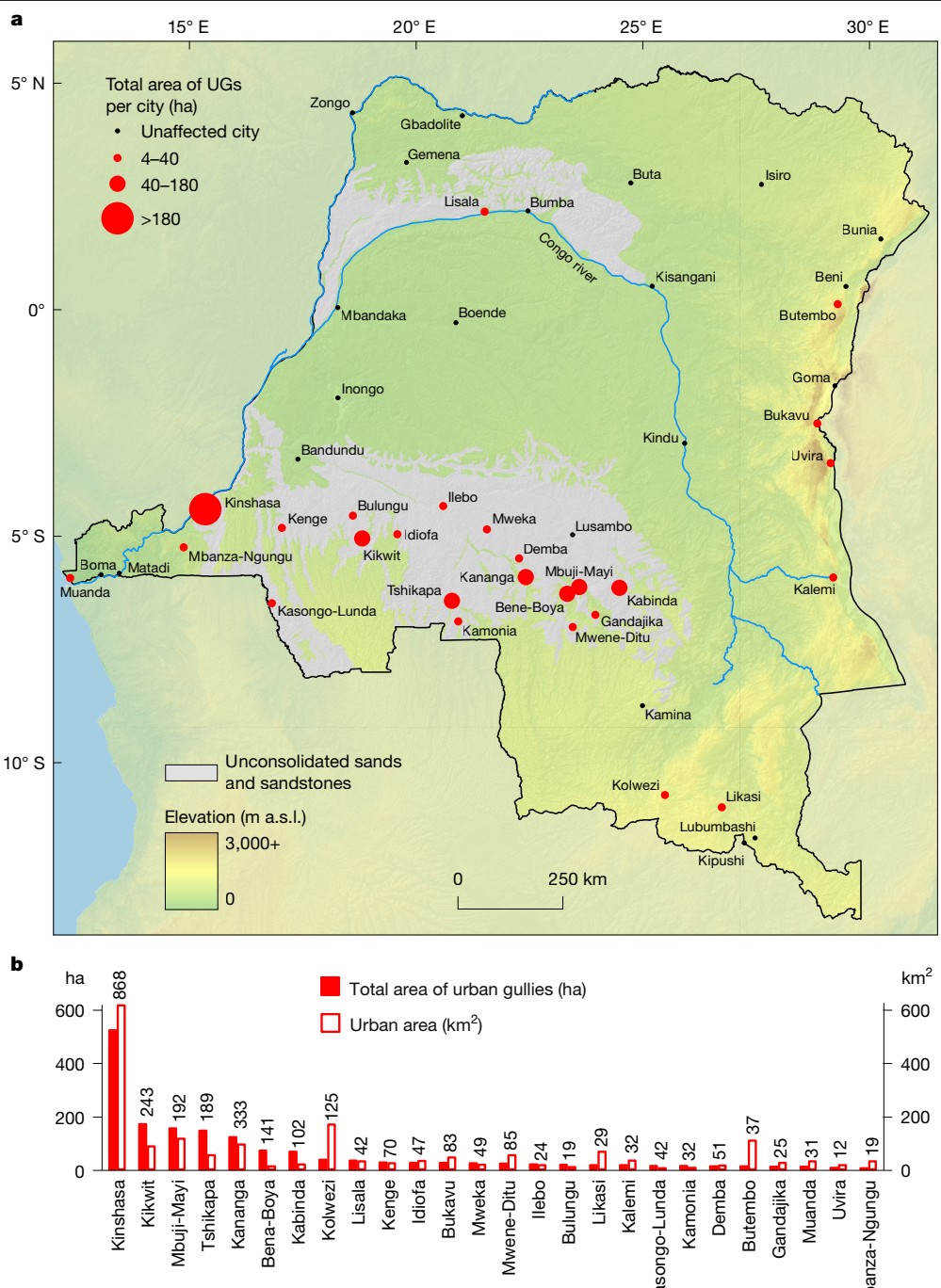

**Fig. 2 | Occurrence of UGs in the DRC. a**, All cities investigated for the presence of UGs. Cities marked in red (*n* = 26) are markedly affected, that is, ≥10 UGs were detected (Methods, 'Identifying cities significantly affected by UGs'). Seventeen affected cities are located on the Kwango-Kwilu and Kasai Plateaus (indicated in grey[49]), which are mainly characterized by sandy soils. Colour gradient indicates the elevation[50]. **b**, Total spatial extent of the mapped UGs and affected cities. Numbers above the bars indicate the total number of UGs in that city, as mapped on satellite imagery from 2021 to 2023 (Methods, 'Mapping the extent and expansion rates of UGs').

expansion is often associated with the destruction of houses (Fig. 1) and population displacement. By combining all observed cases of gully expansion across all cities with population density estimates (Methods, 'Estimating the displaced population'), we estimate that about 118,600 (±47,440) people were directly displaced by the formation and expansion of UGs between 2004 and 2023 (Fig. 4b). This corresponds to an average of about 5,930 persons yr$^{-1}$. However, more recently (2020–2023), this rate accelerated, with an average of approximately 12,200 persons yr$^{-1}$ (Fig. 4b).

Most of the 8,161 mapped UG expansion events displaced a relatively limited number of people (that is, 97.3% of the events probably displaced less than 100 persons; Fig. 4a). Combined, these events account for 64.8% of the total estimated displacement; 217 events (2.6%) displaced between 100 and 1,000 persons, accounting for around 33.1% of the total. Two events (1.3 ha and 3.4 ha) potentially displaced more than 1,000 persons (Fig. 4a). These numbers are prone to important uncertainties, especially because they rely on estimated population data (Methods, 'Uncertainty assessment'). Nonetheless, visual inspection

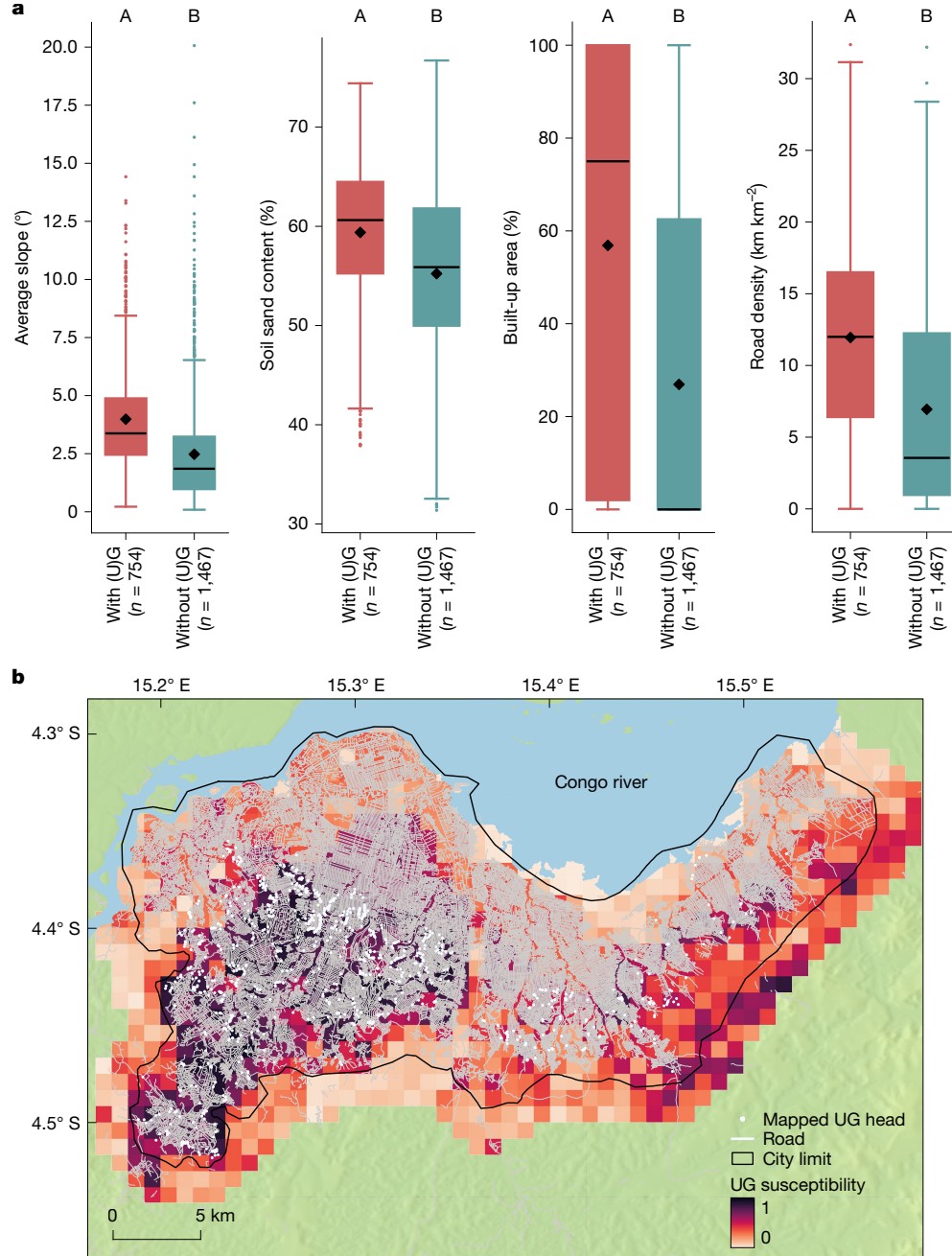

**Fig. 3 | Factors explaining the occurrence of UGs. a**, Boxplots showing the distribution of average slope steepness, sand content, built-up area and road density for cells of about 1 km² with a (urban) gully present ("with (U)G") and cells without such gullies ("without (U)G"). The latter were randomly sampled inside and around city limits (Methods, 'Assessing the factors controlling UG occurrence'). Outside the city limits, gullies were only detected in two cells. These were included in the "with (U)G" group. Rectangles of boxplots indicate the 25% and 75% quartiles of the data, whereas black lines indicate the median. Whiskers were calculated as 1.5 times the interquartile range. Black diamonds indicate the average. Different letters above the boxplots indicate a significant difference in distribution ($P < 0.0001$; Methods, 'Assessing the factors controlling UG occurrence'). **b**, UG susceptibility model applied to Kinshasa and its surroundings (Methods, 'Assessing the factors controlling UG occurrence', Supplementary Figs. 3 and 4 and Supplementary Table 3). Colours indicate the predicted susceptibility, ranging from low (light) to high (darker).

of satellite images confirmed that often tens to hundreds of houses are destroyed by individual expansion events.

To better understand the dynamics of population displacement, we further subdivided all mapped expansion events into three sub-processes: the formation of new gullies, gully head retreat and sidewall widening (Methods, 'Estimating the displaced population'). The formation of new gullies accounts for a relatively small fraction (11% of the expansion events) and typically happens in areas with a lower population density (Fig. 4), displacing on average about 47 persons ha⁻¹.

Consequently, they are responsible for only 2.8% of the estimated total displaced population. Gully head retreat is observed more frequently and typically occurs in more densely populated areas, accounting for about 31.9%. However, gully sidewall widening is the most prevalent process, in terms of both frequency and associated areal expansion rates. As this widening also often occurs in densely populated areas, displacing on average around 131 persons ha⁻¹, it accounts for about 65.3% of the estimated displaced population. Furthermore, its importance seems to have increased over recent years (Fig. 4b).

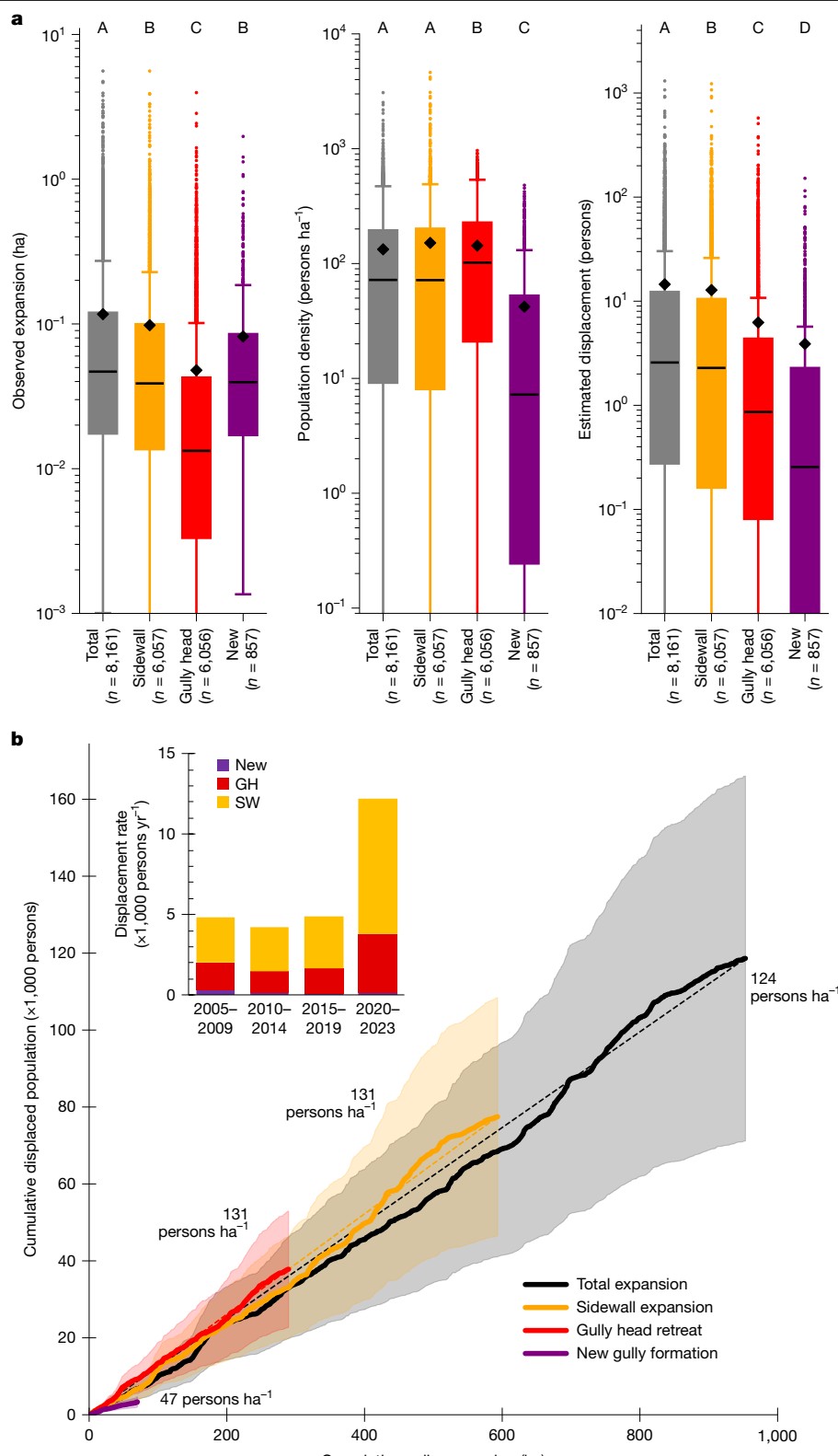

**Fig. 4 | Urban gully expansion and its associated population displacement.**
**a**, Boxplots showing the distribution of the size of observed gully expansion events, estimated population density within expansion areas and estimated population displacement (Methods, 'Estimating the displaced population'). A distinction is made between expansion due to sidewall widening (SW), gully head retreat (GH) and the formation of new gullies (New). 'Total' refers to the entire observed expansion event. Rectangles of boxplots indicate the 25% and 75% quartiles of the data, whereas black lines indicate the median. Whiskers were calculated as 1.5 times the interquartile range. Black diamonds indicate the average. For visualization purposes, *y*-axes are truncated and do not always show the full range of observations. Boxplots with a different letter are significantly different (*P* < 0.0001; Methods, 'Estimating the displaced population'). **b**, Cumulative UG expansion compared with cumulative population displacement. Observed areal expansion events (between 2004 and 2023) are ordered chronologically according to their estimated date (Methods, 'Estimating the displaced population'). Transparent zones around each curve indicate the estimated overall error range of ±40% (Methods, 'Uncertainty assessment').

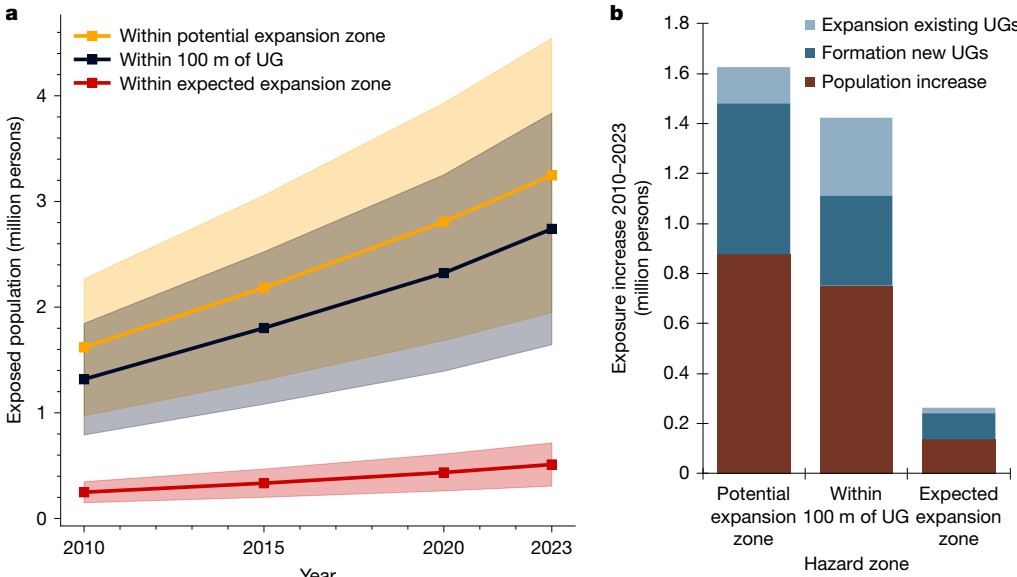

**Fig. 5 | Evolution of the population exposed to UG expansion. a**, Population living less than 100 m away from a UG, within the potential expansion zone of a UG and within the expected expansion zone (Methods, 'Estimating the exposed population'). Transparent zones indicate the estimated overall error range of ±40% (Methods, 'Uncertainty assessment'). **b**, Causes of increases in population exposure between 2010 and 2023 (Methods, 'Estimating the exposed population'). 'Population increase' refers to population growth within hazard zones already present in 2010. 'Expansion of existing UGs' refers to increases in hazard zones due to the expansion of UGs already present in 2010. 'Formation of new UGs' refers to increases in hazard zones due to UGs that were not present in 2010 but have formed and expanded since then. As errors on the population estimates across different years are unknown, the uncertainties on the individual causes of exposure increase could not be quantified (Methods, 'Uncertainty assessment').

Using our inventory of UGs, we also quantified the population exposed to UG expansion (Methods, 'Estimating the exposed population'). In 2023, 2.7 (±1.1) million persons lived less than 100 m away from a UG (Fig. 5a and Supplementary Table 5). This arbitrary 100-m threshold can be considered a simple yet conservative estimate of the zones within which people are confronted with the direct and indirect consequences of UGs. In the same year, an estimated 3.2 (±1.3) million people lived within the potential expansion zone of UGs (Methods, 'Estimating the exposed population') and faced risks of being directly displaced because of UG expansion (Fig. 5a). Of these people, around 550,000 (±220,000) persons lived within the expected expansion zone of UGs (Methods, 'Estimating the exposed population'), facing very high risks of losing their housing. These risks remain difficult to quantify. Yet, comparing the exposed population in 2020 (Fig. 5a) with the average annual displacement rate in the period 2020–2023 (approximately 12,200 persons yr$^{-1}$; Fig. 4b) indicates that around 4% of the population living in the potential expansion zone are likely to be displaced within the next 10 years. For the population living within 100 m of a UG, this rises to about 5%, whereas for people living within the expected expansion zone, this is 28%.

Overall, the population exposed to UG expansion probably doubled over the period 2010–2023 (Fig. 5a). We observe this for all three considered hazard zones. Further analyses (Methods, 'Estimating the exposed population') indicate that an estimated 52–54% of this increase is attributable to population increases within hazard zones already present in 2010. An estimated 25–39% is attributable to the formation of new UGs since 2010, whereas 8–22% is attributable to the further expansion of UGs already present in 2010 (Fig. 5b).

## A new geo-hydrological hazard

Our results demonstrate the massive magnitude of the problem of UGs. With more than half of the investigated cities affected and more than 2,900 gullies mapped (Fig. 2), UGs are a widespread phenomenon. Their large size, but especially their position within densely populated

areas, makes them an important threat to many people. We estimate that in 2010, about 1.9% of the total Congolese population lived within 100 m of a UG[8] (Fig. 5). By 2023, this number increased to about 2.6%. Comparisons with other hazards show that these exposure levels are certainly not negligible. For example, an estimated 1.1% of the total African population in 2015 was exposed to a 100-year flood event within the next 10 years, whereas around 0.4% was exposed to a major earthquake within the next 10 years (ref. 37). Around 3% of the African population lived within 100 km of an active volcano. The fraction of the exposed population that may experience an eruption over the next 10 years is unknown, but probably much lower[38]. Also, in terms of impacts, UGs are notable. We estimate that, since 2020, around 12,200 persons per year have lost their houses because of UG expansion (Fig. 4b). This corresponds to about 50 persons per 100,000 urban inhabitants per year over all significantly affected cities in 2023 (Fig. 2 and Supplementary Table 4). By comparison, rainfall-triggered landslides claim up to six fatalities per 100,000 inhabitants per year in populated rural landscapes in the eastern DRC[39]. Landslides are a different type of hazard that comes with its own challenges and impacts[30,40]. Nonetheless, this comparison suggests that the number of people severely affected by UGs is certainly not to be overlooked.

Moreover, our results show that the problem of UGs is growing. The estimated exposed population doubled between 2010 and 2023 (Fig. 5). This growth is likely to continue and even accelerate over the next decades. First, the African urban population is expected to nearly triple by 2050 (ref. 9). As our analyses show that the occurrence of UGs is closely linked to built-up area and road density (Fig. 3 and Supplementary Figs. 1, 3 and 4), the population potentially exposed to UGs may be expected to increase accordingly. Climate change may further aggravate the problem. Future projections indicate that rainfall intensities in tropical Africa may increase by 10–15% in the coming decades[41]. Yet, intensive rainfall events are also a main driver of gully formation and expansion[25,31,36]. A meta-analysis of observational data worldwide suggests that these predicted increases could easily double gully expansion rates if other factors remain the same[31]. Hence, the

threats and impacts of UGs are likely to greatly increase over the next decades. This is probably true not only for the DRC but also for many other regions in the (sub-)tropical Global South. The growing number of studies reporting on the occurrence of UGs in other countries seems to confirm this[2,4,18,19,42,43].

UGs are also distinctly different from other, better-studied geo-hydrological hazards such as floods or landslides[15,16,40] and therefore require more explicit consideration in disaster risk reduction strategies. First, they tend to occur in different areas. Hillslopes or plateaus that are not necessarily prone to landslides or flooding may be highly susceptible to UGs (Figs. 1 and 3). Nevertheless, UGs can also occur together with landslides or floods and even aggravate overall impacts as compound events[26,44]. Second, UGs are—once formed—a persisting problem. As our results show, most of the population displacement is attributable to the continued expansion, and especially widening, of existing UGs over a period of years to decades (Fig. 4). Likewise, observed increases in exposure rates are not just due to the formation of new UGs but mainly due to population increase in already existing hazard zones (Fig. 5b). Case studies in Kinshasa indicate that this further densification of built-up areas is one of the main drivers that keep gully expansion ongoing[26]. Given that many UGs develop in overall poor and unplanned urban neighbourhoods[5,28,35] (Supplementary Fig. 8), the long-term nature of these threats may also further affect the livelihoods of people who are socioeconomically already very vulnerable[13,33]. For example, field surveys in Kinshasa, Kikwit and Bukavu indicated that people living in the vicinity of UGs often invest substantial amounts of money, labour and other resources, aiming to stabilize UGs[28,35]. Living next to a UG also comes with numerous other long-term consequences and indirect impacts, including impeded traffic flow, reduced property value, sanitation concerns, psychological stress and social unrest[2,4,5]. Hence, the consequences of UGs reach beyond direct displacement. Nonetheless, these impacts remain poorly researched.

## Preventing and mitigating UGs

Our mapping efforts showed that UGs frequently initiate on hillslopes in peri-urban zones (Fig. 1c,d) as a result of runoff accumulation due to upslope urbanization and road construction[1,17,21] (Supplementary Fig. 9a). As with gullies in other environments[31], the initial growth phase of UGs can be rapid. Yet, as this often occurs in urban outskirts with lower population densities, the direct displacement caused by the formation of UGs mostly remains limited (Fig. 4). Nevertheless, the rapid and often unexpected nature of UG formation can already pose dangers to the population living in initiation zones. Importantly, this population is not included in our exposure assessment (Fig. 5) because the exact locations and timing of UG formation now remain impossible to predict.

In early phases, UGs mainly expand through gully head retreat (Supplementary Fig. 9b). Many gully heads advance at rates of tens of metres per year (Supplementary Fig. 2), often due to a few large rainfall events[1,26,31], and can cause significant destruction and casualties[5] (Fig. 1). Local mitigation measures are often undertaken to stop or slow down this process. Yet, owing to the limited means available and the large volumes of water associated, most of these initiatives fail[35]. As the gully head further migrates upslope, its contributing area—and by extent the amount of runoff that can accumulate at the head—will decrease, eventually halting the process. Increasing efforts to stabilize the gully head may contribute to this. Nonetheless, UGs usually continue expanding through sidewall widening (Supplementary Fig. 9c). This is overall the most frequently observed form of gully expansion and can continue for years to decades. Gully widening is therefore responsible for nearly 70% of the estimated total displaced population (Fig. 4). Furthermore, the steep banks formed by the large gully channel, the destruction of water infrastructure (for example, water pipes and road drainage channels) and badly implemented gully stabilization efforts, may give rise to new

gully heads that start branching off from the existing UG[24] (Fig. 1). This may further increase the population exposed to both gully widening and head retreat and even initiate a new cycle of UG development.

Understanding these dynamics is relevant for the prevention and mitigation of UGs and their impacts. For example, although many initiatives now focus on stabilizing the gully head[1,35], our results demonstrate that sidewall stabilization is at least as important (Fig. 4). Similar to gullies in other contexts, vegetation may play a crucial role here[45–47]. Nonetheless, given their size and context, stabilizing UGs with vegetation alone is highly challenging. Our mapping efforts indicate that older UGs with vegetation tend to be more stable (Fig. 1). Yet, it is not always possible to distinguish cause from effect here. Furthermore, we observed various cases in which phases of vegetation establishment were disrupted by new expansion events. Overall, integrated approaches are needed that combine both structural and vegetative measures in the gullies with initiatives to capture, store and safely divert runoff upslope[35,46,47]. These measures need to be taken as early as possible because the length of the gully will determine the extent of subsequent gully widening and, consequently, the overall impacts.

Successfully stabilizing UGs does not come cheap. Costs to stabilize a single gully can easily exceed 1 million US$ (ref. 5). Considering this and the devastation and impacts they cause, prevention emerges as an essential and potentially more viable approach. Although UGs partially depend on natural conditions, their formation seems to be nearly always directly linked to human landscape interventions (Fig. 3 and Supplementary Fig. 1). Removal of vegetation, unplanned urbanization on steep slopes, inadequate road construction and/or insufficient water storage or evacuation facilities are frequently cited as root causes[1,3,6,17,21,26]. This indicates that prevention is achievable through adapted infrastructure and improved spatial planning[28]. A critical element towards such planning will be the development of models that can better predict where and when UGs may occur[21,25,28].

Given the demographic and socioeconomic context of DRC and many other Global South countries susceptible to UGs, implementing this planning will be challenging[12–14,48]. Most of the ongoing rapid urbanization happens in ways that leave new citizens with few options for choosing the location of their residence[11,33]. This is already illustrated by the fact that more than half of the increase in exposed population between 2010 and 2023 is attributable to population growth in already existing hazard zones (Fig. 5b). As such, holistic approaches will be needed to thoroughly tackle this problem. A better awareness of this largely neglected geo-hydrological hazard is an essential step in this.

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

# Article

## Methods

### Identifying cities significantly affected by UGs

We first identified all cities in the DRC that were significantly affected by UGs. For this, we checked all urban centres that were assigned the official status of 'city' by presidential decree (Articles 53–55 of Decree Law 081 of 2 July 1998) as well as other urban centres with at least 80,000 inhabitants in 2020 (according to ref. 51) that show characteristics of small cities. This list can be considered exhaustive.

The presence of UGs was checked in all these cities using available Google Earth imagery of very high resolution (that is, a resolution of 1 m or smaller; Supplementary Table 1). We considered a feature to be a UG if it could be recognized as such, based on commonly accepted geomorphic criteria[25,52]. More specifically, the feature had to be recognizable as a channel eroded by concentrated runoff with an elongated shape, a discernible thalweg, a gully head and visible gully edges. Furthermore, the thalweg needed to be oriented along the steepest slope or in another way that allowed effective runoff evacuation (for example, following a road downslope). Moreover, the gully needed to be located within 200 m of buildings. Field surveys of 434 gullies in Kinshasa, Kikwit and Bukavu confirmed that all mapped features were UGs. Nonetheless, these field visits also showed several smaller UGs that were not detected. We, therefore, restricted our analyses to UGs with a thalweg of at least 30 m to avoid potential biases caused by contrasts in detection accuracy.

To verify whether the detected UGs are linked to urban growth, we checked high-resolution panchromatic aerial photographs of each city, taken in the 1950s. These photographs are conserved at the Royal Museum for Central Africa in Belgium[53]. If identified gullies were already present, we examined whether they were inside or outside built-up areas and, when outside, whether they were linked to the road network (that is, the gully formed along a road or lies in its direct extension, within a distance of about 100 m or less). Gullies observed in the 1950s that were located outside built-up areas and not related to the road network were assumed to be of natural origin. They were not retained for further analysis.

### Mapping the extent and expansion rates of UGs

In each affected city (Methods, 'Identifying cities significantly affected by UGs', Fig. 3), we manually mapped the polygons delineating the spatial extent of all UGs on a reference image (Supplementary Fig. 5). This reference image was a cloud-free, very high-resolution image available in Google Earth that was taken between 2021 and 2023 (Supplementary Table 1). Given that many UGs evolve into branched networks of multiple gully heads (Fig. 1), we considered a branching feature as an individual UG if it had an identifiable gully head and a thalweg of at least 30 m long. For most cities, the limits of the UGs could be identified and mapped because of the clear visual contrast between the gully channel and the surrounding environment. A notable exception was the city of Bukavu, for which the Google Earth imagery did not always allow a clear delineation. This is attributable to the clayey soils on which the city is built[54], resulting in insufficient visual contrast. For this city, the mapping was complemented with handheld GPS field surveys.

Next, the areal expansion rates of the UGs were quantified by remapping their limits as observed on older images available in Google Earth of adequate quality (Supplementary Fig. 5). For Kinshasa and Kikwit, this imagery was complemented with Pléiades images (taken on 21 April 2015 for Kikwit and 28 April 2014 or 19 June 2015 for Kinshasa). Depending on their age and the availability of imagery, UGs were digitized one to five times, with image dates ranging between 2002 and 2023. For each gully observed on at least two images with different dates, the gully expansion was calculated by subtracting the area of the gully polygon, as mapped on the older image, from the area on the more recent image. We defined an expansion event as an urban gully showing an increase in mapped extent of at least 10 m² between two image

dates. It should be noted, however, that this observed expansion may, in reality, be attributable to several consecutive, smaller expansion phases that occurred between the two image dates.

For UGs that were newly formed during the observation period, the areal expansion was assumed to be the area of the polygon as mapped on the first image in which the gully was visible. For expansion events of already existing gullies, we further differentiated between gully head retreat (GH) and sidewall widening (SW). Similar to ref. 26, we considered GH to be the part of the gully expansion that occurred upslope of the gully head on the oldest image (demarcated with a straight line, perpendicular to the gully thalweg; Supplementary Fig. 5). SW was quantified as the expansion that took place downslope of this gully head. This way, each mapped gully expansion event could be attributed to the formation of a new gully, gully head retreat and/or gully sidewall widening.

For gullies with three or more suitable images available, two to four expansion rates were calculated using pairs of subsequent images and the procedure described above. Yet, the exact dates on which the observed gully expansion occurred are mostly unknown. To reconstruct the cumulative expansion of gullies over time (Fig. 4b) and estimate the displaced population (Methods, 'Estimating the displaced population'), we, therefore, assumed that each expansion event took place halfway between the two image dates. For newly formed gullies, the formation date was assumed to be the average between the date of the last available image in which the UG was absent and the date of the first image in which it was present.

### Assessing the factors controlling UG occurrence

We conducted bivariate and multivariate analyses to assess the factors that help us to explain the spatial patterns of UGs across DRC. Owing to data constraints (for example, lack of high-resolution digital elevation models) and to allow robust comparisons with areas not affected by UGs, we conducted these analyses at a resolution of 30 arcseconds (about 1 km at the equator).

We first converted our gully inventory to cells of this resolution. A cell was classified as being affected by UGs if it contained one or more gully heads as mapped on the reference image (Methods, 'Mapping the extent and expansion rates of UGs'). This resulted in a dataset of 752 cells in which UGs occur. Next, we randomly generated cells in affected and non-affected cities across the DRC (Methods, 'Identifying cities significantly affected by UGs'), both inside the city limits and in a 2-km buffer around them. We then checked in Google Earth whether signs of gullies were present in these cells. In total, we checked 1,469 additional cells: 754 within the city limits (which all had no gullies) and 715 in a 2-km buffer around it. Only two cells of the latter were found to have gullies and were added to the cells having a UG, bringing the total number of cells with a gully to 754 and the number of cells without to 1,467.

For each cell, we extracted a set of variables from different geospatial datasets[49,55–61] that might be relevant in explaining the presence or absence of gullies (Supplementary Table 2). Most of these variables are commonly used in other studies aiming to explain the occurrence of gullies[25]. Using two-sided Mann–Whitney $U$-tests[62], we analysed to what extent the distribution of these variables differed significantly between cells with or without gullies.

We then constructed a logistic regression model[63] that combines the variables best explaining observed differences between affected and non-affected cells, making sure each variable remained significant within the model. For this, we first standardized all predictor variables to values between 0 and 1, based on the observed minimum and maximum across all cells. Next, we applied a backwards stepwise selection procedure, in which we first fitted a logistic regression model based on all remaining variables (Supplementary Table 2) and then systematically removed all variables that were not significant ($P$-value of $Z$-statistic >0.0001). We also excluded the variable Sand_iSDA (Supplementary Table 2) from the model, as it was probably subject to important uncertainties[59] and strongly correlated to the more robust dummy variable

Soil, which was also highly significant and functionally expressed the same property. The variables that were finally retained in the model were Slope, Soil, Tree_Cover, BUA and Road (cf. Supplementary Table 2).

The overall performance of the model was tested based on a cross-validation, in which we made ten 70:30% random splits between calibration and validation data. Each time, the four retained variables were standardized based on the range of the training data and their corresponding coefficients refitted (Supplementary Table 3). The resulting alternative model was then applied to the independent test data. We used a validation receiver operating characteristic curve and the corresponding area under the curve as a proxy for model performance[64] (Supplementary Fig. 3).

### Estimating the displaced population

To estimate the population that was probably displaced by the formation and expansion of UGs, we made use of JRC GHS population data[8], that is, a series of gridded datasets that provide estimates of population density at a spatial resolution of about 100 m. This dataset was chosen as it covers the whole territory of the DRC and the entire observation period of our study. Furthermore, it was positively evaluated by previous studies[65].

The displaced population caused by each gully expansion event was calculated as

$$DP = (Area_{recent} * PopDens_{recent}) - (Area_{old} * PopDens_{old}) \quad (1)$$

where DP is the total population expected to be displaced due to an observed case of gully expansion or formation; $Area_{recent}$ is the area of the UG as mapped on the most recent of the two images; $Area_{old}$ is the area of the UG as mapped on the oldest of the two images; and $PopDens_{recent}$ and $PopDens_{old}$ are the average population densities in the polygons corresponding to $Area_{recent}$ and $Area_{old}$, respectively (Supplementary Fig. 5). PopDens values were derived from the JRC GHS population data[8], considering the year that the expansion event was expected to take place (Methods, 'Mapping the extent and expansion rates of UGs'). As these data are originally available for 5-year intervals (for example, 2000, 2005), estimates for in-between years were obtained by first linearly interpolating these raster datasets. Summing the DP values for all (chronologically sorted) cases of UG expansion allowed us to calculate the cumulative number of displaced persons (Fig. 4b).

A similar strategy was applied to differentiate between population displaced by the formation of new gullies (New), sidewall widening (SW) or gully head retreat (GH). For the first, $Area_{old}$ was assumed to be zero. For SW, we only considered the area downslope of the previously mapped gully head when calculating $Area_{recent}$. For GH, we only considered the area upslope of the previously mapped gully head when calculating $Area_{recent}$, whereas $Area_{old}$ was assumed to be zero. The population density of each expansion event was calculated by dividing DP by the areal extent of the event. We tested whether the distributions of areal expansion, population density and expected displaced population were significantly different between types of gully expansion (that is, New, SW, GH or Total; Fig. 4a) using two-sided Mann–Whitney $U$-tests[62].

### Estimating the exposed population

To estimate the population exposed to UG expansion in a given year, we considered all mapped gully polygons that were present in that year (Methods, 'Mapping the extent and expansion rates of UGs') and generated hazard zones around them, using different buffer distances (Supplementary Fig. 5). The total exposed population was then calculated as

$$PE = \sum_{i=1}^{n} (PopDens_{year,i} \times AH_i) \quad (2)$$

where PE is the total population exposed to the expansion of UGs, $n$ is the number of hazard zone polygons, $PopDens_{year,i}$ is the population density of hazard zone polygon $i$ in the considered year according to the JRC GHS population data[8], and $AH_i$ is the area of hazard zone polygon $i$. Mapped gully polygons were excluded from the hazard zone polygons, and overlapping polygons were only counted once.

To account for various degrees of exposure, different buffers were considered when creating these hazard polygons. First, we applied a buffer distance of 100 m around the mapped contours of the gullies. Although somewhat arbitrary, this distance provides an intuitively understandable estimate of the population for which the threats of UG expansion are a regular and significant concern. These people are potentially exposed to direct impacts such as damage to their property and/or displacement, but also to numerous indirect and intangible impacts (for example, decreased housing property value, required investments in initiatives to counter further gully expansion, decreased accessibility, increased stress). Almost no data on these indirect impacts are currently available. However, our mapping efforts and field surveys indicate that this distance of 100 m is probably still a highly conservative value. For example, we also observed that people living several hundred metres away from a UG are involved in implementing measures aiming to stop gully expansion. Usually, they do so at their own expense[28,35].

Second, we generated buffer areas that characterize the potential expansion zones of the UGs. People living within these buffers are expected to be directly exposed to potential property damage, displacement and even injury or death. The potential expansion areas were quantified by considering both gully widening and gully head retreat. For widening, we analysed the maximum widths of all UGs that were at least 10 years old as mapped on the reference images (Methods, 'Mapping the extent and expansion rates of UGs'). These widths varied strongly within and between cities (Supplementary Fig. 2). Nonetheless, UGs in sandy substrates (see 'Soil' in Supplementary Table 2; Fig. 2) were significantly wider than those formed in other substrates according to a two-sided Mann–Whitney $U$-test ($P < 0.0001$; Supplementary Fig. 2). Hence, we treated these two samples differently and considered the 95% quantile as the expected maximum width a gully may attain in this substrate (that is, 70 m for UGs in sandy substrates and 51 m for UGs in non-sandy substrates). We then generated buffer areas around the thalweg of each mapped UG with distances equal to half of these expected maximum widths.

A similar strategy was used for the potential gully head retreat. We assessed the average linear gully head retreat rate (quantified as the Euclidean distance over which the head retreated, divided by the observation period) of all UGs for which the available imagery allowed us to do so over a period of a minimum of 10 years (Methods, 'Assessing the factors controlling UG occurrence'). As with the maximum gully widths, these expansion rates varied considerably within and between cities but could be robustly grouped in UGs formed in sandy substrates and UGs formed in other substrates based on a two-sided Mann–Whitney $U$-test (Supplementary Fig. 2; $P < 0.0001$). We considered the 95% quantiles of these two populations (that is, 19.2 m yr$^{-1}$ for sandy and 10.6 m yr$^{-1}$ for other substrates) and multiplied them by 10, as we aimed to assess the population exposed to gully expansion at a decadal timescale. The resulting distances (192 m or 106 m) were used to generate buffer areas around the gully heads present in the considered year. We chose circular buffer areas because the exact direction of gully head retreat is impossible to predict. Numerous gully heads changed direction or bifurcated as they retreated, depending on the road network, local topography and other factors that are hard to quantify (Fig. 1). We then merged these circular buffers around the head with the maximum width buffers around the thalweg and subtracted the mapped UG polygons from them. The resulting polygons indicate the land areas that are potentially prone to further gully expansion and were used to calculate $AH_i$ (equation (2) and Supplementary Fig. 5).

Third, we generated hazard zones that characterize the expected expansion zones of UGs. People living in these areas are severely

exposed to the impacts of UG expansion. For this, we followed the same strategy as for the potential UG expansion zones. Yet, we considered the averages rather than the 95% quantiles to create the buffers. For the maximum gully widths, these corresponded to 32 m in sandy and 19 m in non-sandy substrates (Supplementary Fig. 2). The average linear head retreat rates were 5.4 m yr$^{-1}$ and 3.2 m yr$^{-1}$ for UGs in sandy and non-sandy substrates, respectively.

To further assess the main factors contributing to the increases in UG exposure between 2010 and 2023 (Fig. 5b), we compared different scenarios. More specifically, for each hazard zone, we calculated the population in 2023 that lived in the hazard zones of UGs as mapped in 2010. Comparing this number with the originally exposed population of 2010 indicates how much of the increase in exposure is attributable to population increases in already existing hazard zones. Likewise, we calculated the 2023 population living in the 2023 hazard zones of UGs that were already present in 2010. Comparing this with the exposed population of 2010 indicates how much of the increase in exposure is due to further expansion of already existing UGs. Finally, by subtracting these two contributions from the total increase, we assessed the exposure increase due to new UGs that were formed since 2010.

### Uncertainty assessment

Both our estimates of the displaced (Fig. 4) and exposed (Fig. 5) populations are subject to uncertainties. The main source of these uncertainties is potential errors in the JRC GHS[8] population estimates that were used. As these errors are unknown, an exact calculation of the associated uncertainty is impossible. Yet, to evaluate the overall reliability of our results, we repeated the analyses of population exposure in 2020 (Methods, 'Estimating the exposed population') based on WorldPop Population data. For this, we used the unconstrained and unadjusted datasets[66,67]. WorldPop datasets have a similar spatial resolution as JRC GHS data and are informed by the same underlying census data (CIESIN GPWv4.11), but are based on a different modelling approach[8,66]. Moreover, WorldPop data are only available until 2020. Results show that, although comparable, estimates of the exposed population are lower when using WorldPop data (Supplementary Fig. 6 and Supplementary Table 6). This is particularly true for small cities and, to a lesser extent, for Kinshasa. For large cities (for example, Kikwit, Mbuji-Mayi, Bukavu), the deviations are typically less than 20%. Further analyses showed that this is mainly because WorldPop underestimates population density estimates in large parts of the affected areas (Supplementary Fig. 7). In many pixels in which WorldPop assumes densities of fewer than 50 persons ha$^{-1}$, JRC GHS indicates population densities of 100–1,000 persons ha$^{-1}$ (Supplementary Fig. 7). Further verification in Google Earth (considering the number of houses) confirmed that the latter are probably more accurate. Also, earlier studies indicated that unconstrained WorldPop data are often prone to underestimations in highly populated areas[68,69]. Hence, although the estimates of exposed population are higher when using JRC GHS data, they are probably also more correct. Given the highly similar procedure (Methods, 'Estimating the displaced population'), the same probably holds for the displaced population.

The main limitation of both datasets is that they are global data products based on the interpolation and extrapolation of often very limited population statistics, using proxies that probably correlate to population densities[8,65–67]. This is a necessity as there are generally no detailed and accurate population data available for the DRC. A notable exception to this is the city of Bukavu, for which detailed census data were collected in 2018, based on the methodology presented in ref. 70. Extrapolation of these data to the level of individual neighbourhoods resulted in a population density map that directly builds on field-based observations and provides the most reliable population data that can reasonably be expected. To further assess the uncertainty of our estimates, we quantified the exposed population using this population density layer (according to the gully extents in 2020; Methods, 'Estimating the exposed population') and compared the results with those obtained with JRC GHS and WorldPop data for 2018 (Supplementary Table 7). Although estimates based on the JRC GHS data are very similar to those based on WorldPop data, they are 22.9–39.3% lower than the results obtained from the direct census data. This indicates that, at least in the case of Bukavu, our exposure and displacement rates are probably still underestimations.

Based on these comparisons, and taking into account the effect of other sources of errors (for example, inaccuracies in mapped UG polygons), we cautiously assumed that all estimates of displaced and exposed population are subject to relative errors of up to 40%. Accordingly, we estimated uncertainty ranges by considering values up to 40% higher or lower than the reported figures.

### Data availability

The datasets generated and analysed during this study, including all mapped urban gully extents, are available at https://doi.org/10.48804/HTEZR0. The JRC GHS population data can be downloaded at https://human-settlement.emergency.copernicus.eu/download.php?ds=pop. The WorldPop population data used can be downloaded at https://hub.worldpop.org/geodata/listing?id=29. Nearly all satellite imagery used to map the extent and evolution of urban gullies is freely available in Google Earth Pro (https://earth.google.com/web/). Three of the satellite images used (Methods, 'Mapping the extent and expansion rates of UGs') were provided by CNES/Airbus and cannot be publicly shared. They can be purchased through this link: https://space-solutions.airbus.com/imagery/how-to-order-imagery-and-data/. The historical aerial photographs (Methods, 'Identifying cities significantly affected by UGs') are conserved in hard copy at the Royal Museum for Central Africa (RMCA), Belgium. Digital copies of these photographs can be obtained upon request by contacting the RMCA. Contact information for submitting these requests is available on the Geocatalogue platform of RMCA: https://geocatalogue.africamuseum.be/geonetwork/srv/eng/catalog.search#/home. Source data are provided with this paper.

### Code availability

The codes to calibrate and validate the UG susceptibility model (Fig. 3), as well as other Python scripts used to create figures, are available at https://doi.org/10.48804/HTEZR0.

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

**Acknowledgements** This research was mainly funded through the Belgian ARES-PRD project PREMITURG (Prevention and Mitigation of Urban Gullies), which supported the PhD research of G.I.M. and E.L.L. as well as fieldwork activities. C.M. was supported by the Development Cooperation programme of the Royal Museum for Central Africa, which is supported by the Directorate-General for Development Cooperation and Humanitarian Aid of Belgium through the project HARISSA (Natural Hazards, Risks and Society in Africa: Developing Knowledge and Capacities). E.D. was supported by a PhD scholarship from the Research Foundation Flanders (FWO).

**Author contributions** M.V., G.I.M. and O.D. conceived the idea of this research. G.I.M. and E.L.L. collected the data, with input from E.D., F.M.I., C.B., C.M., C.N., J.P., O.D. and M.V.; G.I.M. conducted the data analyses with the direct contribution of E.D and M.V., and with further inputs from E.L.L., C.M., C.B., J.P. and O.D.; G.I.M., O.D. and M.V. wrote the paper with inputs from all other authors. M.V. coordinated this collaborative study, with the administrative support of A.H., F.M.I. and C.N.

**Competing interests** The authors declare no competing interests.

**Additional information**
**Correspondence and requests for materials** should be addressed to Matthias Vanmaercke.
