## [Peer Review File · Nature]

Mapping urban gullies in the Democratic Republic of the Congo

Corresponding Author: Professor Matthias Vanmaercke

Version 0:

Reviewer comments:

Referee #1

(Remarks to the Author)

Review of manuscript #2022-09-14721 "The growing risks of urban gullies in the Democratic Republic of Congo" submitted by Guy Ilombe Mawe and colleagues

Dear Authors,

In this contribution you report on the distribution of, and risk from, gullies in Central African cities. Gullies are erosional landforms that have been well studied mostly in much less densely populated, especially semi-arid or rangeland, areas. The scientific study of gullies in urban areas seems to be quite a novel field and clearly deserves attention. To this end, you make the case here that more and more residents in cities of the D.R. Congo have been impacted by growing urban gullies in past decades. The methodological approach of spatially intersecting gully outlines mapped from satellite images with gridded estimates of population density is simple and intuitive. Yet this approach may fall short of capturing the complexity of the underlying causes, dynamics, and consequences of urban gullies. A more rigorous enquiry might include analyses and discussion of substrate and drainage conditions, land use, erosion rates, and other key controls on urban gullies. For example, when and where do urban gullies form in cities due to which critical conditions? Do we expect a correlation between population density and urban gully formation?

While you point out some of these issues and problems, your study hardly prods deeper. Instead, you estimate the exposure of urban residents in terms of their spatial adjacency to gullies. Without any further details such estimates might gloss over a lot of important detail that may be necessary to understand, predict, and remediate the impacts of gullies and how people respond to them in cities elsewhere. Demographics and socioeconomic dynamics might be more essential than simple population densities, if you are concerned with the exposure and risk here. While I like the way you point to this possibly under-recognised hazard, I found it difficult to appreciate objectively without the context of other urban geomorphic processes or hazards such as floods or landslides. In essence, I believe this contribution is timely and perhaps of broader interest, but many of your inferences and conclusions rest on too simplistic spatial intersections, while unduly generalising local observations without broader supporting data or rigorous tests.

General Comments:

- The case for studying large urban gullies could be made a bit more stronger. Define what you mean by "large" and highlight better previous work. The introductory notes on large urban gullies offer a few mechanistic and geometric insights that readers may find difficult to extrapolate for the diversity of urban settings in the Global South. The phenomenon of gullies in cities cannot be that novel, judging from the previous studies you duly cite.

- The mechanistic background of gully formation and growth remains a bit shrouded and unduly brushed over in a general manner. This makes it hard to assess which processes are likely responsible for generating hazards and risks to the local people. Your mention of some of the impacts could be much better linked to the life cycles of urban gullies. I suspect that the introduction could work well with a more systematic and detailed review about what we know already and what we might learn from mapping urban gullies.

- The lack of systematic data on large urban gullies surely has several reasons. It would be good for readers to learn more about those reasons and also about whether perhaps other natural hazards in the urban Global South attracted more attention in terms of systematic mapping. This context of hazard and risk awareness might be important.

- The choice of the D.R. Congo as a study area remains somewhat unmotivated and may need some more justification. Where have most studies on urban gullies worked before? Do you wish to close a white spot on the map or expand and consolidate (or even challenge) previous work?

- Mapping of gullies. Whether or not most gullies are linked to urban sprawl depends on the sampling method. By design you mapped gullies in cities, so this dependence may be biased. It might be useful to cross-check whether gullies formed at similar or lower rates outside of the city areas, assuming all other controls constant or comparable at least. The reported dimensions of the gullies seem spurious and it may be helpful to replace them by measures of spread in the data. Without more details about the cities studied I found it hard to assess whether the reported extent and rates of historic gully erosion were low, high, or moderate. Again, a comparison to other rates of land-use change or turnover by other natural processes would help a lot. It might also help to list diagnostic and objective criteria for detecting gullies from satellite images. For example, how do you distinguish gullies from quarries or excavations with an elongate shape?

- The population density estimates have 100-m resolution and are thus much coarse than the dimensions of the mapped gullies. Some error appraisal seems like a logical thing to add to this analysis. The penultimate paragraph in the main text alludes to errors, but describes them in a very superficial way that does the data hardly any justice.

- Exposure vs. risk. The title reports "growing risks of urban gullies", though you hardly offer any risk estimates in the technical sense. The gully mapping reports frequencies and magnitudes at best, which should be converted into hazard levels. Much of the results refer to exposure instead by estimating how many people had been displaced (or injured if not killed) by expanding gullies. Using a linear trend on the cumulative data in Fig. 4 perhaps inadvertently also emphasises that the ratio of people affected over gully area has remained largely unchanged. This trend might also be biased by excluding gullies that have remained stable in shape (as described in the Methods). Defining risk on the basis of proximity to a gully (100 m in this case) seems a bit arbitrary and warrants more explanatory detail and robust testing. To this end, you undersell your data, as you should have headward and lateral expansion rates of gully areas available from your mapping.

- The risk estimates you offer here are mostly exposure estimates in terms of people living close by to urban gullies. Readers might find it very difficult to assess whether, for example, a projected exposure of "33 persons per 100,000 inhabitants per year" (line 229) is high, moderate, or low without any further context. How do other hazards like floods, landslides, windstorms, traffic, or political conflict compare to this estimate? How reliable is this estimate in terms of confidence bounds?

- Many of the inferences and conclusions in the last third of the text read as if they could apply equally well to many other natural hazards. Readers might expect a more targeted discussion and more specific conclusions that rest on your data.

- The language and grammar are generally accessible and correct, though the text is quite wordy, and also vague, in places. I flagged several examples in the annotated manuscript file (see below).

Specific Comments: Please consult the annotated manuscript PDF for more detailed remarks and suggestions.

Referee #2

(Remarks to the Author)

This paper highlights the role of erosional gullies in densely populated areas in the DRC and their substantial impact on those populations. It also suggests this problem is ubiquitous across the Global South, likely displacing 10's of thousands of people per year.

I am pretty familiar with the gully literature and the badland literature, which this paper definitely fits into. So from that perspective, I don't think there is new scientific ground broken. That said, the framing of this issue as an urban hazard, and the demonstration of its pretty staggering impact, I think is original and significant, and for that reason I am supportive of publication.

The methodology make sense to me, and the approach seems reasonable, as do the conclusions.

My main suggestion for improvement of the manuscript is this:

Line 146-156 - I went and looked at a bunch of the gullies in GE, particularly in Kikwit. They are indeed really impressive and clearly quite destructive. One thing that comes out in the imagery however is that there is a competition between gully expansion and revegetation that is apparent. The gullies don't simply unzipper in one direction, but rather seem to be periodically arrested or healed by vegetation regrowth.

This is consistent with observations from other badlands, for example see review from Gallart et al (2013) - Catena, that speak to the important role of vegetation regrowth in stabilizing gullies. Given the important management implications for this, it seems like a missed opportunity (and possibly an oversimplification of the problem) to not delve into the issue of

vegetation and the punctuated growth of these networks.

For this reason, I think it would be great to add a section on the role of de-vegetation and re-vegetation on the erosional dynamics of these gullies.

I also noted one typo:

121 - change "is" to "are"

Referee #3

(Remarks to the Author)

Thank you for the opportunity to review this paper. Key results include the mapping of urban gullies in the D.R. Congo over space and time and referencing urban gully expansion in the context of populations at risk for multiple time points. While I found the content and area of study interesting, I am not sure Nature is the most appropriate publication outlet, at least in the paper's current form. I respond in more detail below.

Comment on the relative importance of urban gullies compared to other geohazards?

To be honest, I am not that familiar with urban gullies but that doesn't mean they are not important. I found the paper very interesting considering this type of hazard is understudied combined with the complexity of addressing sustainable solutions to the issue. I think it would be insightful to expand upon how urban gullies persist in contexts beyond the D.R. Congo – currently this is only noted by citations in Lines 46-47 but the implications of it being a hazard more specific to the Anthropocene could be unpacked more, relevance to both Global South and North or why Global South so much more vulnerable. Authors also reference other cases in Lines 222-224, but again, I think drawing some commonalities or identifying some points of reference that provide the larger "so what" context for this particular hazard is important.

Does the manuscript have flaws which should prohibit its publication? If so, please provide details.

The authors have done a lot of spatial mapping work using Google Earth (mostly) to identify the extent, rate, and existence of urban gullies in the D.R. Congo. I think an interesting aspect lacking from the paper is a compelling argument, in any form, regarding the underlying drivers for the changes. I don't understand how the authors conclude that urbanization (how is urbanization defined?) is the reason for gully expansion (Lines 88-89). Authors note in lines 77-80 the importance of rainfall and changing frequency/intensity of rainfall due to climate change. How do authors address that component in relation to urban expansion and population density increases over time? I also cannot find information relative to environmental characteristics, and notably rainfall information, other than topographic and substrate characteristics portrayed in Figure 3. The visually observed built area ("urban") expansion using aerial photographs from the 1950s - if gullies were absent before the area became urbanized and then appeared – it's purely a function of development? What about other confounding factors? I feel like the analysis could be pushed further to discuss the environmental context in which cities are growing – the relationship of the substrate and topography relative to population increase and settlement/built area expansion – authors state that most PAR are located in Mbuji-Mayi and Kinshasa (Line 197) – what is different about those areas from others? Authors note that there are 166 gullies where no expansion occurred and thus probably are stable. I feel like further investigation relative to unstable gullies would add insight, notably are there any differences on the location of these non-expanding gullies relative to population density and associated proximity, road infrastructure and built settlement development, etc.

I think this is a publication worthy effort but I am not sure for Nature, at least as it stands. An additional strength to this paper would be analysis that better supports the question, What are the processes for creating exposure?

Also, I do wonder if others have done any similar work but with alternative remote sensing sources such as radar, for mapping our urban gullies and expansion rates.

If the conclusions are not original, it would be very helpful if you could provide relevant references.

I am not aware of other work that duplicates these current paper's conclusions.

On a more subjective note, do you feel that the results presented are of immediate interest to many people in your own discipline, or to people from several disciplines? If you recommend publication, please outline, in a paragraph or so, what you consider to be the outstanding features.

I think the topic area could be of interest to those in geographic disciplines. Aspects of this publication that are interesting include 1) drawing attention to a less studied/publicized but impactful natural hazard and 2) proof of concept approach that could be applied in other places for mapping out PAR in relation to urban gully extent and expansion.

When reviewing the paper, we would be grateful if you could pay particular attention to the statistics, if applicable.

The authors state they use WorldPop data and reference "a series of gridded datasets that provide estimates of population density at a spatial resolution of 100 m and on an annual basis (2000-2020)." That is a data product that was parameterized for consistent and comparable mapping across countries, globally. As such, it will not be the most ideal, country-specific option for a contemporary analysis of population distribution when selecting a gridded population map. However, for comparing population counts over time, within the time frame given, this is an appropriate data product and probably the

most reasonable choice

There are a variety of WorldPop products (did they use the UN adj or unadj version?). I would encourage specific language in M3 to note 1) that the global product is a specific WorldPop product, appropriate for use in this analysis but not the only option available from WP and 2) slightly more information on the disaggregation process to derive the gridded products (Stevens et al., 2015 good ref to complement ref #7) and 3) considerations for underlying uncertainty, esp for a country where there does not exist consistent, reliable census counts (Gridded population estimate datasets and tools. - WorldPop). It's important to acknowledge that the data underlying PAR estimates are a function of a statistical model informed by other data – settlement and other land cover, NTL, climate variables, etc that will be influenced by changes in those gullies and subsequently induce error in the population estimates in an interactive way.

One potential suggestion for the authors would be a comparison for a single year placed in the supplemental that shows estimates from different products. Authors could compare their most recent mapping gridded population product from the global data to a more contemporary mapping of D.R. Congo population from the GRID3 project and/or an individually constrained or unconstrained choice from the WorldPop site. There is also JRC GHS-POP data on gridded population that could be compared (and had a temporal component although at a 5-year time step). I only suggest this type of comparison for authors understanding of how varied the gridded estimates might be relative to data product used. But I recognize they are trying to estimate PAR over time and in doing so, their data product of choice is the most reasonable one for their purposes. However, I would include the data source as a real and legitimate source of uncertainty regarding population numbers and PARs stated in the paper (lines 291-292). And perhaps a short explanation for the “evidently” aspect of line 173.

As an additional comment, Figure 3 is a very nice figure. I would suggest making Figure 4 with larger font and legend sizes. Also, lines 313-314 – geomorphic criteria for urban gully needs to be more detailed, not just cite a paper. And then for clarity, in the supplemental - Are the 2010, 2015, and 2020 estimates of # of people in risk zones based on the D.R.I. or the gridded product from that year? That's unclear and a little confusing as the estimated WP total is relative to the D.R.I from Table S1 (if I read that correctly).

Version 1:

Reviewer comments:

Referee #1

(Remarks to the Author)

Review of “The growing problem of urban gullies in the Democratic Republic of the Congo” submitted by G.I. Mawe and colleagues

Dear Authors,

This is my second review of this study, hence I refrain from summarising this any further. Many thanks for providing such a thorough reply letter to the original comments and suggestions by all three referees. I appreciate your efforts of collecting and presenting additional new data, and also redrafting large sections of the original manuscript. Overall, I think you have put together an interesting study. I still doubt, however, that it is novel, ground-breaking, or provocative enough for the journal. Nowhere do your results reveal any new insight to mechanisms and drivers of these gullies, or even their consequences (people killed or displaced, houses destroyed, infrastructure damaged, etc.). Instead, much of your conclusions still rest on the simple, and likely biased, spatial intersection of gully and urban population data. Yet, even the most basic statistical inference you use highlights some substantial uncertainties and inconsistencies that your discussion fails to pick up.

General comments

- The introduction makes some case for urban gullies as an understudied hazard, but could do a much better job by summarising more objectively the current knowledge about causes, triggers, thresholds, growth rates, and impacts (even if based on case studies). Many of the qualitative and unduly overarching descriptions could acknowledge at least the knowledge base from a large literature pool on gullies. My initial reservation about the lack of a more mechanistic summary has not really been addressed in the revisions that well.
- In my previous review, I had also suggested some context about urban gullies compared to the many other sources of potential disasters: perhaps widen the lens a bit to avoid too much bias. While I appreciate your outlook on this (lines 216-227), I think it is misleading to compare the percentage of people exposed to urban gullies in the D.R. Congo to the percentage of all African people exposed to other natural hazards, as the density of both population and hazard sources differs widely; especially short-term rate estimates can be highly volatile (inset to Fig. 4b).
- Your attribution of gully formation to urbanization still solely hinges on spatial proximity. Simply because a gully is close to a road or buildings is no confirmation that these structures have caused the gully. Urban areas have dense road network by design, such that any location (gullied or not) is likely to be close to a road. To be more objective, you may need to demonstrate that the gullies could not have formed without urbanization. This sort of counterfactual approach has become a key ingredient in attribution studies. At the least you may want to correct for effects of road density. Your definition of urban

gullies (“the gully needed to be located within 200 m of buildings”, line 376) might introduce a bias to areas of high population density already. Would it not be fair to map all gullies instead to get a more objective view of the potential role of urbanization.

- Your results still fall short of supporting your conclusions. Fig. 3a shows how the percentage of built-up areas is the most uninformative for distinguishing areas with urban gullies from those without. A more formal and flexible way would be to run a classifier to see how well built-up areas can predict locations with mapped urban gullies. Fig. 4b shows that new urban gullies form most rarely and preferably in areas of low population density; this observation indicates that maybe urbanization alone is not the sole explanation of gully formation. Should the title of your study not more likely read: “The problem of growing urban gullies...” instead of “The growing problem of urban gullies...”.

- Some of the statistical inference needs attention. The distributions of observed gully expansion, population density, and estimated displacement are all positively skewed (Fig. 4a), with the median values up to an order of magnitude below the means (diamond symbols). Hence, the means and derived rates (Fig. 4b) poorly reflect, and in this likely case overestimate, the central tendency. The doubling of mean annual displacement rates (Fig. 4b) in the 2020-2023 interval is curious and needs some more explanation. Moreover, how likely is it that the confidence intervals in Fig. 5 admit negligible changes to the population exposed? Why do you omit these intervals in Fig. 5b? For the potential expansion zone, the upper half of your confidence interval alone is >1.2 million people exposed, and thus close the increase for this zone in Fig. 5b. While this level of uncertainty seems plausible, it should advise against reporting your estimates in an overly spurious way. You may also want to explain why you estimate here that many more people have been exposed to newly formed gullies instead of gully expansion: in Fig. 4b you show and argue the opposite, i.e. that many more people were exposed per unit area of gully expansion. Does this not seem contradictory?

- The revised discussion is the weakest section by far, mainly because it is still unduly long, verbose, and unnecessarily repetitive in stating the significance of urban gullies, and scaling this up to the entire “(sub-)tropical Global South” (line 249) without much further support by data. Here you missed again the opportunity to compare your study to the findings of similar, previous work on urban gullies. You also mention the potential influence of climate change through altered rainfall characteristics; this mention is odd, given that you discarded rainfall early on in your study (lines 124-125). Your claim that your study “contributes to a better understanding of the lifecycle of UGs” (line 286) is unsupported by the data. You show snapshots of urban gully dynamics but offer little clues as to their lifetime. Much of the remaining discussion revolves in lengthy tones about the basic mechanisms of gully formation and expansion that could have equally well fit into the introduction. Fig. 6 is nice, but not really novel; it might miss important details about rainfall, runoff, slope substrate, and groundwater dynamics.

- The thorough revision and rewrite has improved the language and grammar, although many wordy and superfluous phrases have stuck. A number of statements also remain ambiguous, while others feel a bit stagy. Please see my detailed suggestions below: these are not exhaustive, however.

Line-specific suggestions

18: “Large urban gullies wreak havoc in numerous tropical cities” – Perhaps try toning down overly dramatic statements and instead explain more objectively the adverse impacts of gullies.

19: “Such gullies result from inappropriate urban planning” – This reads as if these gullies cannot form naturally?

20: “sensitive to soil erosion as well as insufficient infrastructure” – Ambiguous phrase. Do you mean that “gullies are sensitive to ... insufficient infrastructure”?

22: “they can drastically expand” – See first comment; consider quoting rates of gully expansion instead.

24: Why are gullies a “new type of geo-hydrological hazard” if so much literature exists about them already?

31: “drastically” – See comment on line 22.

32: The comparison between pre-2020 and post-2020 rates might be lopsided; I assume you only have four data points for annual rates since 2020.

33: “potential expansion zone of urban gullies” – This interesting concept might need a brief definition here.

34: “This number doubled since 2010” is in conflict with “Over 3 million people currently live” in line 32.

35: How would “climate change” influence the number of people exposed?

40: Overuse of “drastically”; please provide some objective rates instead.

44: “other problems and risks” – Such as? How would urban gullies compare to these other problems and risks?

48: “Democratic Republic of the Congo (DRC) is particularly affected” – This reads like some availability bias. Or are there any physical reasons why this particular nation is so prone to urban gullies?

52: "typical erosion features of the Anthropocene" - Typical of what specifically?

54: "exceeds the local resistance of the topsoil against incision" – What about groundwater flows and erosion? Or do urban gullies form exclusively by surface erosion?

56: "urbanization of these environments" – Please be more specific. Do you mean landscapes prone to gullies or landscapes that already have gullies?

57: "increases in roofs" → "increases in roof-top area"?

61: "leading to local increases in contributing area" – Really? How does this work? Is the contributing catchment area not set by drainage divides?

68: Somehow I have the feeling that the previous summary does insufficient justice to the "heterogeneous, complex and threshold-dependent conditions under which UGs form". Which thresholds do you mean?

78: "many gullies continue" – Including urban ones?

81: "erratic nature of expansion events" – This reads as we have no clue as the drivers of gully expansion.

81: "enormous size" – How do you define "enormous"?

84: "UGs often form in poor, informal (peri-)urban areas" – Are there exceptions? Urban gullies must form in urban areas by definition. It would be useful to have a comparison with gullies forming in rural areas to appraise any differences or similarities in size, formation rate, etc.

91: "understanding of the magnitude and drivers of the risks associated with UGs is a critical step towards disaster risk reduction" – This really depends on whether urban gullies pose a substantially higher risk than other disaster sources.

101: "Verification" → "validation"? Checking air photos does not warrant that you eliminated all sources of error.

103: "overwhelming majority of these gullies are indeed linked to urban sprawl and road construction" – Based on which criteria? How can the remote sensing data tell you this?

104: "fact" → "observation"; "deforestation was already well advanced" – This is the first time you mention deforestation. I would have expected to see this mentioned earlier in your review of the current knowledge on urban gullies. The second part of the sentence is difficult to appreciate without any time stamp on forest losses.

106: "clearly connected to roads and/or had a significant amount of buildings in their direct vicinity" – How does this confirm that roads or buildings were responsible for gully formation?

109: "typically very large" – Style; how large?

110: Add country to "Kinshasa" for readers unfamiliar with the geography of Africa.

112: "UGs typically occur in areas that are somewhat steeper and sandier" – Avoid "typically" unless you state exceptions as well. What do you mean by "somewhat"? Is this difference in steepness and sand content relevant or not?

116: "Other key factors explaining the contrast between areas with and without UGs are the fraction of built-up area and the road density" – How many possible factors did you look into?

119: "98% of all mapped UGs were clearly connected to the local road network" – Does this high percentage not simply reflect a dense road network in urban areas? One could similarly argue that most houses are connected to the local road network.

121: "well explain" – How well? Please provide some objective measure.

124: "rainfall characteristics did not further improve our model" – Why not, and which model do you mean?

125: "overall rather limited range in rainfall conditions across DRC" sounds unduly generic and hardly informative enough to discard rainfall.

127/Fig. 2: "showing all cities investigated" – How did you select these cities? Inset bar diagram might be difficult to read; why not simply show % of urban area affected by gullies?

135/Fig. 3: "showing the distribution in average slope steepness, sand content, built-up area and road density" – Why not show the raw distributions instead of smoothing these properties? You may want to normalize these distributions by those of the urban/non-urban areas to obtain a relative distribution or frequency ratio. Curiously, built-up area (%) shows the least

conclusive distinction between areas with urban gullies and those without (Fig. 3a).

139: “significant difference in distribution according to a two-sided Mann-Whitney U test” – This test concerns the averages of the samples only, right?

145: “experienced clear (> 10 m²) spatial expansion” – I assume that “clear” refers to “detectable” here? Consider using relative expansion rates instead, as it may be more difficult to detect small expansions for large gullies?

146: “UGs showed important expansion over this period, accounting for 70% of the total observed expansion” – Please clarify: 70% of the absolute increase in total urban gully area happened in these five cities? If so, please add how much of the total gully area these cities had originally in 2004.

150: “we estimate that ca. 142,400 people were directly displaced” – Do you have any independent data to validate your estimates?

152: Is the “average of ca. 7,120 persons.year⁻¹” sufficiently representative, i.e. do the five most affected cities have comparable population densities?

152: “this process clearly accelerated” – How do you know it is just one process: the growing displacement average rate might have several underlying processes.

155: “UG expansion events” – Unclear why these are events. Can this not happen also gradually?

155: “relatively limited” → “lower”.

160: “areal increases” might be better expressed in relative terms instead of absolute areas?

163: “hundreds of houses were indeed destroyed” with each event or altogether?

166: “formation of new gullies is observed relatively rarely ... and typically happens in areas with a lower population density” – This is an interesting point, and might be one key to understanding better the preparatory factors for urban gullies. This observation however also casts doubt on whether urbanization alone is responsible for forming these gullies.

168: “it is responsible” → “it was ...” or “it has been ...”. Delete “As such” here and elsewhere.

171: “important” → “prevalent”?

172: “In addition, this typically occurs” – Wordy style.

174: “importance seems to increase over recent years” – Judging from which metric?

178/Fig. 4: The boxplots seem to be truncated at lower end: please show the full distributions of the data; the numerous outliers deserve some mention. Also please explain the symbols: what do the black diamonds show? Do you assume that population density remained constant “between 2004 and 2023”? The estimated displacement is then summed accordingly over this interval in the right-hand boxplot? Why would gully-head expansion affect areas with higher population density on average than gully side-wall expansion (A vs. B in middle boxplot)? Can you think of any physical or socio-economic explanation? Fig. 4b shows that the mean rates of persons displaced by unit gully area are higher than the medians of population densities in Fig. 4a. The distributions are thus positively skewed, and the mean rates might be a poor measure of central tendency.

186: Please specify the width of the “confidence interval” here and elsewhere.

191: What are “indirect consequences” of urban gullies. Please give some examples. How many people in cities live less than 100 m away from river channels and face indirect consequences from floods?

192: “an estimated 3.24 million people lived within the potential expansion zone of UGs” – This number is nearly 1.2 times more people living within a 100 m of these gullies. Do you really expect that all gullies would expand by more than 100 m?

195: It might be useful to define here the difference between “potential” and “expected” zones. The latter has a statistical connotation.

196: “These risks remain difficult to quantify exactly” – And still you offer exact numbers of people affected. Why not add some error bars here?

196: “comparing the exposed population in 2020 (Fig. 5a) with the average annual displacement rate in the period 2020-2023 (13,668 persons/year; Fig. 4b) indicates that 4.8% of the population” – If I compute $13,668 \text{ persons/year} \times 10 \text{ years} / 3.24 \text{ million persons}$ (line 192), I obtain 4.2%. For the 2.74 million people living within 100 m of urban gullies (line 189), my result is 5.0% instead of the “5.9%” you state in line 200. The same goes for the 557,000 people in expected expansion zones: I obtain 25% instead of the “29%” reported in line 201. I don’t mean to split hairs here, but I would like to understand how you arrived at these estimates. I tried different values, so it cannot be a sole effect of rounding.

203: “This is true for” → “We observe this for”.

204: "53-55% of this increase is attributable to population increases within hazard zones already present in 2010" – Unclear. Please explain.

208/Fig. 5: Panel a suggests that the potential expansion zone of urban gullies is >100 m throughout? Please comment on this. You may also want to define briefly the "expected" expansion zone here. What is this expectation based on? Consider adding the number of people that were displaced by gullies over this time. Do I read the confidence intervals correctly that you cannot rule out that the exposed population has remained constant from 2010 to 2023?

215: "Urban gullies as a new geo-hydrological hazard of the Anthropocene" reads catchy, but "new" hinges on the starting date of the Anthropocene. You may also want to demonstrate that these gullies did not exist previously.

216: "massive scale of the problem of urban gullies" – What is the scale indeed? You have mostly reported absolute numbers, while a scale demands a ratio.

218: "clearly more than local" – Local can refer to cities, if you can exclude that gullies also form outside of cities.

223: "estimated 1.1% of the total African population" – I doubt that you cannot compare the percentage of Congolese population to the African average, for reasons of spatial differences in both population density and distribution of hazard sources. It is nice to see some context of hazards, but this direct comparison is a bit weak.

224: "major earthquake" – Needs a definition.

226: "volcano" – active or extinct?

227: "likely much lower" can mean anything.

228-235: "56 persons per 100,000 urban inhabitants..." I am not sure whether comparing loss of houses to loss of lives is helpful here. You also rely on short-term rates that might be volatile, especially so for landslides.

236: "exposed population doubled between 2010 and 2023" with what (un)certainity?

238: "African urban population is expected to nearly triple by 2050" – Why not use the estimates for your study area instead?

242: "rainfall intensities in tropical Africa may increase by 10-15%" - And yet you stated earlier that rainfall was an uninformative predictor of gully activity (lines 124-125).

246: "increases could easily double gully expansion rates if other factors remain the same" – See previous comment.

248: "This is not only true for the DRC" – Strong statement. I think you refer to your inference here.

249: "The growing number of studies reporting on UGs in other countries seems to confirm this" – Not sure what this uncommented list of references confirms. Here, your discussion should pick up this previous work and build some context for your findings.

253-262: Consider shortening this paragraph; not all of it is relevant to your study.

258: "As such, hillslopes and plateaus that are not prone to landslides or flooding, can be highly susceptible to UGs." - How can you tell?

259: Delete "certainly".

261: "This is especially so since, by nature, UGs tend to occur in densely populated areas, posing a direct threat for the surrounding population." - Verbose. The first half of the sentence partly undermines your analysis of spatially intersecting gullies with urban areas: if you focus on urban gullies, they must occur in areas of high population density by definition.

263: "persisting problem" – It would be useful to learn about the lifetime of these gullies. Do you have any evidence about gullies that stabilised eventually?

266: "but mainly due to population increase" – Difficult to tell given the errors attached to Fig. 5a.

269: "many UGs develop in overall poor and unplanned urban neighborhoods" – In your study area or in general?

279: "certainly" → "may"?

281-283: Redundant.

286: "our work contributes to a better understanding of the lifecycle of UGs" – Where do the data show this?

286-297 and 298-310: These lengthy paragraphs also contain mostly repetitions; mind your overuse of "typically" and

“significant danger”.

332: “Our observations indicate that older UGs with vegetation tend to be more stable.” - Refer to figure that shows this.

342: “Hence, as always, prevention will be better than curing”. - Verbose.

376: “the gully needed to be located within 200 m of buildings” – This filtering might bias your data to high population areas?

Referee #2

(Remarks to the Author)

This is the second time I've reviewed this paper. I was enthusiastic previously, and remain so. I think this is an important, creative, and well executed study. The few comments (regarding vegetation) that I had in my first review were addressed in the revision. I recommend that this be published. I did have two very minor comments:

Line 93 - change “addressing” to “address”

Line 171-175 - is the widening just a geometric consequence of the gully deepening? Here I am imagining a triangular cross-section that just grows self-similarly. Or does the widening happen by some other process (such as slumping)? Anything you can add on this point is potentially valuable context.

Congratulations on a fantastic study. This was a pleasure to read and (more significantly) opened my eyes to a geomorphic process that I am familiar with but which is happening at a rate and scale and with a human impact that it is frankly astonishing.

Referee #3

(Remarks to the Author)

I appreciate the amount of effort and thought the authors put into revisions for this manuscript. I think the overall attention to suggested edits makes this a much stronger and engaging publication. One comment - in terms of M4 and M6, authors could also reference a known understanding with the unconstrained WorldPop data regarding underestimation in highly populated areas for rationale in data choice – see Deville et al., 2014, PNAS or Stevens et al., 2020, IJDE. With M6, I'd adjust language to note that JRC GHS and unconstrained WorldPop data are informed by the same underlying census data – CIESIN GPWv4.11 – however, the modeling approaches are different. The additional effort to assess uncertainty using the Bukavu data as well as the acknowledgement that the gridded datasets must be interpreted with caution is commendable.

Version 2:

Reviewer comments:

Referee #1

(Remarks to the Author)

Review of manuscript #2022-09-14721B “The growing problem of urban gullies in Congolese cities” submitted by Guy Ilombe Mawe and colleagues

Dear Authors,

This is my second review of your study and therefore I shall skip yet another summary. However, I do owe you thanks to what must be the longest rebuttal letter I have had the opportunity to read through. Overall, I very much appreciate your detailed and clarifying replies to most of my remaining concerns. In hindsight, perhaps some of your explanations would have been more suitable for the main text in earlier versions of the manuscript (if not the supplementary file). However, seeing that your study seems to be in editorial favour, I will refrain from reiterating my reservations. That said, I believe that your study is innovative and sufficiently provocative. Below I offer a few suggestions (keyed to line numbers) that you may wish to consider for the final revision:

MANUSCRIPT FILE

21: “new geo-hydrological hazard” – Your study features features surprisingly little about the role of water other than what you cite from the literature. Also, the term “new” seems to ignore that papers about this topic (and area) have been around for some time (e.g. refs. 1-3, 21-23, 26, 28). What are “mega-gullies” in this context (refs. 1, 2)?

88: “Nonetheless, steep and sandy soils are not a strict prerequisite.” - Still, Fig. S2 distinguishes on the basis of “whether the city is located on sandy or non-sandy soil substrates”, and in l. 647 you make the case for a significant difference, at least concerning gully width. On this occasion it would be helpful to reflect on the quality of the soil/sand data: the 30-m resolution

indicated in Table S2 is quite optimistic.

91: "98% of all mapped UGs were clearly connected to the local road network" – Still, the effect of road density seems minimal in your susceptibility model; please see comments below.

93: "Combined, slope steepness, soil type, the fraction of built-up area and road density well explain the presence or absence of UGs (Fig 3b, S3, S4)" – "Soil type" really means sandy fraction in the topsoil? It is curious to see that built-up area has the second lowest weight in your susceptibility model (Fig. S3, Table 2); the effect of road density is even less pronounced (see below).

96: "rainfall characteristics did not further improve our model" – Where do you show this?

97: "overall rather limited range in rainfall conditions across DRC" – This reads unduly simplistic, at least if considering that tropical rainfall meteorology is highly seasonal and influenced by the shifting of the ITCZ, jet streams, easterly waves, ENSO, etc.

140: "exactly" as in "accurately"? Yet, in l. 142 you estimate this to the nearest tenth of a percent ("4.4%").

162: "around 12,200 persons per year lose their house due to UG expansion" – This would be a good location to add some validation. How many reports do you have of people having lost their homes because of gullies. If unavailable, repeat for your readers here how many buildings or built-up area you can confirm to have been lost because of recent gully growth, judging from your remote sensing studies.

173: "our analyses show that the occurrence of UGs is closely linked to built-up area and road density (Fig. 3, S1, S3)" – Yet, your susceptibility model has the lowest (absolute) regression weight by far for road density (Table S3 states " $6.183e-5$ ": please use proper exponential notation, and make sure you use correct decimal symbols). You mention the importance of roads several times in your study, but your susceptibility model fails to support this. If anything, road density has the weakest effect in this model. Any thoughts why this is so? The relative weight of sandy soils present seems to be much more prominent if readers were to take these coefficients at face value.

201: "typically" – Avoid unduly generalising qualifiers: you state in your study that urban gullies are a "new" hazard.

210: "outskirt" → "outskirts".

216: "Once formed, most UGs continue to expand." - Reiterates in parts the train of thought starting in l. 190: "UGs are -once formed- a persisting problem".

226: "As the gully head further migrates upslope, its contributing area –and by extent the amount of runoff that can accumulate at the head– will decrease, eventually halting the process." - What about subsurface hydrology and stream piracy? Or are these irrelevant to urban gullies, where people interfere with the natural drainage frequently.

240: "vegetation may play a crucial role here" – Yes, I noticed "tree cover" in Table S2, but it seems that this predictor never made it into the model.

465: "UG susceptibility model" – Please explain colour scale in caption.

480: "indicate the estimated confidence interval of $\pm 40\%$ (Methods M6)" – From your descriptions in Methods M6 I gather that these cannot be proper confidence intervals. Reword to "estimated overall error" or something similar.

567: "conducted these analyses at a resolution of ca. 1 km² (0.008333°)" - How can you convert unit area to unit angle? Even if using fixed geographic coordinates, the spatial resolution parallel to the equator will change with latitude.

581: "differed significantly between cells with or without gullies" – So if any of the 1-km² grid cells contains at least one gully, it counts as a "gully pixel"? Fig. 2.4 shows that you can have many gullies in a given pixel. How do you account for this inflation effect, assuming that you cannot increase the spatial resolution of your model.

582: "logistic regression model⁶³ that combines the variables best explaining observed differences between affected and non-affected cells" – How did you identify those "best explaining" variables? Table S2 lists 12 candidate predictors but nowhere did I find explanations about this selection. How did you counter excessive model tuning in your predictor search? Please also indicate whether you standardised the predictor values, which is really a must if you wish to compare coefficients in multivariate logistic regression.

583: "making sure each variable remained significant within the model" – How did you make sure. Did you use stepwise regression, etc.?

629: "While somewhat arbitrary, this distance provides an intuitively understandable estimate of the population for which the threats of UG expansion forms a regular and significant concern." - It might be useful to highlight this assumption in the main text also (e.g. insert "arbitrary" in l. 134).

647: “populations” should read “samples”.

647: “considered the 95% quantile as the expected maximum width a gully may attain in such substrate” – It might turn out to be poor practice to set the “expected maximum” below the actually observed maximum, especially in hazard or exposure estimates. Fig. S2-3 indicates that gully widths might differ between cities. Would it make sense to adapt this “maximum” to the city level?

648: “70.3 m” – Please avoid spurious values here and elsewhere, unless you can determine gully dimensions to the nearest decimetre.

701: “Further verification in Google Earth (considering the number of houses) confirmed” – Assuming that the number of houses scales how with population density? Fig. S6 suggests that the systematic mismatch between the chosen two population datasets is at least of the order of 10,000 people for the year indicated. This is of the same order of magnitude that some of your annual exposure rates are of.

723: “all estimates of displaced and exposed population are subject to 40% uncertainty” – Again, this is something to pronounce in the main text. Please also mention whether you believe this to be an absolute relative error or some sort of standard deviation, etc.

SUPPLEMENTARY FILE

1: Please add article title, authors, etc. The figure numbering is a bit awkward with entries such as “Fig. S2-4” (l. 30) versus “Fig. S2.4” (l. 46). Please simplify if possible.

16: Typo on y-axis of Fig. S2-1.

46: The mapped gully heads in Fig. S2.4 hardly stand out before the chosen colour scale. Please enhance contrast.

Fig. S3: How can the logistic regression weights remain constant for a Monte Carlo simulation? Should these not also have errors? Explain what Y means. Note that predictions based on this model have higher uncertainties than the ones you give for the fit in Table S3. Stating the regression coefficients as is makes sense only if you have dimensionless predictors; however, Table S2 indicates units for each predictor.

Referees' comments:

Referee #1 (Remarks to the Author):

Review of manuscript #2022-09-14721 "The growing risks of urban gullies in the Democratic Republic of Congo" submitted by Guy Ilombe Mawe and colleagues

Dear Authors,

In this contribution you report on the distribution of, and risk from, gullies in Central African cities. Gullies are erosional landforms that have been well studied mostly in much less densely populated, especially semi-arid or rangeland, areas. The scientific study of gullies in urban areas seems to be quite a novel field and clearly deserves attention. To this end, you make the case here that more and more residents in cities of the D.R. Congo have been impacted by growing urban gullies in past decades. The methodological approach of spatially intersecting gully outlines mapped from satellite images with gridded estimates of population density is simple and intuitive.

Yet this approach may fall short of capturing the complexity of the underlying causes, dynamics, and consequences of urban gullies. A more rigorous enquiry might include analyses and discussion of substrate and drainage conditions, land use, erosion rates, and other key controls on urban gullies.

For example, when and where do urban gullies form in cities due to which critical conditions? Do we expect a correlation between population density and urban gully formation?

While you point out some of these issues and problems, your study hardly prods deeper. Instead, you estimate the exposure of urban residents in terms of their spatial adjacency to gullies. Without any further details such estimates might gloss over a lot of important detail that may be necessary to understand, predict, and remediate the impacts of gullies and how people respond to them in cities elsewhere.

Demographics and socioeconomic dynamics might be more essential than simple population densities, if you are concerned with the exposure and risk here. While I like the way you point to this possibly under-recognised hazard, I found it difficult to appreciate objectively without the context of other urban geomorphic processes or hazards such as floods or landslides.

In essence, I believe this contribution is timely and perhaps of broader interest, but many of your inferences and conclusions rest on too simplistic spatial intersections, while unduly generalising local observations without broader supporting data or rigorous tests.

We sincerely appreciate the time and effort you have dedicated to evaluating our work, as well as your overall positive feedback and constructive suggestions. We found many of your recommendations highly relevant and have incorporated them into our revised manuscript.

While our earlier manuscript primarily aimed to highlight the growing problem of urban gullies in the DRC, we agree that it provided limited insights into the factors leading to the formation of these gullies. To address this, we now present a detailed spatial analysis of (peri-)urban areas with and without urban gullies to explore the environmental conditions that explain their occurrence. Our findings show that urban gullies are linked to soil and topographic conditions, but also to the presence of roads and the extent of built-up areas (a variable that is more reliable and relevant than population density, which is usually derived from built-up area). In combination with a systematic comparison of historical aerial photographs of the same affected areas from the 1950s, these analyses clearly demonstrate that the formation of these gullies is associated with the rapid growth of many of these cities. Building on these results, we also present a preliminary spatial model that allows the assessment (at a 1-km² resolution) of which urbanized areas in the DRC are susceptible to urban gullying.

Additionally, we now provide an in-depth analysis of the dynamics of gully expansion and its impacts on population displacement. For this, we updated our urban gully inventory to 2023, extending the observation period by 3 to 8 years depending on the city. We subdivided each phase of gully expansion into initiation, gully head retreat, and sidewall widening events, calculating the displaced population associated with each. While most research on gully erosion has focused on gully formation and head retreat, our results show that sidewall widening is the primary driver of population displacement. It occurs most frequently, exhibits the highest expansion rates, continues for longer time periods, and affects more densely populated areas. As discussed in the text, this helps explain why the impacts of urban gullies often persist for years or even decades but also has important implications for the design and implementation of gully stabilization efforts.

Building on these analyses, we also provide a more data-driven assessment of population exposure. More specifically, we now calculate exposure rates in two additional ways, based on observed gully expansion patterns. Using the distribution of observed gully widths and long-term (>10 year) average gully head retreat rates, we define 'potential' and 'likely' expansion zones for

each gully (corresponding to the 95% quantiles and averages of the distributions, respectively), making further distinctions based on the soil type in which gullies develop. This yields hazard zones directly rooted in our observations, making them less arbitrary than the previously used 100-m distance criterion. Our results show that the number of people living within the potential expansion zone of urban gullies (and thus directly exposed to the risk of displacement) clearly exceeds the population living less than 100m from a gully. By comparing this exposed population to observed average population displacement rates, we also quantify what these different levels of exposure imply in terms of the probability of displacement within the next ten years.

It is indeed true that the risks and impacts of urban gullies depend not only on population density but also on demographics and socioeconomic dynamics. Unfortunately, data limitations for the DRC currently largely prevent us from fully assessing this. Nonetheless, quantifying the number of exposed and displaced persons offers clear, objective, and interpretable measures that reveal alarming trends. As we note towards the end of our text, we believe this also underscores the need for further research on this topic.

Despite these limitations, we now offer further insights into these aspects. For example, our updated inventory demonstrates that the number of exposed and displaced people has not only drastically increased since 2010 but has also accelerated in recent years. We now provide an analysis to identify the main reasons for this rise in exposure. Results indicate that population increases in already existing gully expansion zones are a key driver, alongside the formation of new gullies and the expansion of existing ones. Together with our more detailed analysis of the different mechanisms of gully expansion and population displacement, these results offer important insights into the complex drivers and dynamics of urban gullies. Using a conceptual model, we also discuss the implications for impact remediation toward the end of our paper.

As requested, we also better place our results in perspective. To the best of our knowledge, urban exposure rates to other hazards like floods and landslides are unavailable for this region. However, in our discussion, we provide comparisons with previously published average exposure rates to various other hazards across Africa, as well as with a recent study estimating average landslide victim rates in rural areas of the DRC.

Building on this, we now present a much stronger argument that urban gullies are a clearly under-recognized hazard. The key reasons for this are: (i) the staggering number of people exposed and already affected (e.g., over 3 million people currently live within the potential expansion zone of an urban gully in the DRC); (ii) both population exposure and displacement rates are rapidly increasing and accelerating in recent years; (iii) urban gullies affect different areas of a city than landslides or floods and represent a complementary danger; and (iv) urban gullies may continue to cause destruction and impacts for years or even decades after their formation.

Overall, we believe these changes have significantly improved the quality of our work. Further details and responses to more specific concerns are provided below.

General Comments:

- The case for studying large urban gullies could be made a bit more stronger. Define what you mean by "large" and highlight better previous work. The introductory notes on large urban gullies offer a few mechanistic and geometric insights that readers may find difficult to extrapolate for the diversity of urban settings in the Global South. The phenomenon of gullies in cities cannot be that novel, judging from the previous studies you duly cite.

Thank you for this suggestion. We have rewritten the introduction to better explain what an urban gully is, how they form and expand, and what is already known about their occurrence and impacts in other urban settings of the Global South. Furthermore, our method section (M1) now includes a clear definition and set of criteria used to identify and map urban gullies.

As we argue in the introduction, there is indeed a small but growing number of case studies documenting the occurrence of urban gullies. However, a systematic analysis of their spatial patterns, controlling factors, and impacts at larger scales (e.g., country-wide) is clearly lacking. To our knowledge, this is the first study to address this research gap and demonstrate the enormous scale of the problem.

- The mechanistic background of gully formation and growth remains a bit shrouded and unduly brushed over in a general manner. This makes it hard to assess which processes are likely responsible for generating hazards and risks to the local people. Your mention of some of the impacts could be much better linked to the life cycles of urban gullies. I suspect that the introduction could work well with a more systematic and detailed review about what we know already and what we might learn from mapping urban gullies.

Indeed. As mentioned above, we now better explain the mechanisms of urban gully formation and expansion in the introduction, based on what is already known from earlier work. Furthermore, our analyses now take into account different mechanisms of urban gully expansion (i.e. the initiation of new gullies, gully head retreat and sidewall widening). This allowed us to better link the impacts of urban gully expansion to their lifecycle (see, for example, the conceptual model that we present at the end of our paper).

- The lack of systematic data on large urban gullies surely has several reasons. It would be good for readers to learn more about those reasons and also about whether perhaps other natural hazards in the urban Global South attracted more attention in terms of systematic mapping. This context of hazard and risk awareness might be important.

Interesting point. The lack of data on urban gullies likely has several interacting causes. First, urban gullies are primarily a phenomenon affecting the Global South, where, as in many research fields, documentation and research of geomorphological/natural hazards is generally much scarcer. This is likely due several factors, including limited funding, the absence of well-established research institutions and research capacity, and a lack of detailed primary and auxiliary datasets (e.g., on topography, rainfall patterns, demographics, infrastructure, and socioeconomic characteristics). Specifically, this study stems from a research and capacity-building project funded by the Belgian Government (the 2018-2024 ARES-PRD Project PREMITURG: Prevention and Mitigation of Urban Gullies). This project aimed to increase the capacity of Congolese universities and other stakeholders to better address the growing problem of urban gullies. To this end, two Congolese researchers (the first two authors of this paper) pursued their PhDs through joint programs. Together with their Congolese supervisor, Prof. Dr. Fils Makanzu Imwangana (also a co-author of this paper), they are among the first Congolese researchers to focus on this issue.

A second reason for the lack of systematic data and research on urban gullies is likely related to research traditions. Historically, gully erosion has been studied primarily in agricultural settings, with a strong focus on its impacts in terms of land degradation and, to a lesser extent, on aquatic resources. For example, an extensive review by Castillo & Gomez (2016; *Earth-Science Reviews*) revealed that only 3.1% of published articles on gully erosion focused on urban environments, and only 8.9% addressed tropical environments. As such, the idea of gullies as a significant geo-hydrological hazard in urban settings is, in our view, poorly acknowledged and documented in the scientific community.

Thirdly, as discussed in our text, urban gully erosion events often co-occur with other, better-studied hazards, particularly flooding, and may constitute compound events. This may further contribute to the lack of research. For example, during our research project, we observed that news coverage of casualties after intensive rainfall events in Kinshasa often attributed them to flooding, when in fact they were caused by urban gully expansion. Nonetheless, as argued in our manuscript, urban gullies are clearly a distinct type of hazard that warrants much more attention (both in terms of policy and research). By presenting the first systematic, large-scale assessments of the drivers and impacts of urban gully erosion, we believe our work helps highlight this emerging research area, which has relevance across multiple disciplines.

Due to the length limitations of a typical Nature paper, we were unable to fully develop these points in the text. However, our rewritten introduction and the final sections now better discuss the need for greater awareness of this underestimated hazard.

- The choice of the D.R. Congo as a study area remains somewhat unmotivated and may need some more justification. Where have most studies on urban gullies worked before? Do you wish to close a white spot on the map or expand and consolidate (or even challenge) previous work?

As explained above, our focus on the DRC largely arose from the fact that this work resulted from a research and capacity-building project between Belgium and the DRC. Based on the direct field knowledge of the Congolese partners in our project, we knew from the start that the problem was very large. However, until this study, we had no quantitative understanding of its scale.

While urban gullies certainly occur in other countries (e.g., Brazil, Nigeria, Angola, Republic of Congo, Mozambique), existing studies are too few and too scattered to identify clear patterns in urban gully occurrence or research attention. Nevertheless, some earlier studies (particularly the work of Makanzu Imwangana et al., 2014 & 2015; cited in the text) indicated that the problem was particularly large in the DRC (specifically in Kinshasa). This is now also mentioned in the introduction.

- Mapping of gullies. Whether or not most gullies are linked to urban sprawl depends on the sampling method. By design you mapped gullies in cities, so this dependence may be biased. It might be useful to cross-check whether gullies formed at similar or lower rates outside of the city areas, assuming all other controls constant or comparable at least. The reported dimensions of the gullies seem spurious and it may be helpful to replace them by measures of spread in the data. Without more details about the cities studied I found it hard to assess whether the reported extent and rates of historic gully erosion were low, high, or moderate. Again, a comparison to other rates of land-use change or turnover by other natural processes would help a lot. It might also help to list diagnostic and objective criteria for detecting gullies from satellite images. For example, how do you distinguish gullies from quarries or excavations with an elongate shape?

Our original intent was to provide an overview of the scale of urban gullies as a hazard in the DRC, focusing on the exposed and displaced populations. To achieve this, we studied very high-resolution imagery to map every urban gully that could be identified, based on criteria that are now more clearly outlined in the Methods section (M1). For this purpose, our dataset is clearly unbiased.

Nevertheless, we do agree it is relevant to demonstrate that these urban gullies are indeed linked to the process of urbanization itself. To this end, we conducted an additional mapping of 1-km² grid cells, randomly selected across all Congolese cities (n=754) and their non-urbanized fringes (n=715). In the latter category, only two cells were found to have gullies, clearly indicating that large gullies occur very rarely outside urban areas. This finding is further supported by detailed

analyses of historical aerial photographs from the 1950s (i.e., before the development of most urban areas; cf. Methods M1). Of the 2,922 identified urban gullies, only 46 were present in these photographs (Fig. S1), and in all cases, they were linked to the fledgling cities and/or the existing road network. Today, 98% of the identified urban gullies remain clearly linked to the road network. These figures are now clearly mentioned in the text to further clarify that these gullies are causally linked to ongoing urbanization.

Furthermore, we used our dataset of grid cells with and without (urban) gullies to identify the environmental factors best explaining gully occurrence. This resulted in a first spatial model that allows us to identify which areas in the DRC are most susceptible to urban gullies. Apart from slope steepness and soil type, the fraction of urban area and road density within the grid cells are key factors. As we now better explain in the text, we believe these findings unequivocally demonstrate that the formation of these gullies is driven by rapid, ongoing urbanization. Additionally, these results provide a first step toward better predicting where urban gullies may occur, contributing to improved anticipation and prevention strategies.

With respect to the reported dimensions, we agree that our previous reporting of these findings was limited. To address this, we now include boxplots (per city and per dominant soil type) of gully lengths, areas, maximum widths, and observed linear retreat rates in the supplementary information. Very little observational data on erosion rates exists for the DRC, making meaningful comparisons with other processes or land uses difficult, particularly since our approach quantifies only the spatial, not the volumetric, extent of gullies. Such a comparison is also beyond the scope of this research. Nevertheless, based on an earlier compilation of measured gully head retreat rates worldwide (Vanmaercke et al., 2016; see text for full reference), it is clear that the retreat rates of urban gullies observed in this study are very high.

- The population density estimates have 100-m resolution and are thus much coarser than the dimensions of the mapped gullies. Some error appraisal seems like a logical thing to add to this analysis. The penultimate paragraph in the main text alludes to errors, but describes them in a very superficial way that does the data hardly any justice.

Indeed, the major source of uncertainty affecting our estimates is the gridded population datasets used. Unfortunately, more precise or accurate datasets are currently unavailable at the scale of the DRC, which also impedes an exact quantification of the associated errors.

Yet, we now conducted several comparisons to estimate the uncertainty on our quantifications of exposed and displaced population (cf. Methods M6). Specifically, we calculated the population exposed to urban gully expansion for 2020 using two state-of-the-art gridded population datasets: WorldPop and GHS and assessed the differences at both the scale of the DRC and individual cities (Table S6, Fig. S6). The results showed that, while estimates were overall of the same order of magnitude, those based on the more recently published GHS dataset were generally higher. Further analysis revealed the cause of this discrepancy: especially in smaller cities, WorldPop likely underestimates the population, with unrealistically low values for many pixels that are clearly urbanized according to Google Earth imagery (see also Fig. S7). For this reason, and because WorldPop data is only available until 2020 (while our updated urban gully inventory extends to 2023), we now use the more accurate GHS data for our analyses.

Second, we compared the exposed population estimates derived from GHS data with detailed census data for the city of Bukavu, the only city for which such field-based census data are available. Here, the results show that, at least for Bukavu, our estimates based on GHS data likely underestimate the actual exposed population.

Based on these comparisons (and considering potential other sources of error), we now assume that our quantifications of displaced and exposed populations may be over- or underestimated by up to 40%. Where appropriate, corresponding error bars are now included for our estimates. While this 40% error margin is itself an estimate, it is likely a conservative one (i.e., especially at the scale of the DRC, total errors are likely lower as different errors may cancel each other out). Moreover, these errors do not alter the main conclusions of our paper: millions of people are threatened by this problem, and thousands lose their homes each year due to gully expansion, with numbers clearly increasing. This is especially true since our comparison with more accurately derived population estimates in Bukavu suggests that our numbers likely still represent an underestimation.

- Exposure vs. risk. The title reports "growing risks of urban gullies", though you hardly offer any risk estimates in the technical sense. The gully mapping reports frequencies and magnitudes at best, which should be converted into hazard levels. Much of the results refer to exposure instead by estimating how many people had been displaced (or injured if not killed) by expanding gullies. Using a linear trend on the cumulative data in Fig. 4 perhaps inadvertently also emphasises that the ratio of people affected over gully area has remained largely unchanged. This trend might also be biased by excluding gullies that have remained stable in shape (as described in the Methods). Defining risk on the basis of proximity to a gully (100 m in this case) seems a bit arbitrary and warrants more explanatory detail and robust testing. To this end, you undersell your data, as you should have headward and lateral expansion rates of gully areas available from your mapping.

Indeed, thank you for these very valuable suggestions. You were correct in pointing out that our numbers of threatened people reflect exposure rates, not 'risks' (which depend on various other factors that cannot be assessed at the moment due to a lack of auxiliary data). We have corrected this throughout the text. In the discussion, we also briefly address elements that likely further contribute to the associated risks, namely, that (extremely) poor and therefore vulnerable households are likely most affected by the problem.

We also greatly expanded our analysis of population displacement rates. More specifically, as suggested, we now differentiate between different types of gully expansion events (i.e., the formation of new gullies, gully head retreat, and gully widening) and their impact on displaced populations. While the overall ratio of people affected per unit area of gully remains fairly constant over time, these new results clearly show important differences in terms of frequency, expansion rates, population density in expansion zones, and displaced populations. One key finding is that sidewall widening is the dominant process driving population displacement. As we now discuss (based on the lifecycle of a typical urban gully), this has important implications for impact prevention and mitigation. Furthermore, while the population displacement per unit area of gully expansion remained roughly constant over the study period, this is clearly not the case for the average displacement rate per year. Using our updated urban gully inventory, we now demonstrate that displacement rates since 2020 are nearly three times higher than 15 years ago.

Building on this, we also revised our quantification of exposure rates. Apart from the 100m buffer distance (which remains, in our opinion, a relevant and easy-to-understand exposure criterion), we now also define hazard zones based on the statistical distributions of observed maximum gully widths and long-term (10+ years) gully head retreat rates. In these analyses, we differentiate between the population living in the potential expansion zone of urban gullies (and hence potentially at risk of direct displacement) and those living in the expected expansion zone (and thus potentially at much greater risk of displacement). Likewise, we account for pedological differences between the cities, which were found to best explain statistical differences in gully dimensions and expansion rates. By combining these different exposure quantifications with our average displacement rates, we now also link these exposures to estimated hazard levels (defined

as the likelihood of being displaced, losing one's house, or worse within the next ten years). Finally, as mentioned above, we now present an additional analysis that explains the reasons why exposure rates have doubled since 2010.

- The risk estimates you offer here are mostly exposure estimates in terms of people living close by to urban gullies. Readers might find it very difficult to assess whether, for example, a projected exposure of "33 persons per 100,000 inhabitants per year" (line 229) is high, moderate, or low without any further context. How do other hazards like floods, landslides, windstorms, traffic, or political conflict compare to this estimate? How reliable is this estimate in terms of confidence bounds?

Indeed, as discussed above, our estimates of both the displaced and exposed populations now include confidence intervals. Furthermore, we now provide concrete estimates of the displacement probabilities that the exposed population faces.

Unfortunately, very few data on exposure or impact rates for other hazards are available for this part of the world. Nevertheless, based on a literature search, we identified some points of comparison at the African scale (specifically, exposure to flooding, earthquakes, and volcanoes). Additionally, we contrast our findings with reported fatality rates due to landslides in East Congo. This clearly confirms that urban gullies are indeed a significant geo-hydrological hazard, with exposure rates generally higher than those for other hazards. Moreover, the absolute numbers we present offer a compelling demonstration of the significant exposure and impact levels (e.g., over 3 million people live in the potential expansion zone of urban gullies, and currently, over 13,000 people per year are displaced by them).

- Many of the inferences and conclusions in the last third of the text read as if they could apply equally well to many other natural hazards. Readers might expect a more targeted discussion and more specific conclusions that rest on your data.

We agree. As such, we have completely rewritten the discussion section of our manuscript. As already mentioned above, one part now explicitly focuses on providing a stronger argument for why urban gullies should be considered a new and important geo-hydrological hazard. The contrasts with other disasters (in terms of spatial patterns, duration, impacts, and potential solutions) are central to this argument. The second part of our discussion now focuses on how these gullies could be better addressed, building on our findings regarding the life cycle of urban gullies.

- The language and grammar are generally accessible and correct, though the text is quite wordy, and also vague, in places. I flagged several examples in the annotated manuscript file (see below).

Specific Comments: Please consult the annotated manuscript PDF for more detailed remarks and suggestions.

Thank you for your suggestions and efforts to further improve the writing. Given this and all other feedback, we have nearly completely rewritten the text, avoiding long and vague language. Where relevant, also your other specific comments were incorporated.

Referee #2 (Remarks to the Author):

This paper highlights the role of erosional gullies in densely populated areas in the DRC and their substantial impact on those populations. It also suggests this problem is ubiquitous across the Global South, likely displacing 10's of thousands of people per year.

I am pretty familiar with the gully literature and the badland literature, which this paper definitely fits into. So from that perspective, I don't think there is new scientific ground broken. That said, the framing of this issue as an urban hazard, and the demonstration of its pretty staggering impact, I think is original and significant, and for that reason I am supportive of publication.

The methodology make sense to me, and the approach seems reasonable, as do the conclusions.

Thank you for your efforts, constructive suggestions, and positive evaluation of our work. As you indicate, the innovative nature and relevance of this study mainly lies in the fact that gullies in urban contexts hitherto received much less research attention as well as in providing the first quantification of the enormous impacts and population exposure associated with them. By completely updating our inventory to 2023 and rerunning our analyses with more accurate population estimates, we now demonstrate that these impacts are even higher than initially estimated. Additionally, we show that the problem is not only growing but also accelerating.

Following suggestions from other reviewers, we have also conducted various additional analyses on the factors controlling urban gully formation, which has led to the development of a first susceptibility model, as well as on the dynamics of gully expansion. While many of these results indeed align with what could be expected for gullies in other contexts, this is, to our knowledge, the first time such an elaborate assessment has been made for urban gullies. Furthermore, the fact that most expansion is attributable to gully widening is noteworthy, as gully head retreat is typically assumed to be the dominant process.

My main suggestion for improvement of the manuscript is this:

Line 146-156 - I went and looked at a bunch of the gullies in GE, particularly in Kikwit. They are indeed really impressive and clearly quite destructive. One thing that comes out in the imagery however is that there is a competition between gully expansion and revegetation that is apparent. The gullies don't simply unzip in one direction, but rather seem to be periodically arrested or healed by vegetation regrowth.

This is consistent with observations from other badlands, for example see review from Gallart et al (2013) - Catena, that speak to the important role of vegetation regrowth in stabilizing gullies. Given the important management implications for this, it seems like a missed opportunity (and possibly an oversimplification of the problem) to not delve into the issue of vegetation and the punctuated growth of these networks.

For this reason, I think it would be great to add a section on the role of de-vegetation and re-vegetation on the erosional dynamics of these gullies.

Very interesting suggestion. We now discuss the potential role of vegetation in the last section of our paper, where we explore potential avenues to stabilize these gullies and mitigate their impacts (also referring to the study by Gallart et al., 2013). As you indicate (and as we now mention in the text), the role of vegetation is quite complex in these settings. Vegetation may help stabilize existing gullies, but we also frequently observed that vegetated gully sections were reactivated nonetheless. Based on our field observations, we believe that vegetation is often more

a consequence of gully stability than a cause. Hence, there is a need to combine vegetation-based measures (which, when successfully implemented, offer a cost-effective and sustainable solution) with more structural interventions.

Nevertheless, more (field) research is needed to better understand the role of vegetation. Overall, the vegetation dynamics inside urban gullies are very complex and do not only depend on its interaction with erosion. For example, we observed that local populations frequently plant species like bamboo in an effort to stabilize gullies. However, in other cases vegetation is often burned by others. This, because it is seen as a nuisance or to clear space for farming (e.g. banana cropping) or housing(!) inside the gully. Fully understanding these dynamics would clearly be a PhD study on its own and fell outside the scope of this paper. Nevertheless, this opens up interesting perspectives for further research. This is now also better indicated in our discussion.

I also noted one typo:

121 - change "is" to "are"

Thanks for noticing this. As the text was largely rewritten, also this error is now corrected.

Referee #3 (Remarks to the Author):

Thank you for the opportunity to review this paper. Key results include the mapping of urban gullies in the D.R. Congo over space and time and referencing urban gully expansion in the context of populations at risk for multiple time points. While I found the content and area of study interesting, I am not sure Nature is the most appropriate publication outlet, at least in the paper's current form. I respond in more detail below.

We thank you for your review and overall positive assessment. Your suggestions were valuable and helped us improve the quality of our work. This includes a better assessment of the uncertainties associated with our estimates as well as a more in-depth analysis and argumentations as for why these gullies are indeed caused by urbanization (and hence a new hazard of the Anthropocene). Further details are given below.

Comment on the relative importance of urban gullies compared to other geohazards?

To be honest, I am not that familiar with urban gullies but that doesn't mean they are not important. I found the paper very interesting considering this type of hazard is understudied combined with the complexity of addressing sustainable solutions to the issue. I think it would be insightful to expand upon how urban gullies persist in contexts beyond the D.R. Congo – currently this is only noted by citations in Lines 46-47 but the implications of it being a hazard more specific to the Anthropocene could be unpacked more, relevance to both Global South and North or why Global South so much more vulnerable. Authors also reference other cases in Lines 222-224, but again, I think drawing some commonalities or identifying some points of reference that provide the larger "so what" context for this particular hazard is important.

Thank you for your positive feedback and thoughtful comments. From our experience, unfamiliarity with the issue of urban gullies is relatively widespread among researchers and policymakers, which, in our opinion, underscores the timely nature of this work. We now elaborate more extensively on how urban gullies and their impacts are closely linked to the process of rapid (often uncontrolled) urban expansion, positioning them as a new geo-hydrological hazard of the Anthropocene. Linked to this, we now explain more clearly why cities in the tropical Global South are particularly susceptible to this problem and why (as our updated results show) the problem is accelerating. This is supported by additional analyses, by referencing case studies

from other countries affected by urban gullies and with supplementary materials (e.g., Fig. S1 and S8).

Providing exact quantifications of the contrast between the Global North and Global South is, at this stage, impossible, as it would require more detailed data on infrastructure characteristics that are simply unavailable for the DRC and other countries (e.g., how many parcels have facilities to capture rainfall and runoff, or whether roads have appropriate runoff management measures). Moreover, also for other Global South countries affected by urban gullies, no comprehensive assessments of the scale of the problem currently exist. However, based on our earlier work and other case studies, we now better discuss elements of the Global South context that contribute to the risk of urban gully formation. Overall, we believe that the fact that urban gullies are rarely reported in the Global North (where there are fewer data limitations and such events would likely be significant news) further indicates that this new type of geo-hydrological hazard predominantly occurs in the Global South.

We also place our results further into context. For example, a comparison of exposure levels now clearly demonstrates that urban gullies in DRC are at least as significant as other, better studied, hazards like floods or volcanos. Building on this, we now present a more developed argumentation for why urban gullies should receive much more research and policy attention.

Does the manuscript have flaws which should prohibit its publication? If so, please provide details.

The authors have done a lot of spatial mapping work using Google Earth (mostly) to identify the extent, rate, and existence of urban gullies in the D.R. Congo. I think an interesting aspect lacking from the paper is a compelling argument, in any form, regarding the underlying drivers for the changes. I don't understand how the authors conclude that urbanization (how is urbanization defined?) is the reason for gully expansion (Lines 88-89).

This was indeed an element that was somewhat lacking in our previous manuscript. Building on the comments from referee #1, this is now thoroughly addressed. More specifically, our introduction now provides a clearer explanation of what is already known about the mechanisms and driving factors of urban gully erosion.

Furthermore, we conducted additional mapping to create an extensive dataset of 1-km² grid cells with and without urban gullies inside and at the fringe of urban areas. Using this dataset, we now statistically analyze which factors best explain the spatial patterns of these gullies across the DRC (Methods M3). The results clearly show that urban gullies predominantly occur in areas with steeper slopes and sandy soils. However, road density and the fraction of built-up area in each 1-km² grid cell (which we consider key indicators of urbanization) also have a dominant influence on urban gully occurrence. Combined, these four factors allow for reasonable predictions of where gullies occur.

Moreover, for every urban gully, we checked whether it was present in historical aerial photographs from the 1950s (cf. Methods M1), before most of these cities were constructed. Results show that, although deforestation had already occurred in these areas, the overwhelming majority of gullies (>98%) were not yet present. The few gullies that did already exist were clearly linked to road networks and/or in proximity to housing. Even today, we observe that approximately 98% of the current urban gullies are closely associated with the road network. A comparison of identified urban gullies with the zonation plan of Kinshasa (Fig. S8) further shows that most gullies occur in areas of unplanned urban sprawl with older gullies being located closer to the original city center. We believe that these findings clearly demonstrate that the formation and expansion of these gullies are indeed driven by urbanization (i.e., increases in built-up areas

and road networks). As we now discuss more thoroughly, this is consistent with earlier case studies.

Additionally, we now demonstrate that, regardless of these gully dynamics, a significant portion of the increased exposure levels is linked to population density increases within existing hazard zones. Given that many Congolese cities (and many cities in the Global South) are growing rapidly, this clearly indicates that the problem of urban gullies will likely escalate dramatically over the coming decades.

Authors note in lines 77-80 the importance of rainfall and changing frequency/intensity of rainfall due to climate change. How do authors address that component in relation to urban expansion and population density increases over time? I also cannot find information relative to environmental characteristics, and notably rainfall information, other than topographic and substrate characteristics portrayed in Figure 3.

This was an element that was indeed somewhat poorly addressed in the previous version. To address this, we have now included proxies for rainfall (and especially rainfall intensity) as variables potentially explaining contrasts between areas with and without urban gullies (cf. Methods M3). Overall, our results show that these rainfall characteristics do not exhibit significant differences between gullied and non-gullied 1-km² grid cells. This contrasts with, as discussed above, the fraction of built-up area and road density. While this may seem surprising, it is actually logical. Specifically, most cities in the DRC are characterized by very similar rainfall conditions, as they all lie within a tropical climate zone. Therefore, rainfall conditions are not a significant factor in explaining spatial patterns of urban gullies (especially as we also compare areas with and without such gullies within the same city; cf. Methods M3).

However, it is clear that intensive rainfall events are a key driver of urban gully formation and expansion: without rain, there is no runoff and therefore no erosion. As we now briefly discuss in our text, climate change may therefore further exacerbate the problem of urban gullies as it will lead to more rainfall with important intensity. Nevertheless, our new results mainly demonstrate the importance of roads and building density. As such, we expect that further increases in road networks and built-up areas (and their resulting effects on the rainfall-runoff response) will be the primary driver of future urban gully erosion.

The visually observed built area (“urban”) expansion using aerial photographs from the 1950s - if gullies were absent before the area became urbanized and then appeared – it’s purely a function of development? What about other confounding factors? I feel like the analysis could be pushed further to discuss the environmental context in which cities are growing – the relationship of the substrate and topography relative to population increase and settlement/built area expansion – authors state that most PAR are located in Mbuji-Mayi and Kinshasa (Line 197) – what is different about those areas from others?

Indeed. As discussed above and building also on the comments of Referee #1, we now conduct a detailed geo-statistical analyses to see how urban and peri-urban zones (i.e. 1-km² grid cells) with urban gullies differ from those without. Results show that this mainly depends on the road density, extent of built-up area, local topography and soil characteristics. These factors are indeed sometimes correlated to each other. The susceptibility model we present in this paper helps accounting for this.

This model now also makes it easier to understand why Kinshasa, Mbuji-Mayi & Kikwit are the most affected cities (although the ranking slightly changed due to our updated inventory and analyses): they are the largest cities of DRC and/or have extensive zones highly susceptible to UG

formation. This leads to a large number of urban gullies and, by extent, many displaced and exposed persons.

Authors note that there are 166 gullies where no expansion occurred and thus probably are stable. I feel like further investigation relative to unstable gullies would add insight, notably are there any differences on the location of these non-expanding gullies relative to population density and associated proximity, road infrastructure and built settlement development, etc.

Interesting suggestion. However, this number (which is also slightly changed based on our updated inventory) should be somewhat nuanced as it only indicated the number of gullies that were already present on the earliest Google Earth images and showed no indication of expansion over the entire observation period. Many more gullies showed phases of stability over shorter time periods.

Several factors may contribute to gully stability. One element seems to be age: younger urban gullies tend to drastically increase in length first but then show a tendency to stabilize over time. Nonetheless, various reactivations of older gully heads were observed. Furthermore, many urban gullies continue to expand sidewise for over a decade. As such, our data does not seem to suggest there is a certain age after which urban gullies should be stable. A second factor that likely plays an important role is the implementation of gully stabilization efforts (e.g. canalizations of the gully thalweg, reinforcing the gully head). Yet, the exact effect of such initiatives is highly context dependent and therefore impossible to predict. For example, based on extensive field surveys, we recently showed that most of the implemented erosion control measures in Kinshasa do not lead to a significant reduction in expansion rates (Lutete Landu et al., 2023; see text for full reference). Finally, as also suggested by referee #2, vegetation inside the gully might play a role. However, as discussed above, vegetation is certainly not always a direct cause of gully stability. Rather it may be a consequence of it.

Overall, it is therefore very hard to say what exactly causes some gullies to become stable (or better why many gullies remain active). Apart from rainfall variability and the complex topographic setting in which urban gullies form (cf. introduction), analyses are further complicated by the fact that many urban gullies often initiate in peri-urban zones with lower population densities but then expand into more densely inhabited zones. Meanwhile, during these expansion phases, the fraction of built-up area continues to increase (see e.g. the conceptual model presented in Fig. 6 of the new manuscript and the accompanying discussion). This makes it difficult to establish a clear relation between population density and gully growth. Nonetheless, as discussed in the reply to suggestions of reviewer 1, our results do provide some key insights into how urban gullies can be better mitigated. This is now better discussed in the concluding section of our paper.

I think this is a publication worthy effort but I am not sure for Nature, at least as it stands. An additional strength to this paper would be analysis that better supports the question, What are the processes for creating exposure?

Thank you for your assessment. We believe that the new, more in-depth analyses of our reworked manuscript now better identify the factors and mechanisms that lead to population exposure and displacement. Overall, this further supports the central claim of our work: urban gullies are a hitherto largely neglected but important and rapidly growing problem. This message has important consequences across a broad range of stakeholders and scientific domains. As such, we believe *Nature* is a suitable outlet of our work.

Also, I do wonder if others have done any similar work but with alternative remote sensing sources such as radar, for mapping our urban gullies and expansion rates.

No, to our best knowledge, we are the first conducting an analyses on the occurrence, dynamics, population exposure and impacts of urban gullies at larger scale. Smaller case studies on urban gullies exist. Yet, they typically focus on only one or a few gullies and/or provide no systematic analysis on the causes, impacts and spatial variability.

In the future, it may be indeed worthwhile to explore other avenues of remote sensing (e.g. radar) to better map urban gullies and their expansion rates. Nonetheless, this will also come with additional technical challenges. For example, mapping the extent of urban gullies requires imagery of sufficiently high resolution (< 1m) while also the complex urban context may render image interpretation more complex. Overcoming these challenges was outside the scope of this paper, but opens promising perspectives for further research.

If the conclusions are not original, it would be very helpful if you could provide relevant references.

I am not aware of other work that duplicates these current paper's conclusions.

Indeed. As explained above, this is -to our knowledge- the very first study demonstrating that urban gullies are such a large and growing problem.

On a more subjective note, do you feel that the results presented are of immediate interest to many people in your own discipline, or to people from several disciplines? If you recommend publication, please outline, in a paragraph or so, what you consider to be the outstanding features.

I think the topic area could be of interest to those in geographic disciplines. Aspects of this publication that are interesting include 1) drawing attention to a less studied/publicized but impactful natural hazard and 2) proof of concept approach that could be applied in other places for mapping out PAR in relation to urban gully extent and expansion.

Indeed. We believe our work is of great relevance to a wide range of audiences because urban gullies are at the interface of several domains. I.e., they are a hydro-geomorphic phenomenon that is tightly linked to the ongoing rapid urbanization of the Global South. As discussed in the concluding paragraph of our paper, better addressing this new hazard will therefore require an interdisciplinary approach that spans numerous disciplines.

When reviewing the paper, we would be grateful if you could pay particular attention to the statistics, if applicable.

The authors state they use WorldPop data and reference "a series of gridded datasets that provide estimates of population density at a spatial resolution of 100 m and on an annual basis (2000-2020)." That is a data product that was parameterized for consistent and comparable mapping across countries, globally. As such, it will not be the most ideal, country-specific option for a contemporary analysis of population distribution when selecting a gridded population map. However, for comparing population counts over time, within the time frame given, this is an appropriate data product and probably the most reasonable choice

There are a variety of WorldPop products (did they use the UN adj or unadj version?). I would encourage specific language in M3 to note 1) that the global product is a specific WorldPop product, appropriate for use in this analysis but not the only option available from WP and 2) slightly more information on the disaggregation process to derive the gridded products (Stevens et al., 2015 good ref to complement ref #7) and 3) considerations for underlying uncertainty, esp for a country where

there does not exist consistent, reliable census counts (Gridded population estimate datasets and tools. - WorldPop). It's important to acknowledge that the data underlying PAR estimates are a function of a statistical model informed by other data – settlement and other land cover, NTL, climate variables, etc that will be influenced by changes in those gullies and subsequently induce error in the population estimates in an interactive way.

Indeed. While WorldPop clearly comes with limitations it was, at that time, the best product available for our purposes. However, for this new version of our manuscript, we switched to using the JRC GHS population dataset. References and further details for this are provided in the Methods M4. While this dataset is only available for 5-year intervals, we chose this dataset for several reasons. First, analyses and evaluation efforts showed that WorldPop was in many of the smaller affected cities inaccurate, whereas the GHS dataset gave more realistic results (see also below). Second, WorldPop is only available until 2020 while our updated inventory runs until 2023.

Nevertheless, also with this GHS dataset, it is true that population estimates are ultimately the result of a statistical model that makes inferences based land use and other variables that may interact with the presence of urban gullies and therefore remains subject to uncertainties. We believe this is now better acknowledged in our manuscript. For example, we now assess the potential uncertainties on our estimates of exposed and displaced populations (see below).

One potential suggestion for the authors would be a comparison for a single year placed in the supplemental that shows estimates from different products. Authors could compare their most recent mapping gridded population product from the global data to a more contemporary mapping of D.R. Congo population from the GRID3 project and/or an individually constrained or unconstrained choice from the WorldPop site. There is also JRC GHS-POP data on gridded population that could be compared (and had a temporal component although at a 5-year time step). I only suggest this type of comparison for authors understanding of how varied the gridded estimates might be relative to data product used. But I recognize they are trying to estimate PAR over time and in doing so, their data product of choice is the most reasonable one for their purposes. However, I would include the data source as a real and legitimate source of uncertainty regarding population numbers and PARs stated in the paper (lines 291-292). And perhaps a short explanation for the “evidently” aspect of line 173.

Thank you for this valuable suggestion. Following also the feedback of reviewer 1, we now conducted a comparison between our results obtained with JRC GHS-POP data with those obtained through WorldPop. As explained higher-up, GHS typically results in higher estimates. Yet, these results are also more realistic. In addition, we conducted a similar comparison based on field-based census data (which are unfortunately only available for Bukavu). Here, results suggest that estimates based on GHS-POP data may still underestimate the actual number of exposed and displaced population (although less so than WorldPop). Based on these different comparison, we now provide an uncertainty estimate on our assessments of exposed and displaced population (cf. Methods M6 and reply to referee #1). As a result of these changes, also line 173 was rewritten.

As an additional comment, Figure 3 is a very nice figure. I would suggest making Figure 4 with larger font and legend sizes. Also, lines 313-314 – geomorphic criteria for urban gully needs to be more detailed, not just cite a paper.

Thank you. Following our updated analyses, Fig. 4 was remade and now has larger font/legend sizes. With respect to the geomorphic criteria of urban gullies: indeed. Following also the feedback of referee #1, this is now better explained in Methods M1.

And then for clarity, in the supplemental - Are the 2010, 2015, and 2020 estimates of # of people in risk zones based on the D.R.I. or the gridded product from that year? That's unclear and a little confusing as the estimated WP total is relative to the D.R.I from Table S1 (if I read that correctly).

This was indeed somewhat confusing. These estimates were based on the extent of the gullies in that year (2010, 2015 or 2020) as well as the population density as indicated in the gridded population dataset of that year (not the population in the year of the reference image). Following our new analyses, all tables were updated accordingly and the captions were further clarified. To avoid confusion, the corresponding calculation procedures are now also better explained in Methods M5 & M6.

Referees' comments:

Referee #1 (Remarks to the Author):

Review of "The growing problem of urban gullies in the Democratic Republic of the Congo" submitted by G.I. Mawe and colleagues

Dear Authors,

This is my second review of this study, hence I refrain from summarising this any further. Many thanks for providing such a thorough reply letter to the original comments and suggestions by all three referees. I appreciate your efforts of collecting and presenting additional new data, and also redrafting large sections of the original manuscript. Overall, I think you have put together an interesting study.

We thank the referee for his review, the appreciation and the numerous critical suggestions and comments provided. We addressed them in detail and believe that many of them have further improved the quality of our work. Nevertheless, we also disagree with some of the comments or issues raised and believe they may be attributable to a misunderstanding of our objectives, methods and/or results. Detailed replies are provided below.

I still doubt, however, that it is novel, ground-breaking, or provocative enough for the journal. Nowhere do your results reveal any new insight to mechanisms and drivers of these gullies, or even their consequences (people killed or displaced, houses destroyed, infrastructure damaged, etc.). Instead, much of your conclusions still rest on the simple, and likely biased, spatial intersection of gully and urban population data. Yet, even the most basic statistical inference you use highlights some substantial uncertainties and inconsistencies that your discussion fails to pick up.

We trust the editorial team in assessing whether our work meets the requirements to be published in *Nature*. Nevertheless, as authors, we are strongly convinced that our work is sufficiently novel, ground-breaking or provocative. This was clearly also the assessment of the two other referees.

As we indicate in the introduction and further discuss below, several local case studies already provide insights into the mechanisms of urban gully formation. However, so far, no attempts were made to also look at the problem of urban gullies at larger scales. I.e., most work so far focused on a few individual gullies. As a result, it was hitherto unclear to what extent urban gullies (UGs) are a relatively rare phenomenon with overall limited impacts or a frequently re-occurring problem that affects many people. We strongly believe addressing this question is of a higher priority than further finetuning our mechanistic understanding of urban gully erosion.

Our results clearly support this. More specifically: (i) we show that the problem is clearly widespread and growing (with >2,900 UGs already mapped); (ii) we show that ca. 6,000 persons/year lose their house due to urban gully erosion and that this rate is drastically accelerating since 2020; (iii) we show that over three million Congolese are directly threatened by this hazard and that these numbers are rapidly increasing; and (iv) we provide detailed insights into where urban gullies form and how their different processes of expansion (i.e. initiation, gully head retreat and sidewall widening) interact with (changes in) population density to lead to such high displacement and exposure rates. These findings have direct implications for scientists, urban planners, policy makers as well as for efforts aiming to prevent and mitigate urban gullies.

As we further elaborate in response to more specific questions, we also strongly disagree that our dataset is affected by biases or that “even the most basic statistically inferences” we use are prone to “inconsistencies” that we fail to pick up. While our definition of an urban gully indeed implies a proximity to buildings, we clearly demonstrate in numerous ways (including through the additional mapping of the presence/absence of gullies in nearby non-urbanized areas as well as on aerial photos predating the urbanization) that building and road density are indeed critical factors in explaining the occurrence of urban gullies. These factual and counterfactual evidences are further confirmed by several earlier studies (as discussed below).

Yet, even in the highly unlikely case that similar gullies may also occur under natural/rural conditions, this does not invalidate our main findings in any meaningful way. A parallel to other hazards can be drawn here. For example, floods and landslides also occur under natural conditions. However, this does not imply that these hazards cannot become major disasters when they happen in densely urbanized environments. Hence, even if the causal link between urbanizations and the formation of UGs would remain unsure (which we clearly show is not the case), it remains essential to quantify the impacts and degrees of exposure linked to them.

As we explain further down in this reply, also other statistical inferences we make within the paper are clearly reliable, despite their inherent uncertainties.

It is true that our work mainly relies on spatial analyses of mapped urban gullies and population density data. This indeed limits the level of insight we can provide with respect to the different impacts of UGs. However, unlike the referee writes, we clearly do provide quantifications on the number of displaced persons (see e.g. Fig. 4).

As we discuss in our paper as well as in the reply to more specific comments, assessing other impacts (or even comparing them to other hazards) is severely limited by a lack of reliable data in the DRC (and for that matter many other African countries). As such, much work remains to be done in terms of assessing the numerous other impacts of UGs (e.g. in terms of casualties, injuries, specific damage to infrastructure, socio-economic psychological and intangible impacts) as well as better understanding the vulnerability of the exposed population.

Nevertheless, so far, urban gullies are generally not even considered in disaster assessments or reduction strategies. We believe our work provides a critical and urgently needed step to raise more awareness on this hitherto neglected problem.

General comments

- The introduction makes some case for urban gullies as an understudied hazard, but could do a much better job by summarising more objectively the current knowledge about causes, triggers, thresholds, growth rates, and impacts (even if based on case studies). Many of the qualitative and unduly overarching descriptions could acknowledge at least the knowledge base from a large literature pool on gullies. My initial reservation about the lack of a more mechanistic summary has not really been addressed in the revisions that well.

While a more in-depth review on the current state of knowledge on urban gullies may indeed be interesting, we believe this is outside the scope of this paper. One key limitation, is that the *Nature* article format only allows for an introduction of 500 words. As the referee will appreciate, this is only sufficient to provide a brief summary

of the current state of knowledge, including a summary of: (i) what they are and why they are worthwhile investigating; (ii) what is known about their controlling factors; (iii) what is already known about their impacts; and (iv) what are some of the current main knowledge gaps.

While limited in length, we do believe we provide an apt knowledge summary with respect to these elements. In doing so, we do refer to the majority of the relevant literature in this topic, including several earlier review studies that highlight several of the knowledge gaps mentioned here. Examples include the recent work by Kuhn et al. (2023) that highlights the need for quantitative assessments on the impacts of gully erosion, as well as several well-cited review papers on gully expansion (Vanmaercke et al., 2016) and gully erosion modelling (Vanmaercke et al., 2021). Furthermore, we discuss what is known about the main drivers and controlling factors of urban gully erosion, based on the most relevant case studies.

Hence, we believe the introduction provides a brief but relevant and apt review of the current state of knowledge. Given their lack of comments on this, we believe also referee 2 and 3 share this opinion. It is also unclear to us what exactly referee 1 means with their comment that we should “acknowledge at least the knowledge base from a large literature pool on gullies”. However, should they have more specific suggestions on elements or articles to include, we would be happy to consider them.

- In my previous review, I had also suggested some context about urban gullies compared to the many other sources of potential disasters: perhaps widen the lens a bit to avoid too much bias. While I appreciate your outlook on this (lines 216-227), I think it is misleading to compare the percentage of people exposed to urban gullies in the D.R. Congo to the percentage of all African people exposed to other natural hazards, as the density of both population and hazard sources differs widely; especially short-term rate estimates can be highly volatile (inset to Fig. 4b).

We acknowledge that the comparisons we provide with other hazards remains somewhat rudimentary. However, as we further clarify in response to the specific comments below (including through an additional analysis of the international disaster database EM-DAT), reliable assessments on the impacts and exposure rates for other geo-hydrological hazards are largely lacking for the DRC. As such, we remain restricted to this comparison at African scale.

Nevertheless, we do not agree with the fact that this lack of comparisons at the scale of the DRC would induce “too much bias”. What we present, is an assessment on the impacts (in terms of displaced people) of urban gullies and the population exposed to them at the scale of Congo. In absolute numbers, these numbers are extremely concerning. Also when expressed as a fraction of the total Congolese population, these numbers remain clearly significant (as discussed in the text). It is here that the exposure rates to other hazards provide some meaningful insights, even if they are only at the scale of Africa. We believe this limitation is clearly expressed in the text. Moreover, we fail to see how a comparison with other hazards/risks would change our finding. For example, suppose there is a threat in Congo that threatens a much larger of the fraction the population: would this invalidate our conclusion that ca. 3.2 million people are threatened by this problem and that >118,000 people already lost their house due to urban gullies? At no place in the text, we claim that urban gullies are the only or most significant problem.

It is also simply incorrect that the estimates we present were conducted on the short term and therefore ‘volatile’. We believe the referee may have misinterpreted Fig. 4b in this context. Individual expansion events can indeed be volatile (sometimes displacing >1,000 persons at once). However, the assessments we provide here are based on an analysis of >8,000 observed expansion events of ca. 2,900 mapped urban gullies across 26 different cities in the DRC and over a time period of ca. 20 years. Also the inset showing the drastic increase in displacement rates is clearly not just the results of a few extreme events. Our observation that displacement rates have nearly tripled in recent times, is based on the comparison of the country-wide average over a period of 4 years (2020-2023) and thousands of gullies as compared to the average of three consecutive 5-year periods before (2005-2010, 2010-2015, 2015-2020). Given these large numbers of observations and considerable observation periods, we are highly confident that the numbers we present are reliable and representative.

• Your attribution of gully formation to urbanization still solely hinges on spatial proximity. Simply because a gully is close to a road or buildings is no confirmation that these structures have caused the gully. Urban areas have dense road network by design, such that any location (gullied or not) is likely to be close to a road. To be more objective, you may need to demonstrate that the gullies could not have formed without urbanization. This sort of counterfactual approach has become a key ingredient in attribution studies. At the least you may want to correct for effects of road density. Your definition of urban gullies (“the gully needed to be located within 200 m of buildings”, line 376) might introduce a bias to areas of high population density already. Would it not be fair to map all gullies instead to get a more objective view of the potential role of urbanization.

While our definition of an urban gully indeed logically implies the proximity to building, we strongly believe we do provide ample of evidence for the fact that urban gullies are indeed attributable to the presence of roads and buildings, including through additional mapping in areas without dense road networks or urbanization as well as through a counterfactual approach. In addition, numerous earlier studies and field observations directly confirm this. We provide more elaborate responses to these matters in the specific comments below. Nonetheless, we summarize our main results and lines of evidence here:

- 1. For every urban gully we identified, we also checked on a series of historical aerial photographs whether these gullies were already present in the 1950s, i.e. before most of the urbanization took place (see Methods M1). For this, we checked the entire extents of the current city limits and also checked whether other gullies were present. Results showed that the large majority (ca. 98%) of the UGs were not yet present. Meanwhile, large extents of the area were already deforested/cultivated. Yet, almost no gullies were present. In addition, the limited gullies that were already present (n=46) clearly formed in the direct vicinity of the starting cities or road networks that were constructed during this early phase of urbanization (despite the fact that road and building densities were still very low). Examples of this are provided in Fig. S1. The images were clearly of sufficiently high resolution to easily detect gullies. As such, this analyses already shows univocally in a counterfactual manner that these gullies are a relatively recent (i.e. post 1950s) phenomenon that did not occur before the urbanization, even if these areas were already deforested and under cultivation.**
- 2. Also our mapping of urban gullies on recent imagery provided clear evidence that these UGs were indeed caused by urbanization. We frequently observed**

the formation of new urban gullies on available time series in Google Earth (n=857, cf. Fig. 4; also see Fig. 1d&f for an example). Nearly always, this happened after urbanization in the immediate upslope areas occurred. In later stages, these gully heads migrate towards the urbanized areas (i.e. the area that contributes the runoff). The other gullies that were mapped were already present on the first high-resolution image available. Hence, their formation could not be witnessed (this mainly relates to the fact that suitable imagery is only available for recent years). However, these gullies show very similar patterns and evolutions. We also remind the referee that all mapped polygons of UGs are made available. Hence, they can easily verify this in Google Earth.

3. While we used a 200-m threshold to define a gully as an urban gully, we clearly also looked to areas that were further away from buildings (see also next point). However, in these areas, gullies were almost never detected. In addition, for most gullies we identified, the distance to roads and buildings was much smaller than 200m. More specifically (and as briefly discussed in the text), ca. 98% of the urban gullies formed along (i.e. directly on) roads or just downslope of them. This, while the road density in these peri-urban areas is often still relatively low.
4. Following the referee's earlier request, we also randomly sampled 715 grid cells of ca. 1 km² in the non-urbanized areas surrounding the affected cities (cf. Methods M3). Based on the same Google Earth imagery as we used to detect UGs, we checked whether (non-urban) gullies occurred in these cells. We only detected gullies in two of these cells. This clearly indicates that gullies occur only very rarely in non-urban contexts (i.e. probably less than 1 per 100 km²). This finding is also further confirmed by other research (e.g. De Geeter et al., 2023), showing that the susceptibility to gully formation in rural areas of DRC is overall (very) low (see also reply to more specific comments; further below).
5. Contrarily to what the referee indicates, we did also performed univariate and multivariate analyses to identify the environmental factors that best explain the occurrence of UGs, resulting in a counterfactual analysis (cf. methods M3). These analyses were conducted at the same resolution of ca. 1 km² as the previous step. We classified all cells in which at least one (urban on non-urban) gully occurred as '1' (n=754) and all cells without gullies (i.e. the 713 non-urban cells with confirmed absence of gullies) as '0'. To further increase the robustness and reliability of our analyses. We also randomly sampled an additional 754 cells without gullies inside urban areas across all cities of DRC. For each of these cells, we extracted a wide range of variables, describing the amount of built-up area and roads as well as other environmental factors. The results (as discussed in the text, shown in Fig 3 and further elaborated on in the supplementary information) clearly show that the amount of built-up area and road density is clearly significantly higher ($p < 0.0001$) in cells with (urban) gullies as compared to cells without. This, despite the fact that also 754 of the cells without gullies were sampled inside urbanized areas. Also slope and sand content were significantly higher in cells with gullies. However, even when combining all these factors in a logistic regression model, built-up-area and road density clearly remained highly significant predictors of gully occurrence (cf. Table S3). Combined, slope steepness, soil type, built-up area and road density allow already to robustly simulate which areas are most prone to UG formation (cf. Fig. 3b, S3, S4). Overall, we believe this model clearly presents the kind of "counterfactual" analysis the referee is asking for. This is also evident when looking to the output of our model when applied to the different

affected cities and their peri-urban fringe (cf. Fig. 3, S4). In most cases, the areas surrounding most cities clearly show a much lower susceptibility to gullying due to the lower density of buildings and roads. This prediction is confirmed by the overall lack of gullies in these zones.

6. Furthermore, it is worthwhile pointing out that several earlier case studies already clarified how urbanization and the accompanying construction of roads can cause the formation of gullies. Examples of these studies and their key findings are cited in the text and further discussed below. As such, within the current scientific literature review, there appears to be very little doubt that these gullies are indeed caused by urbanization. The key principle behind this is straightforward: the construction of roads and buildings leads to an increased in sealed surfaces. This limits infiltration and hence increases runoff production. In addition, roads are highly effective in concentrating this runoff. As soon as there is some slope steepness, this leads to the ideal conditions for runoff formation. It can also be mentioned that in many sandy soil regions where the DRC cities are built, field-based experiments show that runoff is nearly impossible in natural land cover conditions (e.g. Moeyersons et al., 2015).
7. Last but not least, the formation of urban gullies has been frequently observed on the terrain during our numerous field campaigns (see also e.g. Makanzu Imwangana et al., 2015; Lutete et al., 2023; 2025; Ilombe Mawe et al., 2024). At many instances, we observed how new gullies formed and expanded along existing roads in (peri-)urban areas during or shortly after intensive rainfall events. The pictures below may help providing an intuitive understanding of the context in which these gullies form. In addition, [REDACTED] movie clip provide footage of how runoff accumulates along a road and generates enough shear stress to form a gully head that already destroys houses. The footage was recorded by a citizen observed in Kolwezi on 14th of March 2023, following a heavy rainfall event.

(Kinshasa, November 2019, Photo: M. Vanmaercke)

(Kinshasa, April 2018, Photo: M. Vanmaercke)

[REDACTED]

(Kinshasa. Photo taken from: Makanzu Imwangana et al., 2015, Catena)

- Your results still fall short of supporting your conclusions. Fig. 3a shows how the percentage of built-up areas is the most uninformative for distinguishing areas with urban gullies from those without. A more formal and flexible way would be to run a classifier to see how well built-up areas can predict locations with mapped urban gullies. Fig. 4b shows that new urban gullies form most rarely and preferably in areas of low population density; this observation indicates that maybe urbanization alone is not the sole explanation of gully formation. Should the title of your study not more likely read: “The problem of growing urban gullies...” instead of “The growing problem of urban gullies...”.

We disagree with these remarks and believe they result from a misinterpretation (and potentially incomplete reading) of the figure, text and methods.

Regarding Fig. 3a, the boxplots indeed show a broad range in percentage of built-up area (BUA). However, the reasons for this are already explained above. For cells without (urban) gullies, we sampled cells both inside (n=754) and outside (n=713) city

limits. This to avoid biases in our analyses. For cells with a gully (n=754, of which 752 with an urban gully and 2 with a non-urban gully), several cells show a lower fraction of BUA because they are still located in peri-urban zones. However, in most cases, they then receive runoff from neighboring upslope cells where urbanization is already further advanced (Fig. 1 shows some typical examples of this). Nevertheless, despite this overlapping range, BUA remains a highly informative variable. This is already evident from the boxplots: the median amount of BUA in a cell with a gully is over 70%. For cells without (urban) gully this is 0% (Fig. 3a). This, despite the fact that more than half of these were sampled within city limits. As indicated in the caption of this figure, a two-sided Mann-Whitney U test further confirmed that both distribution are clearly significantly different ($p < 0.0001$). Road densities show an even larger difference.

Furthermore, we did run a classifier (i.e. logistic regression) model to test whether BUA is indeed relevant in explaining observed differences between cells with and without gullies (see also reply to previous comment). The procedure of this approach is explained in Methods M3. The results showed that, over a wide range of variable (cf. Table S2), slope steepness, soil type, BUA and road density are indeed the most significant variables in explaining the occurrence of UGs, resulting in an overall accurate model (cf. Fig. S3, Fig. 3b, Fig. 4). Within this model, the fraction of BUA and road density are clearly significant ($p < 0.0001$) and have a positive coefficient (cf. Table S3). This directly implies that higher amounts of buildings and roads strongly increase the predicted likelihood of having an urban gully in a cell.

We chose to work with a logistic regression model in our paper, as this is a widely used approach that is generally believed to result in more robust predictions than other classifier/machine learning models. In addition, logistic regression offers the advantage that its coefficients (cf. Table S3) remain interpretable. Nevertheless, we also briefly explored the outcome using a Random Forest approach. This yielded highly similar results, i.e. BUA and road density are clearly key features in explaining the occurrence of gullies within a 1-km² grid cell.

With respect to the referee's remark about Fig. 4b: this clearly appears a misinterpretation of our results. Fig. 4b shows that the areal extent of gully formation as a process driving gully expansion is limited. As many of these gullies initially form in peri-urban areas (e.g. at the edge of plateaus or at the bottom of hillslopes), they also form in areas with a lower population density and, hence, result in low population displacement. It is the later expansion of these gullies through gully head retreat and sidewall widening (as detected on subsequent images) that cause most of the displacement. As such, the formation of new gullies is only the first detected phase.

However, it is clearly not true that new gullies form only rarely. As indicated on Fig. 4a (more specifically, in the label below the boxplot) we observed the formation of 857 new urban gullies over the study period (i.e. ca. 30% of all mapped UGs). The other UGs were already present on the first available Google Earth images. However, as detailed in Table S1, such imagery is often only available for recent years (i.e. post-2005 and often even post-2010). On these images, urbanization is typically already well advanced. Meanwhile, our analyses of aerial photographs from the 1950s (i.e. pre-urbanization) showed the absence of these gullies. Earlier work (Makanzu Imwangana et al., 2015) further indicates that, in the case of Kinshasa, many gullies formed from the 1980s onwards as a result of urbanization.

In conclusion, while slope steepness and soil characteristic also play a role, it is simply not true that our data would indicate that urbanization is not responsible. The formation of new urban gullies in recently urbanized areas was frequently observed, while the fact that many gullies were already present mainly relates to a lack of suitable imagery. Statistical analyses further clearly confirmed that built-up area and road density are highly significant factors in explaining the presence/absence of gullies at a 1-km² resolution. Furthermore, regardless of the causes and formation dates of these gullies, our results univocally show that the problem of urban gullies is growing (both in terms of displacement and exposure rates). We therefore also believe that our title is apt.

- Some of the statistical inference needs attention. The distributions of observed gully expansion, population density, and estimated displacement are all positively skewed (Fig. 4a), with the median values up to an order of magnitude below the means (diamond symbols). Hence, the means and derived rates (Fig. 4b) poorly reflect, and in this likely case overestimate, the central tendency.

Thank you for checking this. As noted above, we fully rechecked and repeated these analyses. This resulted in slightly updated values due to inaccuracies for some polygons. However, this did not really alter our findings and also here, we are fully confident that our statistical inferences are correct and reliable.

Fig. 4a indeed shows that expansion and displacement rates are positively skewed. This is further discussed in the text. However, the derived rates (cf. Fig. 4b) do not rely on these averages. They represent the cumulative effect (i.e. sum) of all observed gully expansion events and their estimated associated population displacement. The average number of displaced persons per hectare, as shown on Fig. 4b, simply represents the ratio between the total displaced population and the total observed expansion. Given that these values were based on 8,161 observed expansion events of ca. 2900 UGs, across 26 cities and over a period of ca. 20 years, we believe they are clearly representative. Also note that the effect of 'extreme' events on our total displaced population is limited. As discussed in the text, nearly two thirds of all displacement is caused by the numerous events that displaced <100 people. Events displacing >1,000 people account for only ca. 2% of the total.

Moreover, while the medians are certainly meaningful to report (as they are; cf. Fig. 4a), we believe also the averages are highly important; especially because the data are skewed. Suppose, for example, the following fictitious analogy. One hundred landslides occurred in a study area: 90 of them killed nobody, but 10 killed, combined, 1000 people. Hence, the median landslide event would kill nobody. Yet, on average, landslides kill 10 people/event. We think the latter would be clearly relevant to report...

The doubling of mean annual displacement rates (Fig. 4b) in the 2020-2023 interval is curious and needs some more explanation.

Regarding the doubling of the displacement rate since 2020: this is indeed a worrying result, highlighting the need to spend more attention on the issue of urban gullyng.

A further explanation on how these rates were obtained is given below. Yet, in summary, we are highly confident that this result is reliable and representative. More specifically, we compared the average displacement rates (persons/year) across four different time periods (i.e. 2005-2009, 2010-2014, 2015-2019 and 2020-2023), for this we summed the total population of all observed expansion events in these periods

across all cities and UGs and divided them by the duration of their corresponding time period. Given that, also here, these rates were calculated based on many thousands of expansion events of > 2000 UGs across 26 cities, we clearly have sufficient observations to consider these values representative. The last period (2020-2023) is one year shorter (i.e. 4 instead of 5 year). However, this clearly cannot explain the observed increase. Furthermore (as indicated in Table S1), the observed increase cannot be attributed to differences in image availability. More specifically: for all cities, there was already suitable imagery available before 2020.

Moreover, how likely is it that the confidence intervals in Fig. 5 admit negligible changes to the population exposed? Why do you omit these intervals in Fig. 5b? For the potential expansion zone, the upper half of your confidence interval alone is >1.2 million people exposed, and thus close the increase for this zone in Fig. 5b. While this level of uncertainty seems plausible, it should advise against reporting your estimates in an overly spurious way.

We provide a more detailed response below. However, in short, the probability that the exposed population did not increase between 2010 and 2023 is practically zero. As further explained in the Method sections M6 (and earlier confirmed by referee 3), quantifying the exact uncertainties on our estimates is impossible. This because there are insufficient reliable observations on population density to validate the JRC-GHS population data product across different areas and time periods. This is a limitation inherent to Global South contexts and, in fact, the reason why we must use these JRC-GHS estimates in the first place. As such, the uncertainty on our assessments was simply estimated to be $\pm 40\%$ based on comparisons with other products and census data of Bukavu (cf. Methods M6). This approach allows for a realistic yet probably conservative estimate of the uncertainties on the number of displaced and exposed people (i.e. especially at a country-wide scale, uncertainties are likely much lower).

However, this approach does not allow to quantify the uncertainty on relative changes. More specifically, due to a lack of validation data, we have to assume that uncertainties on the population data are the same across all regions and time periods (an approach with which also referee 3 agreed). Knowing the uncertainty on changes in exposure would require knowing the errors on the population data across different years as well as knowing how potential errors are correlated. This is also the reason why Fig. 5b does include uncertainty intervals on the individual causes of exposure increase. As these estimates are based on scenario analyses that compare population densities across different years (cf. Methods M5), uncertainties on these estimates mainly depend on the degree of correlation between potential errors across these different years (e.g. if the population on a certain location is overestimated in 2010 but also in 2023, the error on the relative change would be smaller than on the total amount). As strong positive correlations can indeed be expected (i.e., because JRC-GHS also uses the same correlates to estimate population across different years), uncertainties can be expected to be much smaller (although their exact value remains unknown). This limitation is now also briefly stated in the caption of Fig. 5b.

This reasoning also helps explaining why, despite the considerable estimated uncertainties on total displacement rates, the observed trends between 2010 and 2023 can clearly be expected to be highly significant and of the correct order of magnitude. More specifically, as Fig. 5b illustrates, roughly half of these observed increases in exposure rates are attributable to increases in the areal extent of hazard zones. The latter are directly linked to the observed extent of UGs and therefore subject to very

little uncertainty at this country-wide scale. Hence, in order for the increase to be insignificant, this increase in area should be offset by a decrease in population density in all these hazard zones. This is extremely unlikely. It would not only contradict the actual trends in JRC-GHS data, which (as also indicated on Fig. 4b) indicates that population densities have overall strongly increased between 2010 and 2023. It would also clearly contradict our observations in Google Earth (where an increase in built-up area can clearly be observed) as well as our scientific understanding of ongoing urbanization trends in DRC (e.g. Bédécarrats et al., 2019).

You may also want to explain why you estimate here that many more people have been exposed to newly formed gullies instead of gully expansion: in Fig. 4b you show and argue the opposite, i.e. that many more people were exposed per unit area of gully expansion. Does this not seem contradictory?

Also this seems to rest on a misinterpretation of our results. Fig. 4 indeed shows that few people are *directly displaced* by the formation of new UGs, as their first mapped extents are typically limited and they mostly form in areas where population density is still limited. It is indeed the later expansion (through gully head retreat and sidewall widening) that causes most of the problems. However, Fig. 4 presents an analysis at the scale of individual observed events (i.e. based on the often short time intervals between two satellite images).

Meanwhile, the 'New UGs' in Fig. 5 refers to the increase in exposed population because of all new UGs that formed since 2010 *as well as their subsequent expansion until 2023*. As explained above, we witnessed the formation of >850 new UGs and many of these already expanded drastically since 2010. This explains why the formation of new UGs since 2010 is also responsible for a considerable amount of exposure increase (Fig. 5b), despite the fact that they may not necessarily cause much displacement in their earliest phases (Fig. 4b). To avoid further confusion on this matter, the meaning of 'New UGs' is now better clarified in the caption of Fig. 5.

- The revised discussion is the weakest section by far, mainly because it is still unduly long, verbose, and unnecessarily repetitive in stating the significance of urban gullies, and scaling this up to the entire "(sub-)tropical Global South" (line 249) without much further support by data. Here you missed again the opportunity to compare your study to the findings of similar, previous work on urban gullies. You also mention the potential influence of climate change through altered rainfall characteristics; this mention is odd, given that you discarded rainfall early on in your study (lines 124-125). Your claim that your study "contributes to a better understanding of the lifecycle of UGs" (line 286) is unsupported by the data. You show snapshots of urban gully dynamics but offer little clues as to their lifetime. Much of the remaining discussion revolves in lengthy tones about the basic mechanisms of gully formation and expansion that could have equally well fit into the introduction. Fig. 6 is nice, but not really novel; it might miss important details about rainfall, runoff, slope substrate, and groundwater dynamics.

A detailed reply to the various points raised is provided further below. However, here we already provide a brief summary:

- We agree that there was indeed some needless repetition. Following suggestions provided we have further shortened and reformulated numerous aspects. Nevertheless, we think several of the suggestions provided were more a matter of differences in personal preference than a shortcoming in our discussion.

- We do not ‘scale up’ our results to the entire Global South. We merely point out that, apart from the DRC, also other countries are affected by the problem of UGs. In our understanding, this also helps addressing the referee’s earlier concern that focusing on DRC may cause biases.
- To do so, we refer to several recent studies that report on the occurrence of UGs in other countries (as this is currently the most reliable source of evidence available). However, we do not believe it is necessary or, given the length and format constraints of a Nature paper, possible to provide a systematic comparison of our findings with this literature. For that matter, it is also unclear to us what exactly would be expected from this. As indicated in the introduction and elsewhere in this reply, this is in many ways the first study of its kind (especially with respect to scale). The other studies cited here report on the occurrence of UGs but often have a strongly different emphasis and focus on very local conditions. As such, to our best knowledge, there is not really ‘similar, previous work on urban gullies’ to refer to.
- Regarding the role of climate change: this is further discussed below. However, it is important to emphasize here that we merely state that rainfall characteristics play no important role in explaining spatial susceptibility to UG occurrence. This relates to the overall small range in rainfall conditions across DRC, limitations of the rainfall data available but also to the fact that our analyses focusses on the inter- and intra-city scale. These results do not imply in any way that rainfall patterns as such are unimportant in explaining the formation or expansion of UGs. In other words, while spatial susceptibility patterns appear not strongly linked to rainfall, UG hazards and expansion rates clearly are. As such, the discussion on the potential effects of climate change do not present any contradiction.
- With respect to the ‘lifecycle of UGs’: large parts of this section were revised and shortened as requested. Nonetheless, as further discussed below, our work and data clearly does provide insights into the lifecycle of UGs and their impacts. More specifically, Fig. 6 (currently Fig. S9) provides a general summary of the typical evolution of an UG, based on >8100 observed formation and expansion events over a period of ca. 20 years. To obtain this information, all affected cities were analyzed at 4 to 5 different moments in time (with the exception of Ilebo where available imagery only allowed this 3 times; Cf. Table S1). While these may indeed be considered ‘snapshots’, it remains unclear what else the referee would expect here.
- The discussion on the mechanism of formation and expansion of UGs relates to the discussion of Fig. 6 (now Fig. S9) with a focus on the implications for efforts aiming to stabilize UGs. This was indeed somewhat lengthy and is now shortened. However, we believe this should be discussed here and not in the introduction. First, as discussed above, the introduction does not allow for an extended discussion of these mechanisms. Furthermore, while our results are corroborated by findings of other studies (which we duly cite) the model of the life cycle of an UG (cf. Fig. S9) is clearly the outcome of our analyses and not a review of existing findings. To our knowledge, this is the first time that a clear description of the lifecycle of an UG in relation to its impacts and effects on population exposure is provided. We do so, building on an unprecedented amount of observations.
- Thank you for your appreciation with respect to Fig. 6 (now Fig. S9). While there are indeed several elements that could be added to this figure, we chose to focus here on the elements that we actually studied. More specifically, the

emphasis of this figure is on how UGs displace people, rather than the hydrological conditions that lead to them. As discussed elsewhere in this reply, subsurface flow is expected to have very limited impact on gully expansion dynamics. This because the large majority of UGs form on very sandy and deep substrates with water tables typically far below the bottom of the gullies.

• The thorough revision and rewrite has improved the language and grammar, although many wordy and superfluous phrases have stuck. A number of statements also remain ambiguous, while others feel a bit stogy. Please see my detailed suggestions below: these are not exhaustive, however.

Thank you for your detailed comments. As further detailed below, we have followed many of them. This further improved the readability of our work. However, at several instances, we believe this is more a matter of personal preference than scientific rigor. Also the other two referees raised very little concern about this. However, we are happy to follow further suggestions of the editorial team, where needed.

Line-specific suggestions

18: “Large urban gullies wreak havoc in numerous tropical cities” – Perhaps try toning down overly dramatic statements and instead explain more objectively the adverse impacts of gullies.

The Meriam Webster dictionary defines “Wreak Havoc” as “to cause great damage”. As our result demonstrate, currently over 12,000 people/year lose their house due to urban gullies in the DRC alone. In addition, urban gullies often cause casualties and various other socio-economic impacts. As such, we believe that using this idiom is justified here and an apt way to efficiently introduce the significance of our research topic. As such, unless also the editorial team believes a reformulation is needed, we prefer to keep this phrasing.

19: “Such gullies result from inappropriate urban planning” – This reads as if these gullies cannot form naturally?

As already argued extensively above, we believe we provide ample of evidence that these gullies are indeed caused by urbanization. Nonetheless, this indeed does not imply that urbanization is the only factor causing them. To nuance the statement we rephrased this to “*They mainly result from inappropriate urban planning*”

20: “sensitive to soil erosion as well as insufficient infrastructure” – Ambiguous phrase. Do you mean that “gullies are sensitive to ... insufficient infrastructure”?

This was indeed a bit ambiguously formulated. The insufficient structure is one of the causes, in addition to lack of urban planning and the fact that these environments are already naturally sensitive to erosion. We clarified this by restructuring the sentence.

22: “they can drastically expand” – See first comment; consider quoting rates of gully expansion instead.

As the referee will have noted, all quantified expansion rates are reported further in the text, figures and supplementary materials. Please note that the statement made here was an introductory sentence of the Abstract and did not yet refer to the results

of this study. Given the fact that a Nature abstract is expected to be < 200 words we simply shortened this sentence, avoiding the use of 'drastic expansion'.

24: Why are gullies a "new type of geo-hydrological hazard" if so much literature exists about them already?

Gullies are in general studied under the lens of soil erosion/land degradation processes that occur in rural environments. In addition, they are generally not studied as hazard. As we extensively argue in the main text, urban gullies can be considered a new type of geo-hydrological hazard as they : (i) are tightly linked to the ongoing process of urbanization in the tropical Global South; (ii) their occurrence and impacts are rapidly increasing; (iii) they are in many ways different from other, much better studied geo-hydrological hazards like floods and landslides; and (iv) they are currently largely neglected in disaster risk reduction strategies.

Furthermore, while the number of studies reporting on urban gullies is increasing, this number remains very limited (especially when compared to gully erosion in general and/or other hazards like landslides or floods). In addition, their potential role as a widespread hazard received hitherto almost no research attention (see also reply above). As such, we believe the formulation '*new type of geo-hydrological hazard*' is appropriate here. To further back up this statement (i.e. the lack of quantitative research on the impacts of such gullies), we now also cite the recent review by Kuhn et al. (2023) which further highlighted the lack of impact assessments of (urban) gullies.

31: "drastically" – See comment on line 22.

Drastically was removed.

32: The comparison between pre-2020 and post-2020 rates might be lopsided; I assume you only have four data points for annual rates since 2020.

We think it's not and believe this is adequately demonstrated in the text itself. More specifically, the reported post-2020 rate refers to the average of the past 4 years (2020, 2021, 2022, 2023) as derived from several thousands of expansion events. Admittedly, four years is shorter than the pre-2020 period. However, as the inset of figure 4b shows, this most recent rate is clearly much higher than all three of the preceding 5-years averages that we could calculate (i.e. for 2005-2009, 2010-2014 and 2015-2019 the average displacement rate was each time below 5000 persons/year).

As such, we see no reason to assume this comparison is lopsided. The only small limitation we see is that the last period is one year shorter. However, this as such cannot explain the observed doubling in displacement rate.

33: "potential expansion zone of urban gullies" – This interesting concept might need a brief definition here.

As the referee will have noticed, this concept is further explained in detail in the main text. As the abstract can only be up to 200 words, and we believe this concept is already sufficiently clear to obtain an intuitive understanding of our finding, we don't think that further clarifications are required within the abstract.

34: "This number doubled since 2010" is in conflict with "Over 3 million people currently live" in line 32.

We don't see the conflict/problem here, as these numbers are directly evident from our results (cf. Fig. 5). However, to avoid potential confusion, the statement was reformulated as follows: “Between 2010 and 2023, the number of people living in the potential expansion zone of urban gullies doubled from 1.6 to 3.2 million.”

35: How would “climate change” influence the number of people exposed?

See further down in the discussion: many studies on gullies in other contexts indicate that both gully formation and expansion rates are strongly driven by rainfall intensities. As the latter are expected to increase as a result of climate change, this may result in a further increase in the number of urban gullies as well as their potential size. In turn, this would likely result in larger hazard zones and, by extent, an increase in exposed population. This increase would be in addition to the exposure increase linked to urban growth.

40: Overuse of “drastically”; please provide some objective rates instead.

To fit the text within the allowed length limits, this sentence was removed. As such, the word ‘drastically’ is no longer used here. Since the rapid growth of urbanized areas is a well-established fact (with the cited source providing detailed quantifications) and further quantifying these growth rates is not really relevant to understanding our problem statement, we believe it is not required to provide further details here.

44: “other problems and risks” – Such as? How would urban gullies compare to these other problems and risks?

As the references cited (and many more studies) discuss, the ongoing process of urban expansion may in some cases lead to increased risks to hazards like flooding and landsliding, as well as potential other disasters. In addition, the rapid growth of cities in the Global South can lead to numerous issues with respect to trafficability, food-accessibility, social segregation and other socio-economic challenges. Evidently, the type and scale of these problems and risks are highly context depended. It is outside the scope of this work to fully discuss this, while also the allowed length of the introduction does not permit this. Hence, we chose to not further elaborate on this in the text. Nevertheless, some key references to work discussing these issues are provided.

48: “Democratic Republic of the Congo (DRC) is particularly affected” – This reads like some availability bias. Or are there any physical reasons why this particular nation is so prone to urban gullies?

To avoid confusion or doubt, we changed “is” to “appear”. Nonetheless, we believe that we already provided a detailed reply to this question in our previous rebuttal.

In short, systematic analyses on the occurrence of urban gullies at larger (e.g. continental scale) are simply lacking. We believe this is one of the key innovative aspects of this study. While some studies (of which numerous cited in the text) indicate the occurrence of urban gullies in other countries as well, the available literature indeed indicates that Congo is likely the most affected country in Africa (with especially Kinshasa being severely affected; cf. Makanzu Imwangana et al., 2014; 2015). There are also good reasons physical reasons to support this idea. Research so far indicates that urban gullies mainly occur in cities with a tropical

climate, some topography and/or sandy soils. Our results (which present to our knowledge the first systematic analysis of factors controlling urban gully occurrence at a larger scales) further corroborate this (e.g. Fig 2; Fig 3). Given its size, its position at the equator and the presence of numerous cities on the sandy Kwango-Kwilu and Kasai Plateaus (cf. Fig. 2), DRC is likely the African country with the highest number of urban centers that meet these criteria. Given that many urban gullies also form due to a lack of urban planning and/or appropriate rainwater retention/evacuation infrastructure, the rampant poverty and lack of well-functioning governmental structures in DRC may further aggravate this problem.

Furthermore, we fail to see how our choice to study the problem of urban gullies at the scale of DRC (a territory the size of > 50% of the European Union for example) would cause a bias. Our goal is to demonstrate that, within this country, UGs lead to significant impacts to the population and that several million people are exposed to them. It could be that other countries in Africa have similar levels of impact/exposure. Further research will be required to confirm this. However, this would merely underline the main message of our work: urban gully erosion is a growing and hitherto largely neglected hazard.

52: “typical erosion features of the Anthropocene” - Typical of what specifically?

This was indeed somewhat vaguely formulated. Because of this and the need to reduce the length of the text, this sentence was left out.

54: “exceeds the local resistance of the topsoil against incision” – What about groundwater flows and erosion? Or do urban gullies form exclusively by surface erosion?

Interesting question. The short answer is: yes, most likely incision by surface runoff is nearly always the dominant mechanism leading to the formation of the urban gullies observed. One of the key reasons for this relate to underlying pedological conditions. As our results show, most of the affected cities are located on sandy plateaus and typically have very sandy soils that are often several tens of meters thick (see Fig. 2 and cited references). As such, most of the water that infiltrates most likely percolates vertically to depths that far exceed the depths of most gullies. Also in cities located on other substrates, gully formation due to subsurface flow was never really observed (by us or by other studies we are aware of). In these non-sandy contexts, it may be that the process of gully expansion is to some extent influenced by subsurface flow. For example, our earlier field based research (Ilombe Mawe et al., 2024) indicates that the expansion of urban gullies in Bukavu is also linked to landslide activity. Nonetheless, the mechanisms behind this remain poorly understood and it is unclear what is cause and what is consequence here. However, this finding does not contradict the fact that the initial formation of these UGs is triggered by accumulated surface runoff.

As such, we think this formulation is entirely correct. While an interesting point for discussion and research, length limitations do not allow us to further elaborate on this in the text.

56: “urbanization of these environments” – Please be more specific. Do you mean landscapes prone to gullies or landscapes that already have gullies?

Indeed, this was somewhat unclear and is now reformulated. As also discussed above, the systematic analyses of historic aerial photographs clearly confirmed that almost no gullies were already present before the conversion to urban area took place. In addition, the few gullies that could be detected appear already linked to urbanization (cf. Fig. S1). Also other recent work (e.g. De Geeter et al., 2023) clearly confirmed that gully densities are overall very low in non-urban areas of Congo (mainly due to the significant vegetation cover).

57: “increases in roofs” → “increases in roof-top area”?

Indeed. This was corrected.

61: “leading to local increases in contributing area” – Really? How does this work? Is the contributing catchment area not set by drainage divides?

Yes, this is reported by several studies (cited in the text), including Makanzu Imwangana et al. (2014) who observed drastic differences between what can be considered the original, topographic contributing area and the actual contributing area as modified by the presence of roads. This finding was later confirmed by our own field-based research in Kinshasa & Bukavu (Ilombe Mawe et al., 2024). Several factors contribute to these large changes, including the presence of ditches or other water evacuation infrastructure next to some roads. Many other, mainly unpaved, roads lack such infrastructure, but may ‘sink’ over time as a result of erosion linked to rainfall and their frequent use. As they sink, they can become more effective in capturing runoff. These effects, can be further reinforced by the fact that large parts of the urbanized areas that potentially accumulate runoff are located on relatively flat plateaus or hilltops (with urban gullies typically starting at the bottom of their somewhat steeper edge; cf. Fig. S9). As such, small changes in topography as induced by roads can indeed greatly influence the topographic area.

Evidently, increases in contributing area at one location imply that other locations experience will experience a reduction in contributing area. Yet, it is typically at the locations where there presence of roads led to an increase in contributing area that UGs form. The work of Carvalho Junior et al. (2010) provides a clear demonstration of this. They present a case study for a part of Gama City (Brazil) where changes in contributing areas were calculated based on detailed digital elevation models before and after urbanization. For the referee’s convenience, we pasted some of their key findings below. While some locations were characterized by a decrease in contributing areas, all urban gullies occurred at locations with an increase in contributing area. Also here, these increases are clearly linked to the presence of roads.

[REDACTED]

[REDACTED]

(Source: Junior, O. C., Guimaraes, R., Freitas, L., Gomes-Loebmann, D., Gomes, R. A., Martins, E., & Montgomery, D. R. (2010). Urbanization impacts upon catchment hydrology and gully development using multi-temporal digital elevation data analysis. *Earth Surface Processes and Landforms*, 35(5), 611-617.)

The influence of roads on contributing areas is of course highly context dependent (e.g. this will depend on the layout of the road network, the original topography and/or specific road characteristics like their elevation and the presence/absence of ditches or sewage systems). As discussed in the context of our susceptibility model, this also makes it currently impossible to predict or even quantify these effects at high resolution across the DRC. Nonetheless, our analyses at coarser scale show that 1-km² grid cells with more roads are also more likely to have urban gullies (cf. Fig. 3). This correlation seems to corroborate the mechanisms discussed above.

As such, this formulation appears entirely correct to us. Unfortunately, the length restrictions of the text do not allow us to elaborate further on this in the text. Nevertheless, the citations provided can guide interested readers to further information on these aspects.

68: Somehow I have the feeling that the previous summary does insufficient justice to the “heterogeneous, complex and threshold-dependent conditions under which UGs form”. Which thresholds do you mean?

We believe ‘heterogeneous and complex’ are sufficiently justified based on the elements provided above. ‘Threshold-dependent’ refers to the well-established fact that gullies typically only initiate when the shear stress of concentrated water exceeds the local resistance of the topsoil against incision (cf. start of the paragraph). As the former mainly depends on the amount of Hortonian runoff that can accumulate, the initiation of most gullies (and for that matter also urban gullies) only starts when certain topographic, land cover and rainfall conditions are exceeded (see also e.g., the work of Torri & Poesen, 2014, *Earth-Science Reviews* for an extensive review on the threshold-dependent nature of gully erosion as well as Makanzu Imwangana et al., 2014, *Geomorphology* for a discussion of this concept applied to urban gullies).

Unfortunately, length limitations do not allow us to further elaborate on this concept in the text. As this threshold concept is also not really necessary to understand the

rationale of our work or its results, we therefore removed the term ‘threshold-dependent’ from our text to avoid potential confusion.

78: “many gullies continue” – Including urban ones?

Yes indeed. We changed gullies to urban gullies to avoid confusion.

81: “erratic nature of expansion events” – This reads as we have no clue as the drivers of gully expansion.

Erratic is typically defined as “not even or regular in pattern or movement; unpredictable”. This is indeed the case for urban gullies. During our mapping efforts and field visits, we often observed how urban gullies that remained stable for longer periods (sometimes years) expanded with several hundreds of square meters over the course of a single rainfall event (see also photos and videos provided above).

This statement does not imply that we have no clue on the drivers of gully expansion. Nevertheless, as indicated in the text, such events remain extremely difficult to predict. One element strongly contributing to this is the lack of reliable rainfall data (and predictions) and detailed digital elevation models for most affected cities in the DRC. In addition, as our earlier work show (Ilombe Mawe et al., 2024; cited in the text), many aspects of urban gully expansion do indeed remain poorly understood. As such, we believe this formulation is correct.

81: “enormous size” – How do you define “enormous”?

Size criteria are discussed in the methods (with all our mapped UGs having a length of at least 30m). We also provide detailed information on the dimensions of these gullies (with lengths that easily exceed several hundreds of meters and widths and depths that can exceed tens of meters) further in the text and the supplementary information. Figure 1 also already gives a good impression of the typical size of these urban gullies. In addition, while typically very large, the exact impacts and UGs depend on numerous other factors (as indicated in the text). As such, we believe it is not meaningful to provide a definition of ‘enormous’ here in the introduction.

84: “UGs often form in poor, informal (peri-)urban areas” – Are there exceptions? Urban gullies must form in urban areas by definition. It would be useful to have a comparison with gullies forming in rural areas to appraise any differences or similarities in size, formation rate, etc.

Our mapping and field-visits clearly indicate that the majority of UGs indeed occur in relative poor and recent neighborhoods (i.e. typically less than 10 years old) that grew in an unplanned and informal way around the already existing city. Fig. S8 illustrates this for Kinshasa. There are exceptions to this, in the sense that UGs can also form in already older, well-established urban areas. However, we observed very few gullies in the rural areas surrounding the cities.

The latter is further confirmed by a more systematic analysis, presented later in the text (see also our reply to some of the general comments above). Following the referee’s earlier request, we checked 715 grid cells of 1 km² in the rural areas surrounding the investigated cities. Within these cells, we only detected two gullies (cf. Methods M3). This indicates that gullies occur only rarely in rural contexts within the DRC (i.e., our mapping in rural areas suggests a density of less than 0.01

gullies/km², while cities affected by urban gullies have an average density of ca. 1.7 gullies/km²).

This finding is further corroborated by the findings of De Geeter et al. (2023; *Environmental Research*), who constructed a gully susceptibility model for Africa, based on the detailed mapping of gullies for over 1,200 sites across the continent. Their results show that most of the DRC are characterized by (very) low susceptibility values (see below). In steeper agricultural land, gullies can occur. However, observed densities are clearly much lower than the density of urban gullies we observe in the affected cities.

[REDACTED]

(Source: De Geeter, S., Verstraeten, G., Poesen, J., Campforts, B., & Vanmaercke, M. (2023). A data driven gully head susceptibility map of Africa at 30 m resolution. *Environmental Research*, 224, 115573.)

91: “understanding of the magnitude and drivers of the risks associated with UGs is a critical step towards disaster risk reduction” – This really depends on whether urban gullies pose a substantially higher risk than other disaster sources.

We strongly disagree with this statement. As we elaborately argue in our text, urban gullies form a hitherto largely neglected hazard that may lead to disastrous consequences. These impacts occur *in addition to* those of other, more frequently considered hazards. Only considering the hazard with the highest risk would clearly lead to substantial underestimations of the total risks. For example, it is not because floods often lead to most impacts that we should not consider other hazards like landslides. This is especially so because each of these hazards has its specific geography and triggering mechanism. Moreover, our results clearly show that urban gullies are indeed associated with very substantial risks.

101: “Verification” → “validation”? Checking air photos does not warrant that you eliminated all sources of error.

The term ‘verification’ was indeed perhaps not fully suitable as the goal of this checking was just to evaluate whether the detected gullies were already present before most urbanization. Hence, also ‘validation’ would not be really suitable. We therefore changed verification to “cross-checking”.

As with any analysis or mapping step, this process may indeed be subject to some errors. However, given the very high resolution of the aerial photographs (similar to the resolution of the satellite images available in Google Earth) in combination with the typical very large size of the gullies (+ the fact that we already knew the potential location of these gullies through our mapping in Google Earth), these errors are most probably minimal and certainly not of a scale that it would invalidate this finding.

103: “overwhelming majority of these gullies are indeed linked to urban sprawl and road construction” – Based on which criteria? How can the remote sensing data tell you this?

As indicated in the text, the procedure is further clarified in the Method section (M1). We slightly expanded this explanation to further clarify the procedure:

To verify whether the detected UGs are linked to urban growth, we checked high-resolution panchromatic aerial photographs of each city, taken in the 1950s. These photographs are conserved at the Royal Museum for Central Africa in Belgium⁵³. If identified gullies were already present, we examined if they were inside or outside built-up areas and, when outside, whether they were linked to the road network (i.e., the gully formed along a road or lies in its direct extension, within a distance of ca. 100m or less). Gullies observed in the 1950s that were located outside built-up areas and not related to the road network were assumed to be of natural origin. They were not retained for further analysis.

Fig. S1 further clarifies this by providing some typical illustrations of cases where detected UGs were not yet present as well as cases where they were, but most likely already linked to the ongoing urbanization (as they are located within or in the direct vicinity of buildings & the road network).

Overall, we believe this straightforward and robust analysis clearly confirms that the overwhelming majority (i.e. >98%) of the urban gullies detected on Google Earth imagery formed after the 1950s. Furthermore: (i) the remaining 46 gullies seem already linked to ongoing urbanization; (ii) almost no other gullies were detected on these images; (iii) nearly all detected gullies in recent times are located in urbanized areas and/or the direct vicinity of roads; (iv) additional mapping of 715 km² in rural areas surrounding affected cities clearly indicates that gully densities in rural contexts are very low; (v) earlier susceptibility assessments (e.g. De Geeter et al., 2023) confirm this; (vi) our susceptibility analyses clearly confirm that the extent of built-up area and road density are clearly significant variables explaining the presence or absence of detected UGs at a 1-km² resolution and (vii) numerous earlier studies already demonstrated how urbanization may lead to the formation of urban gullies (while this can also be clearly observed in the field). As such, we believe it stands beyond reasonable doubt that the UGs observed and analyzed in this study are indeed directly caused by urbanization.

104: “fact” → “observation”; “deforestation was already well advanced” – This is the first time you mention deforestation. I would have expected to see this mentioned earlier in your

review of the current knowledge on urban gullies. The second part of the sentence is difficult to appreciate without any time stamp on forest losses.

Indeed, ‘fact’ was changed to ‘observation’. We now also briefly mention in the introduction that the removal of vegetation may also be a factor that contributes to the formation of UGs (in addition to increases in rooftop areas, road construction and lack of rainwater retention/evacuation infrastructure).

As the referee will understand, the aerial photos from the 1950s studied here provide the earliest available source of information on land use and forest extent. Hence, producing a timestamp for deforestation before these photos is impossible. Nonetheless, as is also illustrated in Fig. S1, large patches of our study area (i.e. the current spatial extent of Congolese cities that are now affected by UGs) were already deforested on these images. As such, the statement that “deforestation was already well advanced” seems correct to us.

106: “clearly connected to roads and/or had a significant amount of buildings in their direct vicinity” – How does this confirm that roads or buildings were responsible for gully formation?

We believe this is sufficiently explained in our introduction, where we summarize how earlier work already indicated the mechanisms of how roads and increases in built-up area lead to the formation of urban gullies (see also reply to comments above). To summarize again:

- **Roads, buildings and other infrastructure increase runoff production (by limiting infiltration)**
- **Roads greatly help in concentrating this runoff**
- **Roads can also greatly alter (and thus at some places increase) the contributing area; leading to larger runoff volumes.**

Given that the available imagery (i.e. Google Earth imagery + aerial photos from the 1950s) indicate that gullies occur only very rarely in the absence of buildings/roads and earlier work further confirmed this, the construction of roads & buildings clearly seems the most plausible cause for gully formation.

109: “typically very large” – Style; how large?

“Typically very large” was replaced by “predominantly large”. How large, is further indicated in the second part of the sentence, while Fig. S2 (to which we clearly refer in the text) gives the size distributions of all gullies mapped.

110: Add country to “Kinshasa” for readers unfamiliar with the geography of Africa.

We believe that, at this stage in the text, it should be sufficiently clear that our work was conducted in the Democratic Republic of the Congo and that all cities mentioned are located within this country (see e.g. Fig. 2, which provides the location of all cities)

112: “UGs typically occur in areas that are somewhat steeper and sandier” – Avoid “typically” unless you state exceptions as well. What do you mean by “somewhat”? Is this difference in steepness and sand content relevant or not?

We believe this is clearly illustrated in Fig. 3a, to which we refer here. Grid cells with an urban gully (or in a few cases a gully in rural context; cf. Methods M3) have in 50%

of the cases a slope between 2.5° and 5°, while for cells (inside or outside cities) without gullies, this is ca. 1.1° to 3.3°. As the figure shows, the boxplots do show some overlap. However, as also indicated, the difference between both distributions is clearly highly significant ($p < 0.0001$ according to two-sided Mann-Whitney U test). The same applies for sand content (see also Fig. 3a): cells with (urban) gullies have a median sand content slightly above 60%. For non-gullied cells the sand content is overall 5% lower. Yet the boxplots show overlap and some gullies were observed in cells with sand contents below 40%. The latter is also clear from Fig 2: while most affected cities are located on sandy plateaus, there are exceptions to this; especially in the East of the country which is overall steeper. Nevertheless, the difference in distribution with respect to sand content is also highly significant (cf. Fig. 3a).

116: “Other key factors explaining the contrast between areas with and without UGs are the fraction of built-up area and the road density” – How many possible factors did you look into?

As indicated in the text, further details on the analyses are provided in the Methods, section M3. As we explain there, a full list of all considered variables and their original data sources is provided in Table S2. In total, we considered twelve variables that describe the topography, soil characteristics, rainfall conditions, tree cover, built-up area and road density in every cell.

119: “98% of all mapped UGs were clearly connected to the local road network” – Does this high percentage not simply reflect a dense road network in urban areas? One could similarly argue that most houses are connected to the local road network.

Evidently, road densities are high in these urbanized areas, while also houses are indeed ‘connected’ to the road network. However, as already extensively discussed above, the role of roads in the formation of urban gullies is already well-established in literature. Moreover, as we clarify in the next sentence, this being ‘connected’ is more than occurring in each other’s vicinity. Based on detailed image observations and taking into account the topography, we observed that 48% of all urban gullies occurred along existing road networks (i.e. they start at a point along the road and subsequently destroy it; see e.g. Fig 1 as well as the photos and videos provided above). The other 50% directly receive runoff from roads. As explained above, also the presence of buildings may contribute to the formation of urban gullies. However, as clearly shown by our results, both factors are significant in explaining the occurrence of urban gullies (in an univariate analyses as well as a multi-variate analysis accounting for potential intercorrelations).

121: “well explain” – How well? Please provide some objective measure.

This is clearly indicated in the figures to which we refer at the end of the sentence, while the evaluation procedure is explained in the method section M3. More specifically, Fig. 3b shows for Kinshasa that our model which includes the four listed variables (i.e., the slope steepness, soil type, the fraction of built-up area and the road density of a 1-km² cell) predicts overall very high susceptibility values in cells where an urban gully was indeed detected and much lower in cells where this was not the case. Especially the non-urbanized fringe around the city (where building and road densities are much lower) are characterized by very low susceptibility values. As explained earlier, no gullies were indeed detected in these zones. The spatial patterns of predicted susceptibility and their comparison with actually mapped urban gullies is shown in Fig. S4 for all other affected cities. Meanwhile, Fig. S3 presents the Receiver

Operating Characteristic Curve for ten Monte Carlo Cross Validations (cf. Methods M3). Overall, ROC curves and their corresponding Area Under the Curve (AUC) are the most commonly used metric to evaluate models that simulate the presence/absence of features (in our case UGs). In our case, the AUC value is 0.81, which indicates that model performance is clearly reliable. Fig. S3 further shows fairly little variations between different validations ROC curves. This clearly indicates that the model is also robust (i.e. independent of a few specific/exceptional points in the training or test data).

124: “rainfall characteristics did not further improve our model” – Why not, and which model do you mean?

As explicitly detailed in Methods M3, and following the referee’s earlier suggested, we conducted an extensive mapping of areas that are not affected by urban gullies, both inside the affected cities as well as the rural areas surrounding the city limits. This, and our dataset of mapped UGs were converted to cells of 1-km² resolution that either contained at least one (urban) gully (1) or not (0). For each of these cells we extracted a diverse range of variables, describing the environmental conditions of these cells in terms of topography, tree cover, soil characteristics, rainfall conditions, built-up area and road density (cf. Table S2). We used these variables to fit a logistic regression model that can simulate the likelihood that an UG occurs in a cell, based on the most significant variables listed above. As also discussed in the previous comment, this model performs well. It is by no means perfect, which is easily understandable due to the numerous other factors & complexity that may influence the formation of UGs (e.g. the presence of water retention structures, the complex role of roads & their effect on local topography; see above) and the lack of detailed data allowing to characterize these elements. Nonetheless, this model clearly confirms the results already evident from the univariate analyses (cf. Fig. 3a): even after accounting for spatial correlations between these factors, the occurrence of UGs is mainly related to steeper hillslopes, sandy soils, elevated fractions of built-up area and elevated road densities.

The fact that rainfall characteristics did not increase the accuracy of our model should not come as a surprise in this context. First, as shortly indicated further down in the text, all cities in the DRC are characterized by a tropical climate (i.e. A-climate according to the Köppen-Geiger classification), meaning that they all experience significant amounts of, often intensive, rainfall. In addition, accurate/detailed rain gauge data is largely missing for this part of the world. As such, the only rainfall estimates available rely on satellite observations and re-analysis products at course resolution (0.1°). This induces uncertainties, but also does not allow to clearly differentiate between 1km-grid cells that are (1) or are not (0) affected by urban gullies. More specifically, many of these 0-cells occur in the same cities (and hence the same 0.1° gridcell of rainfall data) as the cells with an urban gully.

This is further illustrated in the boxplots below, where we show the distribution in different potentially relevant rainfall proxies (see Table S2 for further details) for (i) a randomly selected set of 50,000 points across Africa; (ii) the selection of these random points located in the DRC; (iii) all cells without a gully used to train our model; and (iv) all cells with a gully. As can clearly be seen on these boxplots, cells with a gully tend to have rainfall values that are slightly higher than cells without gullies. However, the effect is minimal and ranges are overall very limited; certainly when compared to the overall range of rainfall conditions in the DRC and Africa. Our

multivariate analyses further showed that these differences become even less significant once you account for topography (i.e. slope steepness). Only for the maximum registered daily rainfall data, cells with a gully tend to show a clearly larger range (potentially corroborating the idea of a high sensitivity to e.g. climate change). However, this rainfall proxy is also highly susceptible to potential errors and uncertainties, while also adding this variable did not significantly improve our model.

Overall, this gives a logical explanation why we could not detect a significant effect of rainfall in the spatial occurrence of UGs within and across affected cities of DRC. It is worthwhile noting that, also at the continental scale of Africa (where the range in rainfall conditions is obviously much larger), the same rainfall proxies as considered here have overall a limited influence on the spatial patterns of gully susceptibility (see, e.g., the earlier cited study of De Geeter et al., 2023, where this issue is further discussed).

Nevertheless, as indicated further in the text, this clearly does not imply that rainfall is not important for urban gully hazards. While data limitations force us to only consider long-term average spatial patterns of rainfall, it is of course the occurrence of specific rainfall events that determine the specific timing and frequency with which UGs may form. Likewise, while overall susceptibility patterns may not be strongly influenced by

rain patterns, their further expansion rates clearly are (e.g. Vanmaercke et al., 2016; Ilombe Mawe et al., 2024). This indicates that hazards related to UGs may be sensitive and likely further aggravated by climate change. However, this still involves large uncertainties (including around the predicted changes in rainfall intensity). It is outside the scope of this paper to fully address this.

125: “overall rather limited range in rainfall conditions across DRC” sounds unduly generic and hardly informative enough to discard rainfall.

See reply above. Due to space constraints and because we deem this issue is not essential for our further results and conclusions, we chose not to elaborate much further on this. However, as indicated above, also the limited resolution of the rainfall products available may contribute to this lack of improvement. This is now further clarified in the text.

127/Fig. 2: “showing all cities investigated” – How did you select these cities? Inset bar diagram might be difficult to read; why not simply show % of urban area affected by gullies?

This is elaborately explained in Methods M1, to which we referred in the caption. For the referee’s convenience, we cite the relevant sentences below:

We first identified all cities in DRC that were significantly affected by UGs. For this, we checked all urban centers that were assigned the official status of “city” by presidential decree (Articles 53 to 55 of Decree-Law 081 of July 2, 1998) as well as other urban centers with at least 80,000 inhabitants in 2020 (according to OCHA, 2022) that show characteristics of small cities. This list can be considered exhaustive. The presence of UGs was checked in all these cities using available Google Earth imagery.

While somewhat dense in information, we believe the inset of Fig. 2 is clearly readable. Exact numbers may be hard to derive from this graph. However, as indicated in the text, all data and values with respect to the occurrence of UGs at city level are listed in Table S1. This table includes the % of urban area as affected by gullies (see last column). It would indeed have been an option to show this instead of the total area of the city extent & gullies. However, we believe this may lead to confusing interpretations. For example, some cities have a very high fraction but are relative small. Other cities are severely affected by UGs (e.g. Kinshasa), but are also very large, leading to relatively lower fractions. As we aim to focus on the total impacts and exposure rates caused by these UGs, we believe reporting the absolute values (in combination with the extent of the city) is the most correct way to represent these data.

135/Fig. 3: “showing the distribution in average slope steepness, sand content, built-up area and road density” – Why not show the raw distributions instead of smoothing these properties? You may want to normalize these distributions by those of the urban/non-urban areas to obtain a relative distribution or frequency ratio. Curiously, built-up area (%) shows the least conclusive distinction between areas with urban gullies and those without (Fig. 3a).

As the n-values mentioned under each boxplots indicate, these boxplots are based on very large numbers of observations (i.e. 754 cells with (urban) gullies, 1467 cells without gullies). As such, it is very hard to actually show the ‘raw distribution’ in a clear and sufficiently concise way. In addition, boxplots are a very common and broadly accepted way to visualize data distributions. They do not ‘smoothen’ properties, but summarize the key characteristics, including the 25%, 50% and 75%

quantile, whiskers (defined as the 1x times the Q25-Q75 distance) as well as potential outliers. Please note that the source data with all actual values will also be provided.

We believe that also normalizing these distribution would be an unclear (and in fact incorrect) way to represent the data. It would also prevent readers from appreciating the actual values and units. In addition, the relevant comparisons can be directly made, given that the corresponding distribution of non-gullied areas is plotted directly next to the gullied ones. One could indeed look to the frequency ratios (and a visual comparison of the boxplots already allows this). However, we believe the two-sided Mann-Whitney U tests we conducted (confirming that all distributions shown are indeed highly different between cells with and without (U)G; cf. Methods M3) provide a more robust indication here. In addition, odd-ratios are known to be highly sensitive to intercorrelations. This is why we chose to work with a logistic regression approach instead (see also previous comments), as this allows to account more robustly for the combined effect of different variables.

Regarding the remark that built-up area would be “least conclusive”, this is clearly a misinterpretation. It is true that the interquartile ranges of both boxes are rather large and show much overlap. However, this is to be expected. For cells with no gullies, we randomly sampled both inside and outside the city extent (leading, of course, to a wide spread). Similarly, many cells with gullies have a built-up area close to 100%. However, cells at the urban fringe can also have an UG, despite lower built-up areas. As already discussed above, these are often cells at the edges of urbanizing plateaus or hillslopes that receive runoff from neighboring cells upslope. Despite this spread, the difference is clearly conclusive. Cells with an (urban) gully have a median fraction of built-up area of >70%. For the non-gullied cells, this is 0%. Due to the skewed nature of the data, the averages are closer to each other, but still show a difference of ca. 30%. A two-sided Mann-Whitney U test further confirmed that the differences are highly significant ($p < 0.0001$). Also after controlling for potential autocorrelations with other relevant factors, this difference remains significant. This is, for example, evident from Table S3, showing that Built-up area has a highly significant positive effect on the susceptibility to UGs, even when the effects of slope, soil type and road network are already accounted for.

139: “significant difference in distribution according to a two-sided Mann-Whitney U test” – This test concerns the averages of the samples only, right?

No. The two-sided Mann-Whitney U test (also known as Wilcoxon rank-sum test) is a non-parametric test to check whether two populations have a similar distribution. This well-known test was used as not all potential predictor variables considered (cf. Table S2) are normally distributed. As such, this test allows for a more robust comparison (e.g. potential outliers have lower impacts) and mainly tests whether the medians are different.

145: “experienced clear (> 10 m²) spatial expansion” – I assume that “clear” refers to “detectable” here? Consider using relative expansion rates instead, as it may be more difficult to detect small expansions for large gullies?

Indeed. ‘clear’ was reformulated to ‘detectable’. At an earlier stage, we have indeed considered using a threshold based on the relative expansion. However, since the focus of this mapping effort was to quantify the total rate of expansion and displaced population, quantifying the expansion in absolute terms was seemed a more logical

choice to us. Moreover, relative thresholds are more difficult to interpret as they also depends on the measuring period (i.e. the time between two satellite images) and are more sensitive to error propagation. In addition, we observed that the larger gullies are typically also the more active ones.

146: “UGs showed important expansion over this period, accounting for 70% of the total observed expansion” – Please clarify: 70% of the absolute increase in total urban gully area happened in these five cities? If so, please add how much of the total gully area these cities had originally in 2004.

Indeed. Space restrictions do not allow us to elaborate on this here. However, as the referee will have noticed, we refer to Table S4 where all observed expansion rates are reported per city. Table S1, reports the extent of UGs in each city on the reference image used. From this, the original extent can be easily derived for readers that are interested to know this specific information. More specifically, the total extent of urban gullies in 2004 across all cities in the DRC was 629.71 ha. The total extent of UGs in these five most affected was ca. 443.96 ha (or 70.5%). Hence, the contribution of these five cities in terms of total observed gully expansion matches well with the observed fraction of the UG extent at the start of our observation period.

We further note that the entire dataset of all mapped UGs across all cities and time periods will be made publicly available with the publication (and is already accessible to referees through the confidential link provided). As such, readers will be able to check the exact extent and location of all UGs as they see fit.

150: “we estimate that ca. 142,400 people were directly displaced” – Do you have any independent data to validate your estimates?

As the referee will understand, the answer here is no. As we explain in the introduction and at many instances in this and our earlier reply, this is the first assessment on the occurrence, impacts and exposure of urban gullies at larger scales. Nonetheless, we can be fairly certain that our estimate (which was slightly revised; see comment at the beginning of this reply) is clearly in the right ballpark. The presence and expansion of UGs can be clearly detected and accurately mapped on the images available in Google Earth. As we explain further down in the text, also the actual destruction of houses can clearly be detected on these images and was also regularly detected in the field (e.g. Fig. 1).

Mapping the exact extent of the UGs can induce some uncertainties (e.g. due to local photo interpretation difficulties). However, such potential errors can clearly be expected to be negligible for this estimate at the scale of DRC. More specifically, this estimate reflects the sum of 8,161 observed expansion events of individual ca. 2,900 urban gullies across 26 cities. With respect to mapping, we believe that the biggest source of uncertainty is the possibility that some UGs remained still undetected (leading to an underestimation). However, as we revisited all our mapping efforts during the previous revision of the manuscript, we clearly expect that our dataset used is comprehensive.

Another, more important source of uncertainty affecting this estimate is evidently the gridded population dataset used. However, as discussed in detail in the Methods Section M6, we used the best product available. Likewise, to assess the degree of uncertainty, we repeated our analyses with alternative population datasets (resulting in the uncertainty margin around our estimate, shown in Fig. 4b). As discussed in

Methods M6, comparisons with detailed census data (which are unfortunately only available for Bukavu) suggests that our assessment may still be an underestimation.

Hence, we are overall confident that this total reported number of displaced persons provides a robust and likely still conservative estimate.

152: Is the “average of ca. 7,120 persons.year⁻¹” sufficiently representative, i.e. do the five most affected cities have comparable population densities?

We believe sentence may have been misinterpreted by the referee. This reported average rate represents the average annual displacement across all cities; not just the five most affected ones. This is now better clarified in the text. Moreover, values per city can be consulted in Table S3.

152: “this process clearly accelerated” – How do you know it is just one process: the growing displacement average rate might have several underlying processes.

With ‘this process’ we simply meant the fact that people are getting displaced through the expansion of urban gullies. As we further demonstrate and discuss in the text, this displacement can indeed be attributable to different underlying processes (e.g. the formation of new gullies, sidewall expansion and headcut retreat). Likewise, as discussed in the context of exposure, also population dynamics within the UG hazard zones can play a large role (cf. Fig. 5b).

To avoid confusion, this statement was reformulated.

155: “UG expansion events” – Unclear why these are events. Can this not happen also gradually?

Indeed. In the context of our analyses, an ‘event’ is an observed increase of >10m² in UG extent between two image dates. However, in reality, the cause of this increase may indeed be related to multiple individual events. Unfortunately, the infrequent nature of the available imagery does not allow to assess expansion events at the scale of individual rainfall events. To avoid confusion on this matter, this is now further clarified and defined in the relevant part of the Methods, i.e. section 2:

“We defined an event as an urban gully showing an increase in mapped extent of at least 10 m² between two image dates. However, it should be noted that this observed expansion may in reality be attributable to several consecutive, smaller expansion phases that occurred between the two image dates.”

155: “relatively limited” → “lower”.

We believe ‘relatively limited’ is a more suitable expression here, as ‘lower’ would imply that we first report a higher value (however, this only done in the following sentences).

160: “areal increases” might be better expressed in relative terms instead of absolute areas?

As already discussed above, we believe the absolute values are actually better suited here. It is the direct increase the gully area that leads to the displacement of people. This may be due to small UGs expanding very rapidly (high relative increase) as well as already very large gullies expanding further (low relative increase). For the resulting number of displaced persons, this does not really matter. In addition, as explained in the text, some observed expansion events relate to the formation of new

gullies. Such cases cannot be expressed relatively (i.e. that would imply an infinite relative increase event).

163: “hundreds of houses were indeed destroyed” with each event or altogether?

This was indeed somewhat vague. Following also other edits to the text, this is now reformulated to: “*Nonetheless, visual inspection on satellite images confirmed that often tens to hundreds of houses are destroyed by individual expansion events.*”

166: “formation of new gullies is observed relatively rarely ... and typically happens in areas with a lower population density” – This is an interesting point, and might be one key to understanding better the preparatory factors for urban gullies. This observation however also casts doubt on whether urbanization alone is responsible for forming these gullies.

The formation of new urban gullies indeed often starts in less dense, peri-urban areas and often don’t cause much damage in their initial phase. It is mainly their head retreat and sidewall widening that displaces people. We agree this is interesting and that’s also why we included Fig. 6 (currently Fig. S9) to explain and demonstrate this dynamic.

However, as already elaborately discussed above, we don’t agree that this causes doubt on the (already broadly accepted) idea that urbanization is directly responsible for their formation. Most of these urban gullies form at the edge of plateaus or at the bottom of initially poorly populated hillslopes because the urbanization on these plateaus or hilltops leads to a drastic increase in runoff (including through the effect of roads, discussed above). Surely, also other factors like topography and soil characteristics play an important role in this (cf. Fig. 3). However, the direct cause for the gully formation is clearly the increase in runoff production and accumulation due to urbanization. Under natural conditions, these soils can typically not generate much runoff (e.g. Moeyersons et al., 2005).

Apart from the overall low gully susceptibility/densities under rural conditions and the fact that the overwhelming majority of these gullies was not yet present before urbanization took place (cf. discussions above), the fact that the heads of these urban gullies also migrate in the direction of the urbanization (i.e. the place from where the runoff is coming; cf. Fig. 1; S9) provides further evidence for this.

168: “it is responsible” → “it was ...” or “it has been ...”. Delete “As such” here and elsewhere.

Indeed, we reformulated the sentence, avoiding the use of ‘as such’, we did the same at numerous other locations throughout the text.

171: “important” → “prevalent”?

That’s indeed a better choice of word. We changed important to prevalent.

172: “In addition, this typically occurs” – Wordy style.

Indeed, this was reformulated to: This widening often occurs ...

174: “importance seems to increase over recent years” – Judging from which metric?

This is clearly evident from Fig. 4b to which refer at the end of the sentence. More specifically, the inset of this figure clearly shows that the total sidewall expansion rate is almost 3x higher in the period 2020-2023 than in the three 5-year periods before (2005-2009, 2010-2014 and 2015-2019).

178/ Fig. 4: The boxplots seem to be truncated at lower end: please show the full distributions of the data; the numerous outliers deserve some mention.

Some boxplots are indeed truncated. However, we believe this is perfectly justifiable, given the logarithmic scale we use and the fact that these low values have no effect on our results (i.e. to the total expansion and displacement rates, displaced in Fig. 4b). More specifically:

- **For the observed expansion of the events (first set of boxplots), we indicated earlier that we only considered events with an observed total increase of 10 m^2 between two images. Hence, the lower limit of the y-axis of 10^{-3} ha ($= 10 \text{ m}^2$). Below this threshold, mapping uncertainties became too large.**
- **For the second set of boxplots (i.e. the population density in expansion zones), we set the axis limits to 0.01-10000 persons/ha. As the referee will appreciate, this lower value is already very low and indicates that in fact nobody lives in the expansion zone. Two, 2 of the >6000 sidewall expansion events had a displacement density of > 10000 persons/ha. However, as these displacements densities (associated with gully widening) were obtained by dividing the difference in population living in the gully extent polygons downslope of the gully head by the corresponding areal extent (which were in these cases very low), they can be clearly seen as outliers resulting from uncertainty propagation that have no effect on our total estimate. A remark was added in the methods (M4) to further clarify this.**
- **The third set of boxplots (the number of people displaced during events) reflects the total population living in the areas affected by gully expansion. Thus, events with a very low expansion and/or a very low population density will result in a very low number of people displaced by this event. Whether that number is 0.01 or 0.001 will have no influence here on our overall estimate of ca. 118,000 people displaced. This is especially so, taking into account the important uncertainty margins around these estimates (cf. Fig. 4b; Methods M6; See also discussion above).**

To be more transparent about this, the figure caption now clearly mentions that the axes of these boxplots do not show the full range of all values. In addition, the source data of these graphs (including values beyond these axes ranges) is made fully available.

Also please explain the symbols: what do the black diamonds show?

Indeed. This is now explained. The rectangle of the boxplots indicates the 25% and 75% quantiles of the data, while the black line indicates the median. Whiskers were calculated as 1.5 times the interquartile range. Black diamond indicates the average.

Do you assume that population density remained constant “between 2004 and 2023”?

No. As explained in the Methods (section M4), we each time used the population densities corresponding to the year in which the event was expected to have occurred.

The estimated displacement is then summed accordingly over this interval in the right-hand boxplot?

It is not entirely clear what the referee is asking here, but indeed: all events were ordered chronology and the cumulative sum of their areal expansion was plotted against the cumulative sum of their associated population displacement. This method is further explained in section M4.

Why would gully-head expansion affect areas with higher population density on average than gully side-wall expansion (A vs. B in middle boxplot)? Can you think of any physical or socio-economic explanation?

Interesting question. The dynamics of this displacement are further explained in the discussion section of our manuscript, where Fig. 6 (currently Fig. S9) provides a schematic representation of the process. In short, most gullies typically start at the bottom of hillslopes or plateaus in areas that are not yet strongly urbanized (hence, the lower population densities & initially very low population displacement rates). However, as these gullies expand through headcut retreat, they expand into more densely urbanized areas (e.g. higher up the hills or on the plateaus). Population densities are higher here, resulting in already more displacement. Meanwhile, sidewall expansion takes place along the whole gully (including the lower, less densely urbanized zones where the gully started). This results in a somewhat lower median population density for sidewall widening as compared to gully head retreat. Nevertheless, as also visible on the boxplots, there are important exceptions to this. Especially when the urban gullies are already somewhat older and urbanization is already well advanced, sidewall widening can also occur in very dense urban areas. The photos provided in Fig. 1 can further help clarifying this process.

As for why then most of the cities started developing on the plateaus or hilltops: also here, there are various reasons explaining this (e.g. relating to the colonial history of many of these cities, socio-economic stratification). It is outside our scope to discuss these in detail. Also the length restriction of the paper do not allow this. However, our recently accepted paper (Lutete Landu et al., 2025) as well as the work of Bédécarrats et al. (2019) and Kayembe wa Kayembe (2020) (cited in the text) provide some further insights into this for Kinshasa.

Fig. 4b shows that the mean rates of persons displaced by unit gully area are higher than the medians of population densities in Fig. 4a. The distributions are thus positively skewed, and the mean rates might be a poor measure of central tendency.

As discussed above, these distributions are indeed skewed. However, we do not see how this implies that the average reported rates are a poor measure. The values presented here show the relation between the total amount of expansion and the associated total number of persons displaced over recorded all events. The fact that these total values are somewhat influenced by a few extreme events (in terms of gully expansion and/or displacement rate) does not imply that these averages are not meaningful.

To draw a simple analogy: most of the landslides that occur may be small and affect only very few people; while some will be very large and/or occur in a densely populated area. As such, the total number of people affected by landslides will mainly depend on a few disastrous events. However, provided that the population of events is sufficiently large, this does not imply that it is meaningless to calculate the average number of affected persons per unit of area.

186: Please specify the width of the “confidence interval” here and elsewhere.

The confidence intervals are $\pm 40\%$. As the referee will have noticed, we refer here to the method section M6 where the size and justification of these intervals is explained in detail.

191: What are “indirect consequences” of urban gullies. Please give some examples. How many people in cities live less than 100 m away from river channels and face indirect consequences from floods?

A justification of this distance, including examples of indirect consequences and citations to earlier studies that report them are provided in the Methods, section M5 (to which we also refer in the text). For the referee’s convenience, we quote the relevant paragraph here again:

To account for various degrees of exposure, different buffers were considered. First, we applied a buffer distance of 100 m around the mapped contours of the gullies. While somewhat arbitrary, this distance provides an intuitively understandable estimate of the population for which the threats of UG expansion forms a regular and significant concern. These people are potentially exposed to direct impacts like damage to their property and/or displacement, but also to numerous indirect and intangible impacts (e.g. decreased housing property value, required investments in initiatives to counter further gully expansion, decreased accessibility, increased stress). Almost no data on these indirect impacts are currently available. However, our mapping efforts and field surveys indicate that this distance of 100 m is likely still a highly conservative value. For example, we observed that also people living several hundreds of meters away from an UG are involved in implementing measures aiming to stop gully expansion. Usually, they do so at their own expense (Lutete Landu et al., 2023; 2024).

As far as we are aware, there are no studies that assess the indirect consequences of flooding within such buffer distance in the DRC. However, we also believe it is outside the scope of this work to assess and discuss this.

192: “an estimated 3.24 million people lived within the potential expansion zone of UGs” – This number is nearly 1.2 times more people living within a 100 m of these gullies. Do you really expect that all gullies would expand by more than 100 m?

Yes, absolutely. This is not only expected, but has also been frequently observed. Some telling examples are already provided in Figure 1 (c&e + d&f). As the referee can see, these gullies expanded with several hundreds of meters. While some UGs expand at a slower rate, these cases are no mere outliers. Evidence for this is provided in Fig. S2, where we show the distribution of sizes and expansion rates for all UGs. The median length of an UG on their reference image (which is in many cases not necessarily their final length as these UGs may be recent) is ca. 150 m. However, numerous cases of over 1 km were recorded as well (Fig. S2-1). Similarly, many of the UGs show drastic expansion with gully head retreat rates well over 10 m/y (Fig. S2-4). It is important to note here that these rates were only calculated for UGs that could be observed over a period of min. 10 years. Hence, these high rates are not the result of short measuring periods with one extreme event. As detailed in the Methods section M5, the 95% quantile of linear gully head retreat rates on sandy substrates was 19.2 m/y. Over a period of 10 years (i.e. the relevant time interval of our exposure rates),

this corresponds to a linear expansion of 192 m. Gully widening happens of course at lower linear rates (and for them we delineated potential hazard zones based on observed max. widths; cf. Methods M5). Yet, the fact that long-term gully head retreat rates can indeed easily exceed 100 m/decade explains why the population living in the potential expansion zone of gullies indeed exceeds the number of people living < 100m away from a gully.

195: It might be useful to define here the difference between “potential” and “expected” zones. The latter has a statistical connotation.

Indeed. The difference is explained in detail in the Methods section M5, to which we refer here in the text. In short, potential expansion zones were based on the 95% quantiles of observed gully head retreat rates (over a 10-year period) and observed gully widths, making a further distinction based on the underlying soil type. The ‘expected’ expansion zones are based on the average retreat rates and max. widths. This means that the latter are typically much smaller (i.e. expose less people; cf. Fig. 5), but also involve greater risks. Estimates of the likelihoods that persons living in these different zones will indeed be displaced within the next 10 years are provided in the text.

196: “These risks remain difficult to quantify exactly” – And still you offer exact numbers of people affected. Why not add some error bars here?

As we indicate, these are estimates that rely on a comparison of the people exposed in 2020 and those already displaced between 2020 and 2023. The assumption behind this estimate here is that this ratio will also hold in the near future. While probably reasonably, we currently have no way of knowing this as the actual number of future displaced persons will depend on numerous factors that are simply impossible to predict (e.g. the expansion rates & expansion direction of individual gullies, population dynamics within the hazard zones, etc.). For example, as the inset in Fig 4b, displacement rates have recently been accelerating. It is currently unclear how this trend will continue. In addition, as discussed earlier, also assessing the exact uncertainty on the (changes in) population estimates is impossible. As such, quantifying a meaningful confidence interval on these numbers is impossible.

196: “comparing the exposed population in 2020 (Fig. 5a) with the average annual displacement rate in the period 2020-2023 (13,668 persons/year; Fig. 4b) indicates that 4.8% of the population” – If I compute $13,668 \text{ persons/year} * 10 \text{ years} / 3.24 \text{ million persons}$ (line 192), I obtain 4.2%. For the 2.74 million people living within 100 m of urban gullies (line 189), my result is 5.0% instead of the “5.9%” you state in line 200. The same goes for the 557,000 people in expected expansion zones: I obtain 25% instead of the “29%” reported in line 201. I don’t mean to split hairs here, but I would like to understand how you arrived at these estimates. I tried different values, so it cannot be a sole effect of rounding.

Thank you for double-checking this. However, as we here aim to assess which percentage of the exposed population in 2020 is indeed likely to be displaced within the next 10 years (based on the most recent average displacement rate of 13,668 persons/year between 2020-2023), we used the people exposed in 2020. In your calculations, you used the values of 2023 (i.e. our most recent estimates, which are indeed reported in the text). The amount of exposed population in 2020 can be estimated from Fig. 5 and is also reported in the supplementary tables (Tables S5-1,

S5-2 & S5-3) that report all values per period and city. More specifically, using the exposure values from 2020, one would obtain:

- For the 100m buffer zone around gullies: $(13,668 \times 10) / 2,323,412 = 5.9\%$
- For the potential expansion zone: $(13,668 \times 10) / 2,809,349 = 4.9\%$
- For the expected expansion zone: $(13,668 \times 10) / 471,540 = 29\%$

Note, however, that these values are now slightly revised in the text as we detected some inaccuracies in the population displacement of some events, linked to the GIS software algorithm used (see general remark at the start of this reply). The new values are respectively: 5.3%, 4.4% and 28%.

203: "This is true for" → "We observe this for".

This is indeed better. We changed the formulation as suggested.

204: "53-55% of this increase is attributable to population increases within hazard zones already present in 2010" – Unclear. Please explain.

As the referee will have noticed we here refer to Methods M5, where this calculation is explained in detail. In short, we estimated this percentage based on a counterfactual scenario analyses. In this scenario, we assumed that the different hazard zones as mapped in 2010 remained constant (i.e. as if no new UGs had formed since 2010 and existing gullies did not further expand) and looked to the population increase within those zones between 2010 and 2023. Comparing this increase with the total increase in exposure rates (cf. Fig 5a) thus indicates that (depending on the hazard zone considered) 53 to 55% of this increase is attributable to the fact that more people went to live in already existing hazard zones. Some implications of this are explained further down in the text.

208/Fig. 5: Panel a suggests that the potential expansion zone of urban gullies is >100 m throughout? Please comment on this.

Yes, see our further clarification above: more people live in the potential expansion zones of UGs than that live within 100 m of such UG. The reason for this mainly relates to the very high gully head retreat rates that these gullies may have (exceeding ca. 196 m/decade in 5% of the cases on sandy substrates) and is further discussed the relevant methodological section (M5).

You may also want to define briefly the "expected" expansion zone here. What is this expectation based on?

As the referee will have noticed, we refer to Method section M5. There we provide a full definition and explanation of the different types of hazard zones. Out of necessity to be brief and as providing an all-to-short description of what these zones represent may cause misunderstanding, we prefer not to further elaborate on this here.

Consider adding the number of people that were displaced by gullies over this time. Do I read the confidence intervals correctly that you cannot rule out that the exposed population has remained constant from 2010 to 2023?

Interesting question. The exposure rates of 2023 indeed fall within the upper limit of the estimated confidence intervals of the corresponding values of 2010. Nevertheless, the probability that exposure rates actually remained constant over these period will be close to zero. The reason for this relates to the way how these confidence intervals

were estimated (see Methods M6). As explained there and already discussed earlier in this reply, potential errors in the population density data product used are the main source of uncertainty. Based on comparisons with other products & census data for Bukavu, we estimated the range of uncertainty to be $\pm 40\%$ on the exposure and displacement rates reported. While this uncertainty margin is only a crude estimate it is, to our knowledge, the best possible assessment & likely conservative (i.e. most errors will likely be lower than 40%, especially at a country wide scale, cf. Methods M6).

On the other hand, the observed increases in major exposure rates are mainly driven by an increase in the spatial extent of the different hazard zones (i.e. due to the formation of new UGs & the further expansion of existing ones) as well as population increases in existing hazard zones. As Fig. 5b illustrates, each of these processes account for roughly half of the observed increase. As a result, a scenario where the exposure rates in any of these hazard zones did not increase between 2010 and 2023 would require that any increase in the extent of the hazard zones (which is directly based on our mapping of existing UGs and therefore highly certain) is compensated by a similar *decrease* in population density in those hazard zones. (E.g. if the extent of the hazard zone would increase with 50% between 2010 and 2023, the population density in these hazard zones would need to decrease with 50%). As Fig. 5b illustrates, this is clearly not what's happening and also all the Google Earth imagery clearly indicates that population density is increasing rather than decreasing. Likewise, such large-scale decrease would contradict all available evidence that Congolese cities are indeed growing (see e.g. work of Bédécarrats et al., 2019 and other sources cited in the introduction).

215: "Urban gullies as a new geo-hydrological hazard of the Anthropocene" reads catchy, but "new" hinges on the starting date of the Anthropocene. You may also want to demonstrate that these gullies did not exist previously.

Indeed, but we clearly believe we do demonstrate this. As already elaborated discussed elsewhere in this, reply, there is ample of evidence that these urban gullies are indeed caused by urbanization. The most direct evidence for this, is the fact that almost none of the 2,922 mapped urban gullies were already present on the aerial photos of the 1950s (cf. Methods M1; Fig. SI1). The dates of these photos clearly coincide with the currently accepted start date of the Anthropocene. As such, these UGs were clearly formed during the Anthropocene.

Furthermore, even if one would still be able to argue that these UGs are caused by natural processes (which is, based on the available evidence presented in this and earlier studies, highly unlikely), this would not undo the fact that they cause enormous impacts in the urbanized areas surrounding them. Hence, we believe this formulation is fully justified.

216: "massive scale of the problem of urban gullies" – What is the scale indeed? You have mostly reported absolute numbers, while a scale demands a ratio.

Indeed. While we also express these exposure rates as percentages at a later stage, the word 'scale' is in this sentence, strictly speaking, incorrect. We therefore replaced it by magnitude.

218: “clearly more than local” – Local can refer to cities, if you can exclude that gullies also form outside of cities.

With respect to the role of urbanization in causing UGs we refer to our replies to earlier comments. However, to avoid confusing, this was reformulated to “...UGs are clearly a widespread phenomenon.”

223: “estimated 1.1% of the total African population” – I doubt that you cannot compare the percentage of Congolese population to the African average, for reasons of spatial differences in both population density and distribution of hazard sources. It is nice to see some context of hazards, but this direct comparison is a bit weak.

We understand this concern. However, while this may indeed be somewhat comparing apples and oranges and this should not be driven too far, this comparison does show that the percentage of the Congolese population that is exposed is indeed significant.

The reason for this comparison at African scale is the lack of estimates at the scale of DRC. To our knowledge, no comprehensive studies exists that assess the impacts of or exposure to other hazards – other than the ones cited here. (Should the referee be aware of any, we would of course be happy to include them). The study of Pesaresi et al. (2017), which we cite here, only provides estimates at continental level.

To further illustrate this lack of data (as well as the significance of Urban Gullies), we consulted the International Disaster Database (EM-DAT). EM-DAT is widely used within the scientific community and generally considered the most comprehensive dataset on hazards and their impacts. On 5 March 2025, we consulted this database for DRC. According to this database (which the referee can also consult; <https://www.emdat.be/>), there were 49 landslide, flood, volcanic or earthquake disasters reported for DRC between 2004 and 2023 (i.e. the period over which we calculated displacement rates). These include 3 earthquake events (leading to 20 deaths and 35 injured people), 1 lava flow (claiming 32 lives), 11 landslide events (causing 426 casualties and 17 injured people) and 34 flood events of various types (claiming 3,981 lives and causing 1,596 wounded people). Summing up all these injured persons and casualties across all disasters, results in a total of 6,110 persons that were reported injured or killed by geo-hydrological hazards in the DRC.

This number (corresponding to 306 victims per year) is 5.1% of our estimated number of persons displaced by UGs over the same period. Evidently, being displaced does not imply also being wounded or killed. We also prefer not to include these statistics in our paper as they may lead to severe misinterpretations (e.g. we certainly do not wish to give the impression that UGs are 20 times more important than other geo-hydrological hazards in DRC). However, this quick comparison shows several things. First, the amount of studies and reliable data on (geo-hydrological) hazards is severely limited for DRC. This point was also already emphasized by Depicker et al. (2021) and several other articles we cite in the text. Second, even if these reported disaster impacts represent only a small fraction of the actual impacts, population displacement due to UG expansion is clearly highly significant in the DRC.

224: “major earthquake” – Needs a definition.

Given article length constraints – and as we believe this is not really essential for our discussion here – we leave it to the reader to consult the work of Pesaresi et al. (2017) for further details. However, to calculate this exposure rate, Pesaresi et al. used

estimates of the Maximum Peak Ground Acceleration (PGA) expected to occur with a 10% chance in 50 years. While this PGA value and map is commonly used in earthquake risk assessments, there is no specific earthquake magnitude associated with this value, as this is highly context-dependent.

226: “volcano” – active or extinct?

Active volcanos. This is now clarified in the text.

227: “likely much lower” can mean anything.

Indeed. As also stated in the original source (Pesaresi et al., 201), the return period of volcanic hazards can currently not be quantified in a reliable way across Africa. As such, the exposure value presented here reflects the fraction of the African population that lives within 100 km of a volcano that is deemed active. How active is unknown, but it is highly unlikely that each of these volcanos will erupt with a 10-year period. We can therefore not provide any further specification than this. Further discussing this is also outside the scope of our paper and impossible due to length constraints.

228-235: “56 persons per 100,000 urban inhabitants...” I am not sure whether comparing loss of houses to loss of lives is helpful here. You also rely on short-term rates that might be volatile, especially so for landslides.

Indeed. Yet, we believe that this is also properly acknowledged in the text. Moreover, as discussed a few comments earlier, reliable data on other geo-hydrological hazards are simply lacking for the DRC. Hence, the study by Depicker et al. (2021) cited here provides -to our knowledge - on of the few points of comparisons available.

236: “exposed population doubled between 2010 and 2023” with what (un)certainty?

See earlier discussion. While Fig. 5 provides uncertainty estimates on the exposure rates for individual years, we cannot assess the uncertainty on this relative increase. More specifically, this would require knowing the errors on the population density data in those individual years as well as their intercorrelation (e.g. to what extent does an overestimation in one area for 2010 also corresponds to an overestimation in 2023). As discussed in Methods M6 (and confirmed by referee 3), such information is not available. However, given that much (ca. half) of this increase is directly related to increases in the extent of hazard zones (i.e. due to the formation new UGs and the expansion of existing ones) and population estimates across different years are overall strongly correlated, we expect that the uncertainty on this relative increase will be much smaller than the $\pm 40\%$ used elsewhere.

238: “African urban population is expected to nearly triple by 2050” – Why not use the estimates for your study area instead?

The DRC is predicted to follow a very similar trend with the urban population expected to more than triple by 2050 (UN, 2019; See reference in text). However, we chose to report the value for Africa as we here aim to place our results in a broader context in this part of the text. As we indicate at the end of this paragraph, DRC is by no means the only African country affected by this problem.

242: “rainfall intensities in tropical Africa may increase by 10-15%” - And yet you stated earlier that rainfall was an uninformative predictor of gully activity (lines 124-125).

Indeed. Yet, as discussed above, some nuance is needed here with respect to the interpretation of our results. Our analyses show that spatial differences in average rainfall characteristics at decadal timescales (cf. Table S2) are insignificant in explaining spatial differences in the *susceptibility* to UG formation. This likely has multiple reasons, including: (i) the overall similar climatic conditions of cities across Congo (all characterized by a tropical climate); (ii) the fact that we compared 1-km² gridcells with and without UGs between but also within the same cities and their surroundings; and, linked to this, (iii) the low resolution and considerable uncertainties associated with this rainfall products. Furthermore, also earlier studies at continental scale (representing a much larger range in rainfall conditions) showed that rainfall conditions have a fairly limited effect on gully *susceptibility*. See e.g. the earlier cited work by De Geeter et al., 2023; as well as the review by Vanmaercke et al., 2021 (full references in text) for a more elaborate discussion on this matter.

However, such susceptibility assessments only indicate spatial differences in likelihood that gullies may form. They tell nothing about the actual probability that gullies will form during a certain period (*hazard*), nor about their potential scale or expansion rates. As gully formation & expansion is directly driven by rainfall (with especially average daily rainfall intensity being an important predictor; see e.g. Vanmaercke et al., 2016; Hayas et al., 2017), increases in rainfall intensities are likely to further aggravate the problem (in addition to increases in susceptibility due to continued urbanization). As such, our own results are clearly not in contradiction with this statement.

246: “increases could easily double gully expansion rates if other factors remain the same” – See previous comment.

See previous comments: there is an important difference between (urban) gully susceptibility (where spatial differences are not necessarily linked to contrasts in rainfall conditions, but mainly to local topography, soil characteristics and the presence of roads and buildings) and the formation or expansion rates of gullies (which are clearly linked to rainfall conditions).

248: “This is not only true for the DRC” – Strong statement. I think you refer to your inference here.

While we believe this statement is justified, we added ‘probably’ to indicate that this is indeed still an inference.

249: “The growing number of studies reporting on UGs in other countries seems to confirm this” – Not sure what this uncommented list of references confirms. Here, your discussion should pick up this previous work and build some context for your findings.

The studies cited here mostly present specific case studies of urban gullies and their impacts for different cities across Africa. As the referee will understand, it is outside the scope of this paper to fully review the findings of this earlier case studies. Furthermore, the length requirements of this type of article do not allow this. Yet, we find it relevant to cite them here to provide further evidence for the fact that the problem of UGs is not restricted to the DRC alone. The sentence was reformulated to make this clearer.

253-262: Consider shortening this paragraph; not all of it is relevant to your study.

Indeed. Some of this explanation was indeed less essential. We therefore significantly shortened this paragraph.

258: “As such, hillslopes and plateaus that are not prone to landslides or flooding, can be highly susceptible to UGs.” - How can you tell?

We believe this was evident from the explanation we provided (which is, following the referee’s earlier request, now reduced in length). Our analyses show that UGs mainly occur on sandier soils and on hillslopes that are somewhat steep, but not very (cf. Fig. 3 and the results of our susceptibility model + the discussion in this rebuttal relating to this). In reality, this means that many UGs form at the bottom of hillslopes and expand their way to hilltops or plateaus. From numerous earlier studies, it is well-known that landslides occur mainly on steeper hillslopes (typically on clayey substrates), while floods are more to be expected in valleys, e.g. near rivers. From this, it is clear that the hazard zones of UGs (which, e.g., often include plateaus) and floods or landslides do not necessarily overlap. This can have important implications for risk assessments and planning.

As this is indeed a less essential element in our discussion, and given the request to shorten this part, this is now no longer mentioned. The relevant statement is now formulated as: ... *hillslopes and plateaus that are not necessarily prone to landslides or flooding, may be highly susceptible to UGs*

259: Delete “certainly”.

Ok, done.

261: “This is especially so since, by nature, UGs tend to occur in densely populated areas, posing a direct threat for the surrounding population.” - Verbose. The first half of the sentence partly undermines your analysis of spatially intersecting gullies with urban areas: if you focus on urban gullies, they must occur in areas of high population density by definition.

Following the referee’s earlier request, this sentence was removed. However, as elaborately discussed at several other locations in this reply, our analyses (and several earlier studies) clearly show that urban gullies are indeed mainly caused by urbanization increases in built-up area and road construction. Furthermore, even if this would not be the case (i.e. if these would be gullies that simply happen to expand in urbanized areas), this would not lessen their impacts. Hence, we believe our focus on urban areas is fully justified.

263: “persisting problem” – It would be useful to learn about the lifetime of these gullies. Do you have any evidence about gullies that stabilised eventually?

Interesting question. As discussed in some of our earlier, field-based research in Kinshasa & Bukavu (Lutete Landu et al., 2023; Ilombe Mawe et al., 2024; see references in text), it is often very difficult to predict when an UG is actually stabilized, with long phases of stability suddenly disrupted by great expansion (e.g. due to an extreme rainfall event). We currently lack the (rainfall) data to make reliable estimates about this.

However, what is clear, is that UGs may continue to expand over the course of several decades. This is also evident from the data presented in this study, including the very high gully head retreat rates, even when quantified over periods of 10+ years (Fig. S2), as well as the continued (even accelerating) expansion and displacement of mapped gullies over the entire study period (e.g. Fig. 4).

266: “but mainly due to population increase” – Difficult to tell given the errors attached to Fig. 5a.

See reply to earlier comments: these uncertainty ranges present a conservative estimate on the absolute numbers (cf. Methods M6). However, relative increases (and their associated causes) are most likely robust. This statement simply follows from a decomposition of the attained increase.

269: “many UGs develop in overall poor and unplanned urban neighborhoods” – In your study area or in general?

Certainly in our study, but likely also in many other cases. We provided further references to support this idea. However, as more research would be needed to confirm this, we prefer not to further specify this.

279: “certainly” → “may”?

Ok, changed.

281-283: Redundant.

Ok, this part was removed.

286: “our work contributes to a better understanding of the lifecycle of UGs” – Where do the data show this?

This is evident from the whole of our analyses and results presented, which relied on the recurrent mapping of 2,922 UGs at different instances (often representing observation periods of 10+ years), comprising 8,161 individual expansion events. These data were summarized in Fig. 4, as well as in the numerous figures and tables included in the supplementary information. In addition, all mapped extents of UGs are made available to the referees and readers. It is based on these data that we present the generalized model shown in Fig. S9 and further discussed in this section. Yet, as this sentence appeared unclear to the referee it was rephrased.

286-297 and 298-310: These lengthy paragraphs also contain mostly repetitions; mind your overuse of “typically” and “significant danger”.

Given that this is the concluding section and our focus here is on discussing ways to prevent and mitigate UGs, we believe some repetition is permitted and even needed to present a clear synthesis. Nevertheless, we have reduced this as much as possible and shortened these paragraphs. In doing so, we limited the use of ‘typically’ and ‘significant danger’

332: “Our observations indicate that older UGs with vegetation tend to be more stable.” - Refer to figure that shows this.

This was observed for numerous UGs during our mapping efforts (once again: all mapped polygons are available, so readers & referees can confirm this for themselves). A detailed analyses of the vegetation dynamics within gullies in relation to their expansion rate was also outside the scope of this paper. However, a clear example of this can already be seen in Fig. 1 (d & f). The upper UG expanded relatively little between 2004 and 2018 and a more dense vegetation cover clearly developed in this gully. Meanwhile, the lower UGs showed much greater expansion rates and clearly showed little vegetation cover.

As requested, we now refer to this figure to further illustrate this point.

342: "Hence, as always, prevention will be better than curing". - Verbose.

Indeed. This sentence was reformulated.

376: "the gully needed to be located within 200 m of buildings" – This filtering might bias your data to high population areas?

See reply to earlier comments. We are confident that we provided ample of evidence that these urban gullies are caused by urbanization (including the analyses of aerial photos predating urbanization; confirmation that gullies occur only rarely in the surrounding non-urban areas; univariate + multivariate analyses of various potential controlling factors; and confirmation of this idea by earlier work).

Nevertheless, even in the extremely unlikely case that this inference would still be flawed (which would imply that similar gullies would also frequently occur in natural and/or rural settings), this would not invalidate the relevance, nor the major conclusion of our work: gullies occurring in urban settings in DRC have large impacts and threaten several million Congolese.

Referee #2 (Remarks to the Author):

This is the second time I've reviewed this paper. I was enthusiastic previously, and remain so. I think this is an important, creative, and well executed study. The few comments (regarding vegetation) that I had in my first review were addressed in the revision. I recommend that this be published. I did have two very minor comments:

Many thanks for the time and efforts you have spent in reviewing our work, as well as for the great suggestions (during this and the previous review round). This is greatly appreciated.

Line 93 - change "addressing" to "address"

Thanks. This is corrected.

Line 171-175 - is the widening just a geometric consequence of the gully deepening? Here I am imagining a triangular cross-section that just grows self-similarly. Or does the widening happen by some other process (such as slumping)? Anything you can add on this point is potentially valuable context.

Thanks for this very interesting question. The short answer to this is that we don't fully know. One key limitation from preventing us answering this question today is that the satellite products currently available do not allow to estimate gully depth.

However, based on the numerous field visits we conducted throughout the project, it appears that gully depth is indeed a crucial factor in determining gully width. Especially for urban gullies that formed on sandy substrates, cross-sections are indeed mostly triangular and gully top width is often strongly correlated to depth, as sidewalls tend to evolve to a slope of ca. 35° (i.e. close to the angle of repose). A typical example of this can be seen in Fig. 1a. However, it appears that the evolution of slopes towards this angle sometimes happens very fast and sometimes much slower. On more clayey substrates, gully walls are often more vertical (as is to be expected). However, also here, local widening can take place. In addition, it remains unclear what exactly control the depths of these gullies. E.g. our preliminary analyses, based on field surveys, indicated that gully depth is overall only very weakly correlated to the local relief.

Apart from gully depth and the angle of repose of the soil material, also the width and characteristics of the gully channel appear to play an important role. We expect that the width of this channel is generally controlled by the maximum peak discharges that occur, which are, in turn related to the runoff production in the upstream area. Our recently published paper on a few case study gullies in Kinshasa & Bukavu (Ilombe Mawe et al., 2024, Catena) provides some evidence for this. However, we also observed that some urban gully channels somewhat meander (despite their typically steep slope angle). We expect that this can greatly contribute to gully widening.

Unfortunately, length restrictions (and the fact that many observations are still provisional) do not allow us to elaborate much on these processes in detail. Nevertheless, this is clearly something we wish to address in follow-up studies.

Congratulations on a fantastic study. This was a pleasure to read and (more significantly) opened my eyes to a geomorphic process that I am familiar with but which is happening at a rate and scale and with a human impact that it is frankly astonishing.

Many thanks for your kind words. It is indeed our sincere hope that this paper will help raising attention to this hitherto severely underestimated problem.

Referee #3 (Remarks to the Author):

I appreciate the amount of effort and thought the authors put into revisions for this manuscript. I think the overall attention to suggested edits makes this a much stronger and engaging publication.

We sincerely thank the referee for the appreciation and their reviews of our manuscript, both of this version and the previous one. We believe, the suggestions provided have greatly helped in further improving the quality of our work.

One comment - in terms of M4 and M6, authors could also reference a known understanding with the unconstrained WorldPop data regarding underestimation in highly populated areas for rationale in data choice – see Deville et al., 2014, PNAS or Stevens et al., 2020, IJDE.

Thank you for this suggestion. We were not yet aware of these earlier assessments, but indeed: they further confirm our finding that WorldPop most likely underestimates population densities and, by extent, the number of displaced and exposed people. This work is now also discussed and cited in our discussion on error assessment, cf. Methods M6.

With M6, I'd adjust language to note that JRC GHS and unconstrained WorldPop data are informed by the same underlying census data – CIESIN GPWv4.11 – however, the modeling approaches are different.

Indeed, this is now formulated as such in the Methods, M6. In our opinion, this provides a further indication that our assessments of displaced and exposed population are still an underestimation (even with JRC-GHS data, as these census data may underestimate the actual population). However, we chose not to further emphasize or discuss this matter further, as we can only offer a comparison for Bukavu.

The additional effort to assess uncertainty using the Bukavu data as well as the acknowledgement that the gridded datasets must be interpreted with caution is commendable.

Many thanks for your appreciation.

Referees' comments:

Referee #1 (Remarks to the Author):

Review of manuscript #2022-09-14721B "The growing problem of urban gullies in Congolese cities" submitted by Guy Ilombe Mawe and colleagues

Dear Authors,

This is my second review of your study and therefore I shall skip yet another summary. However, I do owe you thanks to what must be the longest rebuttal letter I have had the opportunity to read through. Overall, I very much appreciate your detailed and clarifying replies to most of my remaining concerns. In hindsight, perhaps some of your explanations would have been more suitable for the main text in earlier versions of the manuscript (if not the supplementary file). However, seeing that your study seems to be in editorial favour, I will refrain from reiterating my reservations. That said, I believe that your study is innovative and sufficiently provocative. Below I offer a few suggestions (keyed to line numbers) that you may wish to consider for the final revision:

We thank the referee for their overall appreciation of our work as well as their detailed feedback during this and earlier review rounds. We believe these suggestions greatly contributed to the quality and clarity of our work.

We adopted all suggestions or provided a further clarification for a few cases where we believe the question may result from a misunderstanding. Overall, these changes are minor and did not change the content or conclusions of our work. The most noteworthy change is that, as requested, we now first standardized the data before fitting our logistic regression model (leading to the same findings, but overall better interpretable regression coefficients). Building on your earlier suggestions and questions, our updated model now also includes tree cover. Based on the principle of parsimony and given its strong correlation with built-up area and road density, we had initially removed this variable. However, following the criteria that are now more clearly described in Methods M3, it is indeed more consistent to include this and more easy to interpret. The updated model also performs slightly better than the previous iteration. Note, however, that also all earlier included variables remained highly significant (both for the full model and the Monte Carlo simulations). As such, this change has no further implication on our findings or conclusions. Rather they further confirm that, even after controlling for other land-cover related factors, the occurrence of urban gullies is strongly linked to the built-up area and road density.

A detailed response to the suggestions is provided below.

MANUSCRIPT FILE

21: "new geo-hydrological hazard" – Your study features features surprisingly little about the role of water other than what you cite from the literature. Also, the term "new" seems to ignore that papers about this topic (and area) have been around for some time (e.g. refs. 1-3, 21-23, 26, 28). What are "mega-gullies" in this context (refs. 1, 2)?

Thanks for these further suggestions, which are now incorporated.

- Regarding the role of water: there is indeed a lot more to be said about this. However, addressing the hydrological aspects of urban gullies was impossible due to a lack of reliable data at the scale of DRC. In addition, it was outside the scope of this study, which mainly aims to present an assessment on the scale and impacts of urban gullies.

- Regarding the term 'new': following the editorial suggestions, this word is now removed. As discussed in our earlier reply, we use this term to indicate that urban gullies are a relatively recent problem (in contrast to other hazards like floods, landslides, earthquakes and volcanos that have affected communities for millennia). Also, urban gullies are typically not yet considered in disaster risk reduction strategies or databases of disaster impacts. We fully acknowledge that other studies have already pointed to the occurrence of this problem (as the cited references indeed indicate). Yet, as clarified in the text, a systematic assessment on the scale and impacts of this problem was hitherto lacking.

- Regarding the term "mega-gullies": Makanzu et al. 2014 (REF 1) and Moeyersons et al. 2015 (REF 2) used the term to indicate large gullies in Kinshasa that have a top-width of >5 m. However, we avoided using this term in our text, as it may be considered somewhat ambiguous and overly dramatic.

88: "Nonetheless, steep and sandy soils are not a strict prerequisite." - Still, Fig. S2 distinguishes on the basis of "whether the city is located on sandy or non-sandy soil substrates", and in l. 647 you make the case for a significant difference, at least concerning gully width. On this occasion it would be helpful to reflect on the quality of the soil/sand data: the 30-m resolution indicated in Table S2 is quite optimistic.

We believe the reviewer may have somewhat understood this statement. While line 88 refers to factors explaining the occurrence of UGs, Fig. S2 refers to their size characteristics. Overall, our results show that UGs are significantly more likely to occur in sandy substrates. Therefore, this factor is also included in our logistic regression model. Yet, while UGs are significantly more likely to occur on sand, this is not a strict necessity and they can also form in non-sandy substrates (cf. Fig. 2). As such, we believe line 88 is correctly formulated. As we further argue in the text (and as Fig. S2 shows), UGs in non-sandy substrates also tend to be smaller than their sandy counterpart.

Nonetheless, with respect to the quality of the soil texture data (iSDA), we fully agree with this concern. It is indeed true that this dataset is subject to large uncertainties (especially in data-scarce regions like central Africa) and that the original resolution of 30m may give an overly optimistic impression. This is also why we did not use this dataset to differentiate between cities on sandy and non-sandy substrates. Instead, we used the *Soil* dummy variable, which is based on the geological map for the study area (cf. Table S2; Fig. 2). As this subdivision is done at the city level and corresponds to our own, field-based understanding of the study area, we are confident that this classification is more robust, reliable and less arbitrary than the sand content. Furthermore, using this dummy variable allows for a larger consistency between the results of our logistic regression model and those relating to size and expansion rates. Following also other questions regarding the construction of our model, this is now better clarified in the text (cf. Methods M3).

91: “98% of all mapped UGs were clearly connected to the local road network” – Still, the effect of road density seems minimal in your susceptibility model; please see comments below.

Indeed. As further discussed below, we initially calibrated our model without prior standardization. As such, the very low (yet clearly significant) coefficient of road density was attributable to the very high values that this variable could achieve (i.e. exceeding 30,000 m/km²). We now recalibrated the model and, as requested, first conducted a standardization of the data. As can be seen (i.e. Table S3, Fig. S3), the coefficient of this factor is now of a similar magnitude as that of the other factors included.

93: “Combined, slope steepness, soil type, the fraction of built-up area and road density well explain the presence or absence of UGs (Fig 3b, S3, S4)” – “Soil type” really means sandy fraction in the topsoil?

No, as explained in the text and Table S2, the variable *Soil* is a dummy variable indicating whether the pixel is located on the Kwango-Kwilu and Kasai Plateaus and likely has soils formed on sandy parent material (1) or not (0) (see Table S2). Evidently, this factor is correlated to the fraction of sand in the topsoil (Sand_iSDA). However, for reasons indicated by the referee and discussed above, we believe soil type allows for a more robust classification than Sand_iSDA (i.e. it is based on actual mapping, rather than machine learning). This is now also further clarified in the Methods (M3).

It is curious to see that built-up area has the second lowest weight in your susceptibility model (Fig. S3, Table 2); the effect of road density is even less pronounced (see below).

As further explained above and below, this was because we did not standardize the data before training the model, but used the values as such. This allowed -in our opinion- for a more straightforward model construction and interpretation. Yet, indeed, it does not allow a direct comparison of model coefficients. We therefore recalibrated our logistic regression model, using standardized data (see Methods M3, Table S3).

While refitting this model, we also decided to add ‘tree cover’ as a predictor variable. Initially, we had left this variable out as we strived to construct a parsimonious model, it is strongly correlated to other land cover use variables (BUA, road density) and only led to a small model improvement. Yet, following the criteria explained in Methods M3 and earlier questions of referee 1 and 2, it can indeed be included (we believe both model choices are justifiable here). It is important to note that also in this updated model with tree cover, the fraction of built-up area and road density remain highly significant in the model as well as in all Monte Carlo simulations (cf. Table S3). As such, this further confirms our earlier claims that the formation of urban gullies is indeed linked to ongoing processes of urbanization.

With respect to relative weight of built-up area and road density: following standardization, these weights are now clearly comparable to that of other factors. They may still be considered somewhat low. However, we see at least two reasons why this is the case. First, Road Density, BUA & Tree Cover are correlated and partially express the same thing. As such, the overall importance of land use/cover (changes) is divided over these three coefficients. Second, as explained in our Methods & our earlier replies, we explicitly sampled a number of cells without UGs inside cities, equal to the number of cells with an UG. This to make sure that also

conditions within urban environments are fully represented (in addition to cells outside the urban area). As these cells will have a high BUA and Road Density, this likely pulls down the value of these coefficients. Nonetheless, even so, they remain highly significant within the model.

96: “rainfall characteristics did not further improve our model” – Where do you show this?

This is now better explained in the Methods, M3. We applied a backward selection procedure to construct the logistic regression model. More specifically, we first started with all variables listed in Table S2 (except for Sand_iSDA; see above) and then systematically removed all variables that had a coefficient that was not statistically significant according the Z-statistic. This applied to all climate variables considered (see also reply to next comment).

97: “overall rather limited range in rainfall conditions across DRC” – This reads unduly simplistic, at least if considering that tropical rainfall meteorology is highly seasonal and influenced by the shifting of the ITCZ, jet streams, easterly waves, ENSO, etc.

As we showed and discussed in our previous reply, the range of the rainfall variables investigated is indeed limited. Meanwhile, earlier work (e.g. De Geeter et al., 2023) showed that rainfall variability has only a limited effect in explaining spatial patterns of gully susceptibility at continental scale (however: not gully expansion; cf. previous reply for a more elaborate discussion). Nevertheless, it is true that our analyses did not look at all aspects that determine rainfall patterns over DRC. The sentence was therefore rephrased to: “This is likely due to the overall rather limited range of the investigated rainfall variables across DRC as well as the limited spatial resolution and accuracy of the rainfall products available.”

140: “exactly” as in “accurately”? Yet, in l. 142 you estimate this to the nearest tenth of a percent (“4.4%”).

Indeed. We clarified the text accordingly by removing the word ‘exactly’ and reporting the estimated percentages as ‘around 4%’ and ‘ca. 5%’ (rather than to a tenth of a percentage).

162: “around 12,200 persons per year lose their house due to UG expansion” – This would be a good location to add some validation. How many reports do you have of people having lost their homes because of gullies. If unavailable, repeat for your readers here how many buildings or built-up area you can confirm to have been lost because of recent gully growth, judging from your remote sensing studies.

As explained in our previous reply, no reliable data on the impacts of UGs (or for that other matter any other hazard) exist for DRC (see e.g. the overview and comparison with the EMDAT database we presented). We therefore believe our work presents an important step in assessing the impacts of UGs.

Also with respect to our visual assessments of houses destroyed, we can unfortunately not offer much more than what is already presented in the manuscript. As already discussed elsewhere in the text, we frequently observed expansion events that destroyed dozens to of houses. However, mapping all destroyed houses would simply be impossible.

Nonetheless, some quick back-of-the-envelope calculations further indicate that our estimates are likely of the right order of magnitude. In total, we observed 953.3 ha of

gully expansion in 19 years (cf. Fig. 4, Table S4), corresponding to an average annual expansion rate of 50.2 ha/y. Assuming that every hectare of gully expansion contained on average 20 houses (see e.g. estimated parcel densities in Lutete Landu et al., 2025) and a reported average household size of 5.3 persons/household, this would correspond to $50.2 * 20 * 5.3 = 5321$ persons/y (a value that closely matches our estimated long-term average of 5930 persons/year).

Likewise, assuming this same average household size of 5.3, we can estimate that all urban gullies combined should have destroyed ca. 22,380 houses to attain our estimated amount of total displaced population (i.e. $118622/5.3$; cf. Table S4). This corresponds to an average of 7.66 houses per urban gully over their entire observation period. Based on our mapping experience as well as the context and enormous size of many of these gullies (cf. Fig. 1; Fig. S2), this clearly seems realistic to us.

173: “our analyses show that the occurrence of UGs is closely linked to built-up area and road density (Fig. 3, S1, S3)” – Yet, your susceptibility model has the lowest (absolute) regression weight by far for road density (Table S3 states “ $6,183e-5$ ”: please use proper exponential notation, and make sure you use correct decimal symbols). You mention the importance of roads several times in your study, but your susceptibility model fails to support this. If anything, road density has the weakest effect in this model. Any thoughts why this is so? The relative weight of sandy soils present seems to be much more prominent if readers were to take these coefficients at face value.

See reply to previous comments: this low value was linked to the fact that our variables were not standardized. In the recalibrated (standardized) model, the coefficient of Roads is clearly of comparable magnitude as the other variables (cf. Table S3).

201: “typically” – Avoid unduly generalising qualifiers: you state in your study that urban gullies are a “new” hazard.

Indeed. The word ‘typically’ was removed

210: “outskirt” → “outskirts”.

Indeed: corrected. Thanks for noticing.

216: “Once formed, most UGs continue to expand.” - Reiterates in parts the train of thought starting in l. 190: “UGs are -once formed- a persisting problem”.

Indeed, we have shortened this paragraph to avoid unnecessary repetition.

226: “As the gully head further migrates upslope, its contributing area –and by extent the amount of runoff that can accumulate at the head– will decrease, eventually halting the process.” - What about subsurface hydrology and stream piracy? Or are these irrelevant to urban gullies, where people interfere with the natural drainage frequently.

Interesting question. As argued in our previous reply, we believe subsurface hydrology has very little effect on the expansion mechanisms of urban gullies. More specifically, since most gullies form on very deep sandy substrates, we hypothesize

that most infiltrated rainwater vertically percolates to depths that far exceed the thalweg of the gullies and therefore not contribute to their expansion. While stream piracy may in some cases be (theoretically) possible, it was not observed. A more important mechanism is likely that people construct measures (e.g. dams) along roads to prevent that runoffs arrives at a gully head. In several cases, this led to the formation of new gullies (typically along the road just downstream of the dam).

These would certainly be interesting elements to further explore in the future. However, since (i) this is outside the scope of this paper, (ii) we currently lack the data to do so, and (iii) the need to keep the paper sufficiently brief, we prefer to not further elaborate on this here.

240: “vegetation may play a crucial role here” – Yes, I noticed “tree cover” in Table S2, but it seems that this predictor never made it into the model.

Indeed. As explained above, tree cover is now included in our updated model. While its effect is overall limited (due to intercorrelations with other land cover variables), this does lead to a somewhat higher model performance. While less parsimonious, this model is perhaps indeed easier to interpret.

Note that the significant but overall limited effect of vegetation is also in line with what we argue in this part of the paper (that deals with gully expansion rather than gully susceptibility): vegetation inside gullies may correlate to gully stability, but that the effect is not necessarily causal. As such, while focusing on different aspects of urban gully erosion, both elements seem to align (see also our reply to referee 2 for a more elaborate discussion on the role of vegetation).

465: “UG susceptibility model” – Please explain colour scale in caption.

Indeed, this is now explained in the caption.

480: “indicate the estimated confidence interval of $\pm 40\%$ (Methods M6)” – From your descriptions in Methods M6 I gather that these cannot be proper confidence intervals. Reword to “estimated overall error” or something similar.

Indeed. Throughout the text, we consistently renamed ‘confidence interval’ estimated overall error range.

567: “conducted these analyses at a resolution of ca. 1 km^2 (0.008333°)” - How can you convert unit area to unit angle? Even if using fixed geographic coordinates, the spatial resolution parallel to the equator will change with latitude.

Indeed this was somewhat confusingly formulated. With 1 km^2 , we meant pixels of $0.008333^\circ \times 0.008333^\circ$. For most cities (which are close to the equator), this corresponds roughly to pixels of 1 km^2 . However, this is indeed only true for the equator. This is now formulated more correctly here and throughout the text (where we refer to the resolution as being 30 arcseconds).

581: “differed significantly between cells with or without gullies” – So if any of the 1-km² grid cells contains at least one gully, it counts as a “gully pixel”? Fig. 2.4 shows that you can have many gullies in a given pixel. How do you account for this inflation effect, assuming that you cannot increase the spatial resolution of your model.

Indeed, sometimes this was the case. Cells with more than one gully head remained one and were considered only once when fitting the model. We have now formulated this more clearly in the Methods.

Note that this approach is similar to many other studies using logistic regression to model spatial patterns of susceptibility (e.g. landslide susceptibility), as it indeed avoids ‘inflating’ the model with overly similar positive cases. A possible downside of this strategy is that our model may be less precise in further differentiating within highly susceptible zones. Yet, overall, it presents the most simply and robust strategy to discriminate susceptible from non-susceptible zones at the scale of DRC.

582: “logistic regression model⁶³ that combines the variables best explaining observed differences between affected and non-affected cells” – How did you identify those “best explaining” variables? Table S2 lists 12 candidate predictors but nowhere did I find explanations about this selection. How did you counter excessive model tuning in your predictor search? Please also indicate whether you standardised the predictor values, which is really a must if you wish to compare coefficients in multivariate logistic regression.

Indeed. This is now fully explained in the Methods, M3. Values were not standardized before (as this was -to our understanding- not really a necessity for fitting the model in the case of logistic regression). However, as this is indeed a better practice, this is now done. Next, we applied a backward stepwise selection procedure, pruning out all variables that had an insignificant coefficient within the model. In addition, we removed the *sand* content variable for reasons discussed above (less reliable and strongly correlated to the *Soil* dummy variable, which is more robust and functionally expresses the same thing).

Overfitting was avoided in several ways. First, we chose logistic regression as it is already clearly less sensitive to overtuning/fitting than other machine learning approaches. Second, the significance of the variables included was checked based on the Z-statistic and further confirmed by our univariate analyses (cf. Methods M3). Finally, the robustness of our model and all factors included was tested through a Monte Carlo simulation. As Fig. S3 shows, our model performs robustly as it shows similar performance across all folds. Table S3 further shows that the range of fitted coefficients across the ten simulations never includes zero. This further confirms that our model is not overfitted and the presence of all variables included is justified.

583: “making sure each variable remained significant within the model” – How did you make sure. Did you use stepwise regression, etc.?

Indeed. This is now clarified in the Methods M3. We indeed applied a backward stepwise selection procedure, where the significance of the variables was tested based on the Z-statistic.

629: “While somewhat arbitrary, this distance provides an intuitively understandable estimate of the population for which the threats of UG expansion forms a regular and significant concern.” - It might be useful to highlight this assumption in the main text also (e.g. insert “arbitrary” in l. 134).

Ok, done.

647: “populations” should read “samples”.

Indeed. Corrected.

647: “considered the 95% quantile as the expected maximum width a gully may attain in such substrate” – It might turn out to be poor practice to set the “expected maximum” below the actually observed maximum, especially in hazard or exposure estimates. Fig. S2-3 indicates that gully widths might differ between cities. Would it make sense to adapt this “maximum” to the city level?

While it can indeed be argued that the potential hazard zones are still somewhat larger than suggested by this 95% quantile, we observed (as can also be seen in Fig S2-3) that some maxima far exceed the rest of the population, while the reasons for this are not always clear. Using these distances would also strongly increase the total exposed population, while being highly sensitive to this one observed maximum. Using the 95% quantile therefore results in somewhat more conservative, but also a more statistically robust estimates. A similar reasoning was followed with respect to defining hazard zones based on soil type, rather than per city. In principle, this could have been defined based on the level of cities. However, as several cities currently have only a few urban gullies that are 10+ years old, the observed 95% quantiles (and certainly maxima) would be subject to very large uncertainties.

648: “70.3 m” – Please avoid spurious values here and elsewhere, unless you can determine gully dimensions to the nearest decimetre.

Indeed. While the resolution of most satellite images uses was often sub-meter, mapping errors likely exceeded one decimeter in these cases. We therefore rounded these numbers (as well as similar distances further down in the text) to 1m. Note that we did not do this for the reported retreat rates, as the minimum observation period for determining retreat rates was ten years.

701: “Further verification in Google Earth (considering the number of houses) confirmed” – Assuming that the number of houses scales how with population density? Fig. S6 suggests that the systematic mismatch between the chosen two population datasets is at least of the order of 10,000 people for the year indicated. This is of the same order of magnitude that some of your annual exposure rates are of.

We believe the referee may be confusing a few different things here. The statement ‘further verification confirmed ...’ just refers to the fact that visually checking areas with a high population density according to the JRC GHS dataset but a low one according to WorldPop (cf. Fig. S7) were generally densely built-up. In line with the other studies cited here, this indicates that JRC GHS is likely more correct than WorldPop (which tends to underestimate population densities). This is indeed also what Fig. S6 shows. However, Fig. S6 deals with the total exposed population per city and per hazard zones; not the displaced population to which the referee seems to be referring. Also, as argued above in the text, we believe the WorldPop data is in many cases incorrect. Nevertheless, it is true that uncertainties can be considerable. This is now better acknowledged in the text.

723: “all estimates of displaced and exposed population are subject to 40% uncertainty” – Again, this is something to pronounce in the main text. Please also mention whether you believe this to be an absolute relative error or some sort of standard deviation, etc.

Indeed. Following also the request of the editor, we now better acknowledge this uncertainty throughout the text (e.g. by adding where possible the expected uncertainty range behind reported displacement and exposure values), including the introductory paragraph. This 40% can indeed be understood as relative errors. We

now clarified this better in the text.

SUPPLEMENTARY FILE

1: Please add article title, authors, etc. The figure numbering is a bit awkward with entries such as “Fig. S2-4” (l. 30) versus “Fig. S2.4” (l. 46). Please simplify if possible.

Ok, we added the article details (as known as they are known so far) to the first page of the document. Given the large amount of supplementary (sub)figures, the numbering can indeed be somewhat complex. However, we made sure all reporting is now coherent.

16: Typo on y-axis of Fig. S2-1.

Corrected. Thanks for noticing.

46: The mapped gully heads in Fig. S2.4 hardly stand out before the chosen colour scale. Please enhance contrast.

Indeed. The figures were remade, so the points stand out better.

Fig. S3: How can the logistic regression weights remain constant for a Monte Carlo simulation? Should these not also have errors? Explain what Y means. Note that predictions based on this model have higher uncertainties than the ones you give for the fit in Table S3. Stating the regression coefficients as is makes sense only if you have dimensionless predictors; however, Table S2 indicates units for each predictor.

As indicated in the caption, the equation shows the coefficients of the model fitted on the entire dataset (i.e. the most robust model possible). For the Monte Carlo simulations, we indeed recalibrated the coefficients of this model using different folds (subsets) and then independently tested it on the remaining subset (resulting in the ten ROC curves shown). These recalibrations indeed result in a range of coefficients, rather than one fixed coefficient. Following your suggestion, these ranges are now also reported in Table S3 (in addition to the earlier reported uncertainty range on the coefficients as based on the entire dataset). The meaning of Y (i.e. the simulated susceptibility of a cell) is now also clarified in the caption.